# Widespread detection of chlorine oxyacids in the Arctic atmosphere

Chlorine radicals are strong atmospheric oxidants known to play an important role in the depletion of surface ozone and the degradation of methane in the Arctic troposphere. Initial oxidation processes of chlorine produce chlorine oxides, and it has been speculated that the final oxidation steps lead to the formation of chloric ($HClO_3$) and perchloric ($HClO_4$) acids, although these two species have not been detected in the atmosphere. Here, we present atmospheric observations of gas-phase $HClO_3$ and $HClO_4$. Significant levels of $HClO_3$ were observed during springtime at Greenland (Villum Research Station), Ny-Ålesund research station and over the central Arctic Ocean, on-board research vessel Polarstern during the Multidisciplinary drifting Observatory for the Study of the Arctic Climate (MOSAiC) campaign, with estimated concentrations up to $7 \times 10^6$ molecule $cm^{-3}$. The increase in $HClO_3$, concomitantly with that in $HClO_4$, was linked to the increase in bromine levels. These observations indicated that bromine chemistry enhances the formation of OClO, which is subsequently oxidized into $HClO_3$ and $HClO_4$ by hydroxyl radicals. $HClO_3$ and $HClO_4$ are not photoactive and therefore their loss through heterogeneous uptake on aerosol and snow surfaces can function as a previously missing atmospheric sink for reactive chlorine, thereby reducing the chlorine-driven oxidation capacity in the Arctic boundary layer. Our study reveals additional chlorine species in the atmosphere, providing further insights into atmospheric chlorine cycling in the polar environment.

Active chlorine cycling has been found in the Arctic boundary layer during the springtime following polar sunrise and is acknowledged to play a key role in the depletion of surface ozone ($O_3$) in this region[1–3]. Chlorine atoms (Cl) are also a strong oxidant in the polar troposphere, where the levels of hydroxyl radicals, another major atmospheric oxidant, are relatively low[4]. It is also well established that the direct reaction with Cl provides a chemical sink of methane ($CH_4$) in the atmosphere[5–8]. The presence of chlorine species such as molecular chlorine ($Cl_2$) and bromine monochloride (BrCl) has been reported in the Arctic, attributed to direct emissions from snowpacks[9,10] and heterogeneous reactions of chlorine species on snow grains and airborne aerosols[11]. Upon photolysis, these chlorine species release reactive Cl atoms (1–2), which rapidly react with $O_3$ to form chlorine monoxide, ClO (3)[3]. ClO is subsequently oxidized by bromine monoxide (BrO) or ClO, producing chlorine dioxide (OClO); reacting with $HO_2$ to form hypochlorous acid (HOCl); or reacting with nitric oxide (NO) and nitrogen dioxide ($NO_2$) to produce Cl atoms and chlorine nitrate ($ClONO_2$) (as shown in reactions 4–7)[12]. Cl atoms can also degrade other hydrocarbons (RH) to generate hydrochloric acid (HCl; 8)[13]. HCl can be converted into chloride ($Cl^-$) via hydrolysis on aerosol surfaces (9)[14]. Chloride can further undergo heterogeneous reactions with HOCl to produce $Cl_2$ (10), which can, in turn, be photolyzed to recycle Cl atoms (1)[15].

$$Cl_2 + h\upsilon \rightarrow 2Cl \tag{1}$$

✉ e-mail: thamyj@mail.sysu.edu.cn; t.jokinen@cyi.ac.cy; a.saiz@csic.es

$$BrCl + h\nu \rightarrow Cl + Br \tag{2}$$

$$Cl + O_3 \rightarrow ClO + O_2 \tag{3}$$

$$ClO + BrO \text{ or } ClO \rightarrow OClO + Br \text{ or } Cl \tag{4}$$

$$ClO + HO_2 \rightarrow HOCl + O_2 \tag{5}$$

$$ClO + NO \rightarrow Cl + NO_2 \tag{6}$$

$$ClO + NO_2 \rightarrow ClONO_2 \tag{7}$$

$$Cl + CH_4 \text{ or } RH \rightarrow HCl + CH_3 \text{ or } R \tag{8}$$

$$HCl \leftrightharpoons Cl^-(aq) \tag{9}$$

$$Cl^-(aq) + HOCl + H^+(aq) \rightarrow Cl_2 + H_2O \tag{10}$$

Despite decades of research on chlorine cycling in the atmosphere, a largely unexplored aspect entails the formation of chlorine oxyacids, such as chloric ($HClO_3$) and perchloric ($HClO_4$) acids. The presence of atmospheric $HClO_4$ was first proposed to be important in the polar stratosphere and is believed to be a missing atmospheric sink process of chlorine[16–18]. Recent studies have hypothesized the potential formation of $HClO_3$ and $HClO_4$ in the lower atmosphere through observations of significant chlorate ($ClO_3^-$) and perchlorate ($ClO_4^-$) levels in rainwater, snow, and Arctic ice core samples[19–22]. Therefore, the atmospheric occurrence of chlorine oxyacids could enhance the chlorine sink, thereby affecting the oxidation capacity of the atmosphere and potentially posing environmental threats once deposited to the Earth's surface. However, to date, there exists no direct evidence of the presence of $HClO_3$ and $HClO_4$ in the atmosphere, thus, limiting our full understanding of the atmospheric chlorine cycle and its associated environmental impacts.

Here, we present ambient observations of $HClO_3$ and $HClO_4$ in the atmosphere. Measurements were made via mass spectrometry in the Arctic at the Villum Research Station, Greenland, Ny-Ålesund, Svalbard, and over the central Arctic Ocean onboard research vessel (RV) Polarstern during the Multidisciplinary drifting Observatory for the Study of the Arctic Climate (MOSAiC) expedition. The measurements show that both chlorine oxyacids are ubiquitous and widespread during spring in the Arctic region. We find that these atmospheric species are not photoactive and therefore represent a previously unconsidered atmospheric sink of reactive chlorine in the pan-Arctic boundary layer.

## Results and discussion

### Observations of gas-phase $HClO_3$ and $HClO_4$ in the Arctic

Figure 1 shows the time series of $HClO_3$ and $HClO_4$ measured at the Villum Research Station, Greenland, and during the MOSAiC campaign. Our observations in Greenland indicated a significant increase in the $HClO_3$ signal measured with a nitrate-chemical ionization atmospheric pressure interface time-of-flight mass spectrometry (CI-APi-TOF; Methods), with an estimated concentration up to $1 \times 10^6$ molecules cm$^{-3}$ in the spring of 2015. The $HClO_3$ concentration began to increase when sunlight increased towards the end of February. $HClO_3$ exhibited no diurnal pattern, but a unique feature is that a significant increase in $HClO_3$ concentration was observed in coincidence with the depletion of $O_3$,

as shown in Fig. 1a. Typically, $HClO_3$ peaked under relatively low $O_3$ levels (<30 ppb). We also measured $HClO_3$ with a nitrate CI-APi-TOF instrument during the MOSAiC expedition in 2019/2020 (Methods section). Similar to the observation in Greenland, the measurements onboard RV Polarstern in different seasons revealed a clear increment in $HClO_3$ starting at the end of February, when solar radiation started to increase after the polar night. The estimated springtime concentration of $HClO_3$ during the MOSAiC campaign ranged from approximately $1 \times 10^5$ to $7 \times 10^6$ molecules cm$^{-3}$ (Fig. 1b). An increase in $HClO_3$ was also observed in coincidence with the depletion of $O_3$ over the Arctic Ocean during the MOSAiC campaign. The $HClO_3$ levels are relatively low in the other seasons, with concentrations near detection limits of $\approx 10^4$ molecule cm$^{-3}$ (Supplementary Fig. S1). Further measurements at Ny-Ålesund, Svalbard also indicated the presence of $HClO_3$ in springtime, with concentrations up to $8 \times 10^5$ molecules cm$^{-3}$ (Supplementary Fig. S2). However, without direct measurement of $O_3$ during the campaign at Svalbard, we are not able to evaluate the relationship between $HClO_3$ and $O_3$.

As shown in Figs. 1, 2, the increase in $HClO_3$ was accompanied by an increase in $HClO_4$. The $HClO_4$ concentrations in Greenland and MOSAiC were estimated to be in the range of near detection limits ($7 \times 10^3$) to $8 \times 10^4$ molecules cm$^{-3}$ and near detection limits ($3 \times 10^4$) to $1 \times 10^6$ molecules cm$^{-3}$, respectively, during springtime, which were typically lower than the $HClO_3$ concentration. The lower concentration of $HClO_4$ compared with that of $HClO_3$ is consistent with the levels of $ClO_4^-$ and $ClO_3^-$ measured in Arctic ice cores, where the annual depositional flux of $ClO_4^-$ was reported to be several times lower than that of $ClO_3^-$ (refs. [21,22]).

The results obtained during our campaigns at the different locations and time periods over the Arctic demonstrate that these chlorine oxyacids are widespread (Fig. 1 and Supplementary Fig. S2), and that their presence is a common phenomenon in the Arctic boundary layer during the springtime. The question arising here concerns the mechanisms leading to the occurrence of $HClO_3$ and $HClO_4$ in the Arctic.

### Potential formation mechanism of atmospheric chlorine oxyacids

Although the initial steps of atmospheric chlorine oxidation are well understood (1–10)[13,14], the final oxidation steps leading to chlorine oxyacid formation are not well characterized. Here, we explore the potential formation mechanisms of the observed $HClO_3$ and $HClO_4$ during spring in the Arctic.

As shown in Figs. 1, 2, the increase in both $HClO_3$ and $HClO_4$ coincides with the decrease in the $O_3$ concentration. Springtime atmospheric surface ozone destruction is a well-known phenomenon in the Arctic and is typically linked to the shallow mixing layer and chemical reactions, including bromine and chlorine chemistry[14,15,23,24]. Our air mass backward trajectory analysis revealed that chlorine oxyacid-laden and $O_3$-depleted air mass originated from the near ground surface, while the high $O_3$ air mass originated from higher altitudes (Supplementary Fig. S3). This indicates that the ground surface, such as snowpacks in the Arctic, may play a role in the observed increases in $HClO_3$ and $HClO_4$ levels. It has been suggested that the heterogeneous reaction of $O_3$ on chloride-containing aqueous and salt surfaces constitutes a potential formation mechanism of $ClO_3^-$ and $ClO_4^-$ (refs. [25–27]). However, the formation of $HClO_3$ and $HClO_4$ via heterogeneous reactions on the aerosol surface or direct emission from the surface of snowpacks is likely not the dominant pathway in the Arctic. This assumption is justified by the remarkably low vapor pressure (6.8 mm Hg under a 70% concentration, at 298 K)[28] and high Henry's law constant ($K_H$) of $HClO_4$ ($9.9 \times 10^3$ mol m$^{-3}$ Pa$^{-1}$)[18]. Although information for $K_H$ of $HClO_3$ is not available, its value is very likely in between $K_H$ of $HClO_4$ and $K_H$ of $HOCl$ (6.5 mol m$^{-3}$ Pa$^{-1}$)[29]. These low

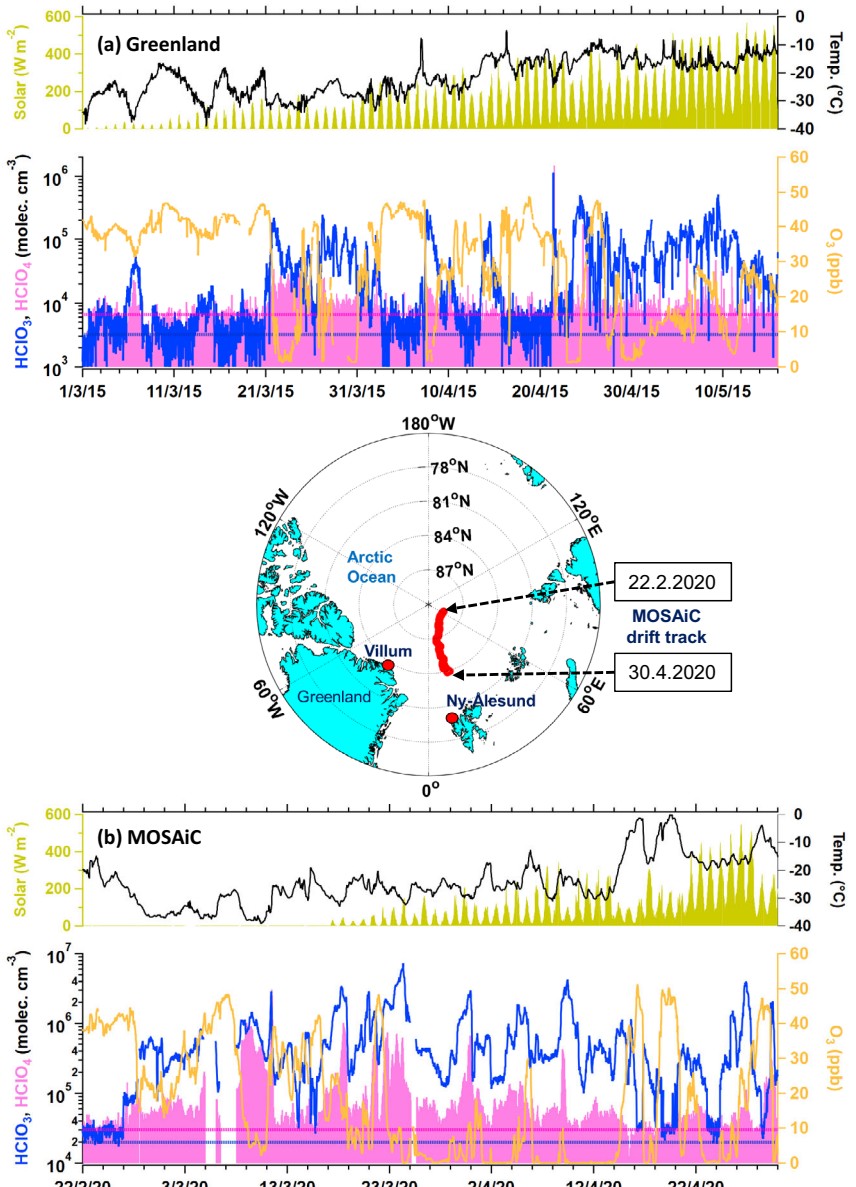

**Fig. 1 | HClO$_3$ and HClO$_4$ over the Arctic.** Time series of HClO$_3$, HClO$_4$, and O$_3$, together with the temperature and incoming solar radiation measured at **a** the Villum Research Station from 1 March–15 May 2015 and **b** during the MOSAiC expedition from 22 February–30 April 2020. The data are displayed at a 30-min-average resolution, and any gaps in the time series are the results of instrumentation offline and maintenance periods. The dashed line represents the detection limits for HClO$_3$ (blue) and HClO$_4$ (pink) measurements. The uncertainty of HClO$_3$ and HClO$_4$ measurements was estimated to be at least a factor of two (see Methods). The map shows the location of the Villum Research Station (Nord) in Greenland, Ny-Ålesund in Svalbard, and RV Polarstern passive drifting track across the Arctic Ocean during the springtime sampling period. Note that all the time reported here is in Coordinated Universal Time (UTC). The map was created by the authors using MathWorks MATLAB (https://www.mathworks.com/products/matlab.html).

vapor pressure and high $K_H$ suggest that the formed ClO$_3^-$ and ClO$_4^-$ on the aerosols or snow surface are unlikely being emitted directly as gas-phase HClO$_3$ or HClO$_4$ into the atmosphere. This is further supported by the detection of low HClO$_3$ and HClO$_4$ atmospheric concentrations in winter when the Arctic is covered by snow (see Supplementary Fig. S1). Furthermore, the observed lack of a clear pattern between HClO$_3$ and HClO$_4$ and the aerosol surface area during the springtime (Supplementary Fig. S4) may point to their limited partitioning from the aerosol phase. Another previously suggested potential HClO$_4$ formation pathway via the heterogeneous reaction of ClO with sulfuric acid (H$_2$SO$_4$)[30] may also not be important, as the results demonstrated no direct relationship between HClO$_3$ (or HClO$_4$) and our measured

H$_2$SO$_4$ concentrations (coefficient of determination, $R^2 \leq 0.04$) during both the Greenland and MOSAiC campaigns (Supplementary Fig. S4).

Here, we propose a more likely formation mechanism of HClO$_3$ and HClO$_4$ over the Arctic environment during springtime, as illustrated in Fig. 3. The snowpack emissions of Cl$_2$ and BrCl[9–11] undergo fast photolysis, leading to the production of Cl atoms, which subsequently react with O$_3$ to form ClO (1–3)[3]. In addition to photolysis, the produced ClO can then react with BrO/ClO to produce OClO, or undergo loss through reactions with OH, HO$_2$, NO and NO$_2$, CH$_3$OO, and CH$_3$COOO[2]. Abundant BrO and ClO must have been present during the encountered ozone-depletion events, as have been previously

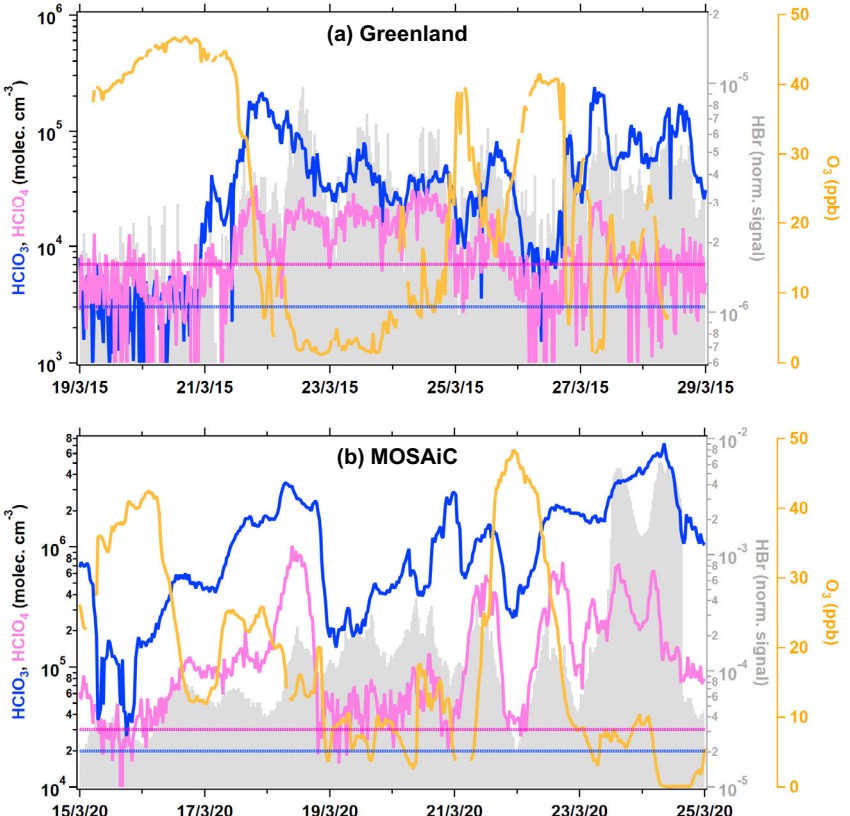

**Fig. 2 | Relationships between HClO₃, HClO₄, O₃, and bromine chemistry.**
Expanded view of HClO₃ (blue solid line) tracking with the HClO₄ (pink solid line) and HBr (gray shaded-area; based on the Br⁻ normalized signal from nitrate CI-APi-TOF measurements which is most likely HBr (refer to Methods) to represent bromine chemistry), at **a** the Villum Research Station, Greenland, from 19 to 29 March 2015; and **b** onboard RV Polarstern during the MOSAiC campaign, from 15 to 25 March 2020. The dashed line represents the detection limits for HClO₃ (blue) and HClO₄ (pink) measurements. The uncertainty of HClO₃ and HClO₄ measurements was estimated to be at least a factor of two.

demonstrated by many studies[2,3,31–37], and significant levels of BrO have been observed in spring during the MOSAiC campaign[38]. By using the previously reported typical ranges of BrO, ClO, and HO₂ levels during Arctic ozone-depletion events[11,32–40], we estimate that the reaction rate of ClO + BrO is much higher than that of the ClO + ClO and ClO + HO₂ channels (section S1 in the Supplementary Information, SI), suggesting that the increase in BrO during ozone depletion events drives the OClO formation. The reaction of ClO + OH, ClO + CH₃OO, and ClO + CH₃COOO are insignificant; however, the presence of typical levels of NOₓ (NO and NO₂) in the Arctic (i.e., 1–40 ppt) can compete with BrO for ClO (section S1 in SI). Indeed, previous ground measurements have detected significant OClO, up to 24 ppt, in the Arctic springtime[35]. Further observational evidence for the key involvement of bromine chemistry in the chlorine oxyacids formations comes from our observations which demonstrated that the recorded bromide signal adhered to the increase in HClO₃ and HClO₄ combined with a drastic decline in the O₃ concentration (Fig. 2 and Supplementary Fig. S5).

OClO can undergo further reactions, including (i) reacting with Cl to yield two ClO molecules; (ii) oxidation by OH to form HClO₃; (iii) reacting with NO to recycle ClO; and (iv) oxidation by O₃ to produce ClO₃ (Fig. 3). Among these reactions, OClO + OH exhibits the fastest rate, with $k_{OClO+OH}[OH]/k_{OClO+Cl}[Cl]$ and $k_{OClO+OH}[OH]/k_{OClO+O_3}[O_3]$ ratios calculated to be in the range of $2 \times 10^{-1} - 1 \times 10^{4}$ and $1 \times 10^{3} - 1 \times 10^{6}$, respectively, while the $k_{OClO+OH}[OH]/k_{OClO+NO}[NO]$ ratios fall in the range of $6 \times 10^{-1} - 3 \times 10^{1}$ (section S2 in the SI). These results suggest that a significant fraction of OClO can be directly oxidized by OH to convert into HClO₃, and produce ClO to recycle OClO.

This is consistent with previous experimental studies on the OH + OClO reaction, where HClO₃ was suggested to be produced at low temperatures[41,42]. Despite the lower O₃ concentration (a source of OH), previous studies have shown that HOₓ (OH and HO₂) chemistry is active during springtime in the Arctic with a reported OH concentration of ≈10⁵–10⁶ molecules cm⁻³ (refs. 2,4,43). This can also be indicated by our observation of significant H₂SO₄ concentrations (Supplementary Fig. S4), and the previously reported increase in methane sulfonic acid (MSA) levels[44,45] during the Arctic springtime, both of which are products of sulfur oxidation reactions with OH and BrO. Therefore, OH is not a limiting factor of HClO₃ formation in the Arctic during springtime. Given the fast reaction rate of OClO + OH, sufficient OH concentration, and enhanced OClO formation due to the increase in BrO during ozone depletion events, HClO₃ production can occur efficiently. As to HClO₄, the limiting factor of formation is likely the ClO₃ concentration (Fig. 3), most likely due to its slow formation process via OClO + O₃ (ref. 46). Therefore, the HClO₄ formation is likely regulated by the O₃ levels as indicated by the higher mean HClO₄ concentrations observed at relatively higher O₃ levels during the depletion events (Supplementary Fig. S6).

Based on these results, we conclude that the observed HClO₃ and HClO₄ over the Arctic atmosphere are predominantly produced through homogeneous reactions of chlorine, involving photochemical processes of HOₓ and bromine chemistry.

## Atmospheric fate of HClO₃ and HClO₄
The fate of chlorine oxyacids determines their importance in the atmosphere. We first evaluate the potential removal of HClO₃

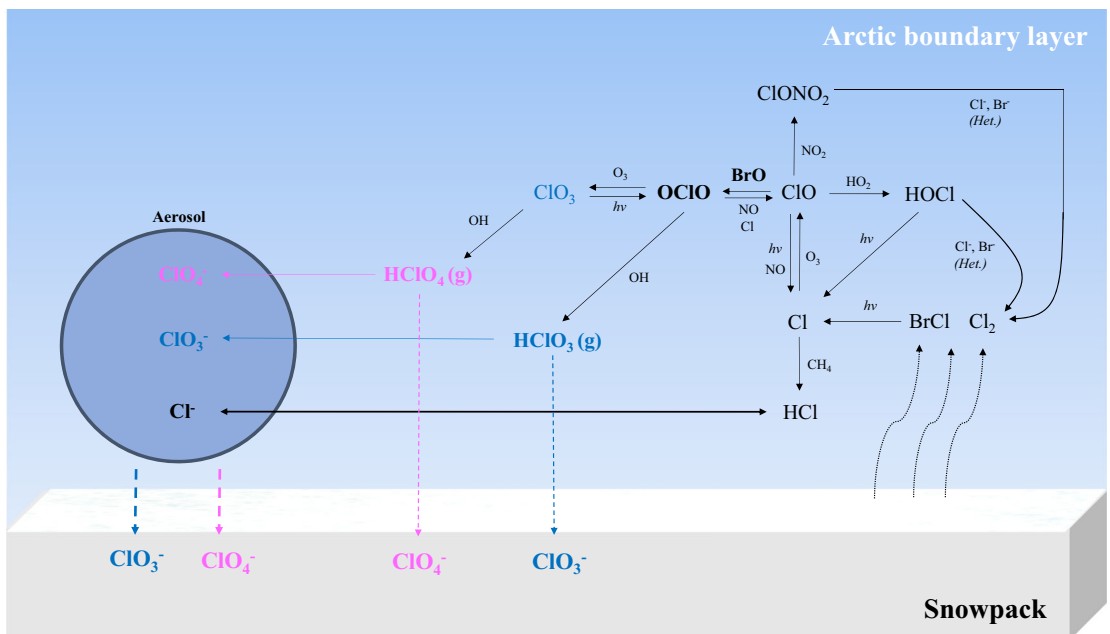

**Fig. 3 | Atmospheric formation and the fate of HClO$_3$ and HClO$_4$.** Simplified diagram of the proposed potential formation mechanism of gas-phase HClO$_3$ (blue) and HClO$_4$ (pink) in the Arctic boundary layer during springtime after polar sunrise. The produced HClO$_3$ and HClO$_4$ can be taken up by the surface of aerosols and converted into ClO$_3^-$ and ClO$_4^-$, respectively. The deposition of aerosols and/or the direct deposition of gas-phase HClO$_3$ and HClO$_4$ onto the ground surface, such as snowpacks, can function as a sink for reactive chlorine in the Arctic troposphere. The reactions are based on the literatures[2,12–14,64]. The mean boundary layer height was reported to vary between 100 and 200 m during the MOSAiC campaign[74,78].

and HClO$_4$ in the troposphere through photodecomposition. Supplementary Fig. S7a shows our computed UV–Vis absorption spectra and cross-sections of HClO$_3$ and HClO$_4$ within the 170–340 nm wavelength range (refer to the Methods section for details). At the relevant wavelengths under tropospheric conditions (>290 nm), the estimated cross-sections of HClO$_3$ and HClO$_4$ are very small, suggesting that these two chlorine species are not photolabile in the troposphere. Based on these cross-sections, the loss rate constants of these oxyacids against photolysis at noon in the Arctic springtime are calculated as $4.4 \times 10^{-12}$ and $2.5 \times 10^{-18}$ s$^{-1}$ for HClO$_3$ and HClO$_4$, respectively (Supplementary Fig. S7b, c).

Another possible removal pathway of HClO$_3$ and HClO$_4$ in the troposphere is their reactions with OH and Cl. Although there are no rate constants available for the reactions of HClO$_3$ with Cl and OH, the reaction rate of HClO$_3$ with either Cl or OH is expected to be low since both barriers for hydrogen abstraction from HClO$_3$ are high[47]. The reactions of HClO$_4$ with Cl and OH radicals (11 and 12, respectively) are also slow at low temperatures, with reported reaction rate coefficients of $1.00 \times 10^{-31}$ and $5.8 \times 10^{-13}$ cm$^3$ molecule$^{-1}$ s$^{-1}$ at 253 K, respectively[48]. Assuming typical Cl ($4 \times 10^5$ molecules cm$^{-3}$)[3] and OH ($5 \times 10^5$ molecules cm$^{-3}$)[4] concentrations in the Arctic, the loss rates of $k_{\text{Cl+HClO4}}[\text{Cl}]$ and $k_{\text{OH+HClO4}}[\text{OH}]$ are estimated to be $4.0 \times 10^{-26}$ and $2.9 \times 10^{-7}$ s$^{-1}$, respectively.

$$\text{Cl} + \text{HClO}_4 \rightarrow \text{HCl} + \text{ClO}_4 \qquad (11)$$

$$\text{OH} + \text{HClO}_4 \rightarrow \text{H}_2\text{O} + \text{ClO}_4 \qquad (12)$$

Given the presence of significant aerosol particle surfaces (up to 100 μm$^2$ cm$^{-3}$) and humidity (which begins to increase in April) in the Arctic troposphere (Supplementary Fig. S8), once HClO$_3$ and HClO$_4$ are formed, they can undergo heterogeneous uptake on the surface of aerosol particles. Although there is no direct information available on the heterogeneous uptake coefficient (γ) of HClO$_3$ and HClO$_4$, previous studies have reported that the

heterogeneous uptake coefficient of other chlorine acids, such as HCl, on aqueous aerosols reaches ≈0.2 at 273 K[49,50]. Both HClO$_3$ and HClO$_4$ are very strong acids with high electronegativity and can thus be easily ionized into ClO$_3^-$, ClO$_4^-$, and H$_3$O$^+$ in liquid water of the aerosol since HClO$_3$ and HClO$_4$ are highly soluble in water. The $K_H$ value of HClO$_4$ in water was reported as $9.9 \times 10^3$ mol m$^{-3}$ Pa$^{-1}$ (ref. [18]), while the $K_H$ value of HClO$_3$ likely varies between the $K_H$ values of HClO$_4$ and HOCl, with that of the latter compound reaching 6.5 mol m$^{-3}$ Pa$^{-1}$ (ref. [29]). These higher $K_H$ values than that of HCl ($K_H = 0.2$ mol m$^{-3}$ Pa$^{-1}$) may indicate that HClO$_3$ and HClO$_4$ could be efficiently accommodated on the surface of aerosol particles and that the fraction evaporating back into the gas phase could be small as well. By assuming that the heterogeneous uptake is accommodation limited and the γ values of HClO$_3$ and HClO$_4$ are similar to that of HCl (γ = 0.2)[49,50], the estimated heterogeneous loss rate coefficients of HClO$_3$ and HClO$_4$ based on a typical aerosol surface area of 20 μm$^2$ cm$^{-3}$ during the MOSAiC campaign are $2.7 \times 10^{-4}$ and $2.5 \times 10^{-4}$ s$^{-1}$, respectively (section S3 in SI). These rates are much (>3 orders of magnitude) higher than the rates of photodecomposition and radical attack (by OH and Cl) estimated above (<$3 \times 10^{-7}$ s$^{-1}$). Therefore, the most relevant fate of HClO$_3$ and HClO$_4$ is their heterogeneous uptake by the surface of aerosol particles and subsequent deposition on the ground surface or undergo wet deposition. However, we cannot exclude the possibility of direct loss of these chlorine oxyacids to the snow surface (Fig. 3). In fact, our hypothesis is supported by previous studies in polar regions that have measured a considerable amount of ClO$_3^-$ and ClO$_4^-$ in ice cores[21,22], snow[51], and aerosols[52], where atmospheric sources are strongly implicated.

Atmospheric chlorine chemistry has been regarded as a "never-ending" reaction since there is no termination process in the cycle. Indeed, the formation of HCl can serve as a sink for chlorine compounds in the troposphere, where HCl is taken up by aerosols and converted into Cl$^-$, followed by an atmospheric deposition process[14,53]; although, in the presence of NO$_x$ and reactive halogens (i.e., HOI and

HOBr), $Cl^-$ can be efficiently activated into reactive gas-phase chlorine again[54–56]. However, as $HClO_3$ and $HClO_4$ are not susceptible to photolysis and radical attack, and their conversion into $ClO_3^-$ and $ClO_4^-$ on aerosol surfaces or snowpacks is efficient in the Arctic boundary layer, the homogeneous formation of $HClO_3$ and $HClO_4$ could terminate chlorine recycling. Therefore, the formation of $HClO_3$ and $HClO_4$ is expected to affect the chlorine-mediated oxidation capacity in the Arctic troposphere. Furthermore, once $HClO_3$ and $HClO_4$ deposit on the ground surface (i.e., snowpack and sea ice), they may have environmental implications as their ions, $ClO_3^-$ and $ClO_4^-$, can accumulate in the polar ice and marine sediment, and may present a toxicity risk to resident biota[57,58].

In summary, our study revealed the observations of $HClO_3$ and $HClO_4$ in the atmosphere and their widespread occurrence over the pan-Arctic during spring. We propose a novel plausible mechanism for the formation and loss of chlorine oxyacids in the Arctic environment. The results provide evidence for chlorine oxyacids to be a previously unconsidered atmospheric sink for reactive chlorine, thereby providing further insights into chlorine chemistry in the Arctic region. We, therefore, conclude that the existence of $HClO_3$ and $HClO_4$ in the atmosphere should be considered when evaluating the environmental impacts of chlorine chemistry in the Arctic.

## Methods

### Sampling locations

This study comprises data obtained during three field measurement campaigns over the Arctic within different time ranges. We conducted a measurement campaign at the Villum Research Station (Station Nord) in high Arctic Northern Greenland (81° 36' N 16° 39' W). This campaign started in mid-February 2015 and continued until the end of August 2015. The measurement station is located approximately 2 km from Station Nord, and the instrumentation was set up at the station location. We also conducted measurements at the atmospheric observatory, Gruvebadet, located 2 km southeast of Ny-Ålesund, Svalbard (78° 55' N, 11° 56' E), from 28 March to 30 May 2017 (spring). Detailed information on the Greenland and Ny-Ålesund sampling site can be found in ref. [44]. The other field study was the MOSAiC expedition, which involved RV Polarstern drifting across the central Arctic from September 2019 to October 2020. The MOSAiC expedition track and detailed campaign information can be found in ref. [59].

### Detection of $HClO_3$ and $HClO_4$

A state-of-the-art chemical ionization atmospheric pressure interface time-of-flight mass spectrometry (CI-APi-TOF) instrument[60] was employed in negative ion mode with nitrate ($NO_3^-$) ions as reagent ions to detect gas-phase $HClO_3$ and $HClO_4$. In Greenland, a straight, stainless steel inlet tube with a length of 1 m and an outer diameter of ¾ inch was applied at ~1.5 m above ground level (a.g.l.) to sample ambient air with a flow rate of 10 liters per minute (lpm). The inlet tube was heated to zero degrees Celsius. At Ny-Ålesund, the inlet tube length was 2 m (outer diameter of ¾ inch) and the sample was taken through the roof (height = 2 m a.g.l.), with a flow rate of 10 lpm. On RV Polarstern, the nitrate CI-APi-TOF instrument was set up in a *Swiss* Container on the bow deck of vessel[61]. The nitrate CI-APi-TOF inlet was connected to a new particle formation (NPF) inlet, through a core sampling flange system, accommodating a neutral air ion spectrometer utilized to create ~60 lpm inlet flow (height = about 15 m above sea level). The zero measurements were conducted occasionally with a high-efficiency particulate air (HEPA) filter for at least 40 min each measurement in both Greenland and MOSAiC, which cover different seasons during the measurement period.

$HClO_3$ was detected as $ClO_3^-$ (82.954 *m/z*), and its isotope peak was clearly observed at 84.951 *m/z*, with a $^{35}Cl$:$^{37}Cl$ ratio of approximately 3:1 (Supplementary Fig. S9). Regarding $HClO_4$ detection, the peak of $ClO_4^-$ (98.949 *m/z*) should be carefully identified, as it is difficult to distinguish it from the isotopic peak of deprotonated sulfuric acid, $HSO_4^-$ (i.e., $H^{34}SO_4^-$ = 98.956 *m/z*) when the $HSO_4^-$ signal is high. The *m/z* values of these two peaks are close to each other and may create interference with lower-mass resolution devices. Therefore, we considered the peak of $^{37}ClO_4^-$ (100.946 *m/z*) to estimate the $HClO_4$ signal in this study. We also detected the $Br^-$ signal from the peak at 78.919 *m/z* (together with $NO_3(HBr)^-$ peak at 141.915 *m/z*; Supplementary Fig. S10), which is most likely attributed to hydrobromic acid (HBr)[62]. The raw data was pre-averaged over 10 min and processed with the MATLAB tofTools package according to the procedures described in Jokinen et al.[60].

The detected $ClO_3^-$ and $ClO_4^-$ were the deprotonated products of $HClO_3$ and $HClO_4$, respectively. Quantum chemical calculations (section S4 in SI) indicated that the binding free energies of $NO_3^-$ with $HClO_3$ and $HClO_4$ are 24.0 and 36.9 kcal mol$^{-1}$, respectively (the binding free energy is simply $-1 \times$ formation free energy in Supplementary Fig. S11), and the deprotonation free energies are 7.1 and 24.8 kcal mol$^{-1}$, respectively, which suggests that both $HClO_3$ and $HClO_4$ are efficiently deprotonated into $ClO_3^-$ and $ClO_4^-$, respectively. The $HClO_3 \cdot NO_3^-$ cluster remains more stable against deprotonation than the $HClO_4 \cdot NO_3^-$ cluster, and some fraction of the former could occur, which is consistent with the detection of the $HClO_3 \cdot NO_3^-$ cluster (refer to Supplementary Fig. S9c, d), while the $HClO_4 \cdot NO_3^-$ cluster was not present in the spectrum. The isomers of $HClO_3$ and $HClO_4$ detected via nitrate CI-APi-TOF likely occurred in the form of $HOClO_2$ and $HOClO_3$, respectively, as these components are most energetically stable in the atmosphere[42,63,64].

Currently, there are no available methods for $HClO_3$ and $HClO_4$ calibration of CI-APi-TOF instruments. More importantly, the handling of $HClO_3$ and $HClO_4$ is dangerous, as these substances are very corrosive and could cause violent explosions in reactions with organics, making calibration becoming very difficult. Based on quantum chemical calculations, the binding free energies of $NO_3^-$ with $HClO_3$ and $HClO_4$, i.e., $HClO_3 \cdot NO_3^- \rightarrow NO_3^- + HClO_3$ and $HClO_4 \cdot NO_3^- \rightarrow NO_3^- + HClO_4$ (24.0 and 36.9 kcal mol$^{-1}$, respectively), are similar to (the former is slightly lower than) that of $NO_3^-$ with $H_2SO_4$ ($H_2SO_4 \cdot NO_3^- \rightarrow NO_3^- + H_2SO_4$; 34.4 kcal mol$^{-1}$). However, similar to $H_2SO_4$ ($H_2SO_4 \cdot NO_3^- \rightarrow HSO_4^- + HNO_3$ = 20.3 kcal mol$^{-1}$), their binding free energies are higher than their deprotonation free energies (7.1 and 24.8 kcal mol$^{-1}$ for $HClO_3 \cdot NO_3^- \rightarrow ClO_3^- + HNO_3$ and $HClO_4 \cdot NO_3^- \rightarrow ClO_4^- + HNO_3$, respectively), which suggests that deprotonation of the formed clusters is efficient. Therefore, if $HClO_3$ and $HClO_4$ are dissociated during ionization, they could preferably form $ClO_3^-$ and $ClO_4^-$, respectively, and are detectable via nitrate CI-APi-TOF. This indicates that the detection of $HClO_3$ and $HClO_4$ is very likely as efficient as the detection of $H_2SO_4$ by $NO_3^-$, whose reaction rate is expected to occur within the kinetic limit range[65]. Thus, it is reasonable to assume that the instrument sensitivity to $HClO_3$ and $HClO_4$ is similar to the sensitivity determined for the $H_2SO_4$ measurement.

$H_2SO_4$ calibration for the nitrate CI-APi-TOF was conducted before or immediately after the campaign with a method presented in ref. [66]. Regarding the Greenland and Ny-Ålesund measurements, the obtained $H_2SO_4$ calibration factor was $1.48 \times 10^9$ (ref. [44]). During the MOSAiC campaign, calibration was completed twice after the campaign, and the average of two-factor values was $6 \times 10^9$. This factor includes the losses at the NPF inlet. Based on these calibration factors, the detection limits of $HClO_3$ and $HClO_4$ were estimated as $3 \times 10^3$ and $7 \times 10^3$ molecules cm$^{-3}$ (10 min-average, $3\sigma$), respectively, during the Greenland measurement campaign and $2 \times 10^4$ and $3 \times 10^4$ molecules

cm$^{-3}$ (10 min-average, $3\sigma$), respectively, during the MOSAiC measurement campaign. The detection limit for HClO$_3$ measurements in Ny-Ålesund was calculated to be $1\times10^3$ molecules cm$^{-3}$ (10 min-average, $3\sigma$).

The HClO$_3$ and HClO$_4$ concentrations in this study were computed with Eq. (13).

$$\left[\text{HClO}_x\right] = C \times \frac{\text{ClO}_x^- + (\text{HClO}_x)\bullet\text{NO}_3^-}{\text{NO}_3^- + (\text{HNO}_3)\bullet\text{NO}_3^-} \qquad (13)$$

where x equals 3 or 4, and $C$ is the calibration factor, which was assumed to be similar to the H$_2$SO$_4$ calibration factor. If the detected HClO$_3$ and HClO$_4$ clusters are not charged as efficiently as H$_2$SO$_4$, it could lead to underestimating the concentration of HClO$_3$ and HClO$_4$. The sum uncertainties of the HClO$_3$ and HClO$_4$ measurement from the collision limit of the target compound with its charger ions and potential inlet losses were predicted to be at least a factor of two.

### Ancillary measurements

O$_3$ was measured with a UV ozone analyser during both the Greenland and MOSAiC campaigns. More details on the O$_3$ measurement setup in Greenland can be found in ref. [67]. Regarding the MOSAiC O$_3$ measurements, we used here an hourly merged dataset that combines cross-evaluated measurements performed by three independent instruments, as has been detailed in refs. [68,69]. The particle number size distribution, 10–500 nm (9–915 nm in Greenland), was measured with a scanning mobility particle sizer (SMPS). The information on the SMPS setup in Greenland can be found in ref. [70]. As for the MOSAiC campaign, the SMPS was measured from the United States Department of Energy Atmospheric Radiation Measurement (ARM) Aerosol Observation System container [71,72]. Water vapor was measured during the MOSAiC campaign via cavity ring-down spectroscopy (CRDS) with a commercial Picarro instrument (model G2401) also connected to the interstitial inlet of the Swiss container. Meteorological parameters (temperature and solar radiation) during the MOSAiC were obtained from the meteorological observatory Polarstern [73]. Details on the atmospheric and meteorological equipment during the MOSAiC campaign can be found in ref. [59] and ref. [74].

### HClO$_3$ and HClO$_4$ photolysis rates

To obtain the photolysis rates of HClO$_3$ and HClO$_4$, we applied the estimated absorption cross-sections of these two compounds (refer to section S5 in SI for the cross-sections computation) in an explicit Tropospheric Halogen Chemistry Model (THAMO): a one-dimensional atmospheric chemistry model [75] that has been used in many previous studies (e.g., ref. [56]) to simulate the halogen chemical processes (including the photolysis) in the boundary layer. The THAMO simulations were conducted for 24 h in an Arctic environment (with a latitude of 81° 21′ N, similar to the location of Villum Research Station, Greenland) during spring (1 May) to derive the photolysis rates of HClO$_3$ and HClO$_4$ in the Arctic boundary layer.

## Data availability

The data that support the findings of this study are available online in the following repositories. All of the data obtained from Greenland (Villum Research Station) and Ny-Ålesund observations, and absorption cross-sections and photolysis rates of HClO$_3$ and HClO$_4$ have been deposited in Zenodo: https://doi.org/10.5281/zenodo.7655981 and https://doi.org/10.5281/zenodo.4292239 (for H$_2$SO$_4$).

The MOSAiC dataset used in this study are available at https://doi.org/10.1594/PANGAEA.944393 (for merged O$_3$), https://doi.pangaea.de/10.1594/PANGAEA.956085 (for HClO$_3$ and HClO$_4$), https://doi.pangaea.de/10.1594/PANGAEA.956087 (for HBr and H$_2$SO$_4$), https://doi.org/10.5439/1225453 (Kuang et al.[72] for the particle number size distribution), https://doi.org/10.1594/PANGAEA.935265 (for meteorological data), https://doi.org/10.1594/PANGAEA.954232 (for water vapor), and https://doi.org/10.17632/bn7ytz4mfz.1 (for BrO).

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

## Acknowledgements

This study received funding from the National Natural Science Foundation of China (42175118), European Research Council Executive Agency under the European Union's Horizon 2020 Research and Innovation Program (Project ERC-2016-COG726349 CLIMAHAL, ERC-StG GASPARCON - grant agreement no. 714621, the EMME-CARE project – grant agreement no. 856612, and grant no. 101002728, ATM-GTP Contract No. 742206), Climate Relevant interactions and feedbacks: the key role of sea ice and Snow in the polar and global climate system (CRiceS, grant number 101003826) and Horizon Europe project Non-$CO_2$ Forcers and their Climate, Weather, Air Quality and Health Impacts, (FOCI, project number 101056783), ACCC Flagship funded by the Academy of Finland grant no. 337549, Academy professorship funded by the Academy of Finland (grant no. 302958), Academy of Finland (project nos. 296628, 328290, 311932, 316114, 325647, 325681, 334792, 337549, 329274, 333397, 328616, 1325656, 347782, 349659, and 334514), "Quantifying carbon sink, CarbonSink+ and their interaction with air quality" INAR project funded by Jane and Aatos Erkko Foundation, University of Helsinki via ACTRIS-HY, SIOS funding (Project no: 2020_003), United States National Science Foundation (grant nos. 1807496 and 1807163), Swiss National Science Foundation (grant 200021_188478), and the Swiss Polar Institute (grant no. DIRCR-2018-004). J.S. holds the Ingvar Kamprad Chair for Extreme Environments Research, sponsored by Ferring Pharmaceuticals. J.C.-G. acknowledges "La Caixa" Foundation (ID 100010434) for the fellowship LCF/BQ/DR20/11790027. D.R.-S. is thankful to the Spanish "Ministerio de Ciencia e Innovación (MICINN)" for funding (Project ref. CTQ2017-87054-C2-2-P, Unit of Excellence María de Maeztu CEX2019-000919-M and "Ramón y Cajal" grant RYC-2015-19234. We acknowledge that the data reported in this manuscript were produced as part of the International Multidisciplinary drifting Observatory for the Study of Arctic Climate (MOSAiC) expedition with the MOSAiC20192020 tag, with activities supported by *Polarstern*[76] expedition AWI_PS122_00. The authors gratefully acknowledge the National Oceanic and Atmospheric Administration (NOAA) Air Resources Laboratory (ARL) for the provision of the Hybrid Single-Particle Lagrangian Integrated Trajectory (HYSPLIT) transport and dispersion model and/or Real-time Environmental Applications and Display sYstem (READY) website (https://www.ready.noaa.gov) employed in this publication. The authors also acknowledge the contributions of Oskari Kausiala, Daniela Wimmer, Lauri Ahonen, and Jyri Mikkilä during the Greenland field measurement. We acknowledge Matthew Shupe for his contribution to the $O_3$ measurements during the MOSAiC campaign. SMPS data of the MOSAiC campaign were obtained from the Atmospheric Radiation Measurement (ARM) User Facility, a U. S. Department of Energy (DOE) Office of Science User Facility Managed by the Biological and Environmental Research Program. We thank all those who contributed to MOSAiC and made this endeavor possible (ref. [77]). We acknowledge Ilann Bourgeois, Jeff Peischl, Chelsea R. Thompson, and Thomas B. Ryerson for collecting and providing the ATom $NO_x$ data. This work represents a contribution to CSIC Thematic Interdisciplinary Platform PTI POLARCSIC.

## Author contributions

Y.J.T., T.J., and A.S.-L designed the research. N.S., H.A, L.L.J.Q, I.B., T.L., L.J.B., M.B., O.P., R.C.T., D.Ho., B.B., S.D.A., L.Ba., K.P., J.H., D.He. H.-W.J., H.J., A.S.M., A.M., H.S., M.Si., J.S., and T.J. performed the field measurements. Y.J.T., N.S., S.I., H.A., X.-C.H., Q.Z., and T.J. analysed the data. S.I. conducted the quantum chemical calculations of the binding energies. Q.L., J.C.-G., A.B.-S., and D.R.-S. computed the absorption spectra, cross-sections, and photolysis rates of $HClO_3$ and $HClO_4$. Y.J.T., N.S., S.I., M.K., J.S. F., J.S., T.J., and A.S.-L. wrote the paper with the contributions of all co-authors.

## Competing interests

The authors declare no competing interests.

## Additional information

Yee Jun Tham [1,2,3] ✉, Nina Sarnela [1], Siddharth Iyer [4], Qinyi Li [5,19], Hélène Angot [6,7], Lauriane L. J. Quéléver[1], Ivo Beck[6], Tiia Laurila[1], Lisa J. Beck [1], Matthew Boyer[1], Javier Carmona-García [8], Ana Borrego-Sánchez[9], Daniel Roca-Sanjuán [8], Otso Peräkylä [1], Roseline C. Thakur[1], Xu-Cheng He [1], Qiaozhi Zha[1], Dean Howard[10,11,12], Byron Blomquist[11,12], Stephen D. Archer [13], Ludovic Bariteau[11,12], Kevin Posman[13], Jacques Hueber[10,20], Detlev Helmig[10,21], Hans-Werner Jacobi [7], Heikki Junninen [14], Markku Kulmala [1], Anoop S. Mahajan [15], Andreas Massling [16], Henrik Skov [16], Mikko Sipilä[1], Joseph S. Francisco [17], Julia Schmale [6], Tuija Jokinen [1,18] ✉ & Alfonso Saiz-Lopez [5] ✉

[1]Institute for Atmospheric and Earth System Research/Physics, Faculty of Science, University of Helsinki, 00014 Helsinki, Finland. [2]School of Marine Sciences, Sun Yat-sen University, Zhuhai 519082, China. [3]Guangdong Provincial Key Laboratory of Marine Resources and Coastal Engineering, Zhuhai 519082, China. [4]Aerosol Physics Laboratory, Tampere University, Tampere FI-3720, Finland. [5]Department of Atmospheric Chemistry and Climate, Institute of Physical Chemistry Rocasolano, CSIC, Madrid 28006, Spain. [6]Extreme Environments Research Laboratory, École Polytechnique Fédérale de Lausanne, (EPFL) Valais Wallis, Sion, Switzerland. [7]Univ. Grenoble Alpes, CNRS, INRAE, IRD, Grenoble INP, IGE, 38000 Grenoble, France. [8]Institut de Ciència Molecular, Universitat de València, P.O. Box 22085 València 46071, Spain. [9]Instituto Andaluz de Ciencias de la Tierra, CSIC-University of Granada, Av. de las Palmeras 4, 18100 Armilla, Granada, Spain. [10]Institute of Arctic and Alpine Research, University of Colorado, Boulder, CO 80309, USA. [11]Cooperative Institute for Research in Environmental Science, University of Colorado, Boulder, CO 80309, USA. [12]Physical Sciences Laboratory, National Oceanic and Atmospheric Administration, Boulder, CO 80305, USA. [13]Bigelow Laboratory for Ocean Sciences, East Boothbay, Maine, USA. [14]Laboratory of Environmental Physics, Institute of Physics, University of Tartu, Tartu, Estonia. [15]Indian Institute of Tropical Meteorology, Ministry of Earth Sciences, Pune 411008, India. [16]Department of Environmental Science, iClimate, Aarhus University, Roskilde, Denmark. [17]Department of Earth and Environmental Sciences and Department of Chemistry, University of Pennsylvania, Philadelphia, Pennsylvania 19104, USA. [18]Climate and Atmosphere Research Centre (CARE-C), the Cyprus Institute, P.O. Box 27456 Nicosia CY-1645, Cyprus. [19]Present address: Department of Civil and Environmental Engineering, The Hong Kong Polytechnic University, Hong Kong, China. [20]Present address: JH Atmospheric Instrumentation Design, Boulder, CO, USA. [21]Present address: Boulder Atmosphere Innovation Research LLC, Boulder, CO, USA. ✉e-mail: thamyj@mail.sysu.edu.cn; t.jokinen@cyi.ac.cy; a.saiz@csic.es

