## [Peer Review File · Nature Communications]

Widespread detection of chlorine oxyacids in the Arctic atmosphereReviewer #1 (Remarks to the Author):

Tham et al. present unique chemical ionization mass spectrometry measurements that support the observation of HClO₃ and HClO₄ in the Arctic troposphere during springtime. Results are presented from northern Greenland, Svalbard, and during the MOSAiC expedition. These measurements are intriguing and the first for the atmospheric trace gases HClO₃ and HClO₄. As described below, my main comments focus on adding references for currently uncited background knowledge, addressing measurement uncertainties and issues with detection limit reporting, adding the bromide measurement and Svalbard measurements to the methods section, and addressing the role of NO_x in chlorine chemistry.

ClO reaction with BrO, ClO, and HO₂ are considered, with ClO notably reacting with HO₂ to produce HOCl. However, there is no mention of NO_x, or inclusion of NO_x in the calculations or reaction mechanism. Wang and Pratt (2017, JGR-Atmos) showed that the production of ClONO₂ (from the reaction of ClO with NO₂) is estimated to be greater than HOCl production at even background NO₂ levels (even <10 ppt). This should be considered in the reaction mechanism presented on page 2, in the calculations of ClO loss (S1 and Fig S1), and in Fig 3. For section S1 and the associated discussion in the main text, both ClO + NO and ClO + NO₂ should be considered. Custard et al. (2016, Environ. Sci. Technol.) showed that [ClO] is highly sensitive to [NO_x]. Presumably, during MOSAiC, there were also times of significant NO_x influence from ship pollution that impacted the local environment, making this an important discussion point for even that study.

Likewise, for section S2 and the loss rate of OClO (including Table S1), the reaction of OClO with NO should be considered.

In considering ClO loss, the reactions ClO + CH₃OO, ClO + CH₃COOO, and ClO + OH, previously considered by Thompson et al. (2015, Atmos. Chem. Phys.), are not included. In comparison to BrO, ClO, HO₂, NO, and NO₂, how important are these reactions to ClO loss (Fig S1)?

Figure 2 shows "bromide" CI-API-TOF signals, also mentioned on Line 175. This is not included in the Methods section and must be added. Currently, the reader has no knowledge of what this corresponds to, what the uncertainties and LODs are, etc.

The methods discusses the estimated HClO₃ and HClO₄ concentrations based on H₂SO₄ calibration. However, the authors oddly provide detection limits corresponding to 1 standard deviation (presumably during zeros? Please clarify.), as opposed to the well-accepted definition of the limit of detection as 3*standard deviation of the blank, which should be considered. The detection limit is particularly important for the HClO₄ signal, which appears to be below this level for most of the Greenland study in Figure 1, and this is not discussed/addressed in the manuscript. The authors also present these detection limits (on Line 330) with two significant figures, which is not appropriate, especially given that HClO₃ and HClO₄ themselves were not actually calibrated for, due to safety concerns. Also, the phrasing "determined" on Line 330 should be changed to "estimated", since the calibrations were not conducted on the molecules being presented. In addition, there is no discussion of estimated uncertainties, which should be considered, especially given the lack of calibration.

Lines 287-288: Provide more information about the "zero measurements". The HEPA filter removes particles, but with what efficiency does it remove HClO₃ and HClO₄? At what frequency were zero measurements conducted during each of the field campaigns?

Extended Data Fig. 5 presents HClO₃ and HClO₄ levels at <5 ppb, 5-30 ppb, and >30 ppb O₃, but only means and ranges are presented, with no statistical analysis. Lines 194-196 states "higher mean HClO₄ concentrations [were] observed at higher O₃ levels", but mean HClO₄ is actually highest at 5-30 ppb in Extended Data Fig. 5. Also, no statistical

test is provided to support the author's assertion. Further, measurement uncertainties, also not discussed, are not considered in this comparison. In addition, the authors elsewhere consider <20 ppb O₃ to be an ozone depletion event, so it would seem more appropriate to divide the data by <5 ppb, 5-20 ppb, 20-30 ppb, and >30 ppb.

The following sentences that provide background information (e.g. reaction mechanisms, vapour pressures, Henry's Law constants, and other material) are missing references that need to be added: Lines 57-63, 79-81, 129, 145-147, 218, 233, and 237.

The [Cl] stated on Line 218 (without a reference) and SI Line 76 are low compared to that reported by Custard et al (2016, Environ. Sci. Technol.), who derived concentrations from simultaneous measurements of Cl₂ and ClO.

Figure 3: I suggest adding references for the mechanism presented, as it is based primarily on previous work. As discussed above, NO and NO₂ should also be included. Also, I suggest replacing the generic "Ground-Surface" with "Snowpack", as this is specific to the Arctic springtime.

What is the correlation between 35ClO₃- and H³⁵ClO₃NO₃? Adding a scatter plot and correlation coefficient would further support the measurement of HClO₃ as ClO₃-.

Abstract, Paragraph on Lines 85-92, and Methods: Add mention of measurement of HClO₃ at Ny-Ålesund, as this is an important supporting measurement presented in Extended Data Fig. 1.

The paragraph on Lines 95-114 is critical to the manuscript. The flow of this paragraph could be improved, as I had to read it several times to understand it switching between measurements in Greenland, Svalbard, and during MOSAiC. This was also particularly confusing since Svalbard wasn't measured in the preceding paragraph on Lines 85-92. I suggest revising the paragraph to focus on clarity.

Line 108: The statement of "yearlong measurements" is misleading, as Extended Data Figure 2 only presents measurements from Jan.-Jun. Please rephrase the text.

Line 162: The statement "snowpacks/sea-ice emissions of Cl₂ and BrCl" is misleading, as the references cited do not suggest emission of Cl₂ or BrCl from sea ice because the sea ice surface is buffered and R8 requires an acidic surface. To my knowledge, Cl₂ and BrCl have only been measured to be emitted from snow. Please correct.

Lines 163-164: Reference #3 (Custard et al. 2016, Environ. Sci. Technol.) would be appropriate to add here for the reaction of Cl with O₃ to form ClO.

Line 167: Thompson et al. (2015, Atmos. Chem. Phys.) also measured ClO.

Line 335: Fix the typo in the equation.

Extended Data Figure 4a & Extended Data Figure 7a: Fix y axis typos.

Extended Data Fig 6: Add to the caption that these photolysis rates (presumably) correspond to May 1 at Villum (according to the Methods), as these rates would be significantly lower for early spring during MOSAiC, for example. It is important to be clear in the caption as to what is being presented.

SI Line 49 – Ref #7 didn't measure ClO. However, Custard et al (2016, Environ. Sci. Technol.) and Thompson et al. (2015, Atmos. Chem. Phys.) both did measure ClO, and these publications are not cited here.

Reviewer #2 (Remarks to the Author):

Dear Yee Jun Tham et al.

It is with great pleasure that I read your manuscript on providing measurement evidence for novel chlorine species that terminate atmospheric cycles. Clearly, the identification of this termination step is a result noteworthy to the atmospheric science, arctic science, and beyond. Taken the negative impact on ecosystems, that the authors note, I'm confident that this result will inspire other scientist in the wider environmental community.

The manuscript is well written, the methods of data acquisition appear sound to me, conclusions are fully supported by data and/or well-argued based on relevant literature. I'm particularly impressed by discussion data from several regions/field campaigns to draw your conclusions which nicely emphasizes the locally wide spread importance of the findings.

My only comment deals with the impact of surface processes as source of the oxyacids and later as sink process. If I understood correctly, measurements are reported from one specific height above the ground for each data set? Profiles or flux measurements were not done. What is unclear to me is what "air mass originated from the near ground surface" exactly means. I assume that those air masses are in contact with the ground? Would then not also any oxyacid formed in the air – as opposed to in the surface snow-partitioning to the snow and be lost from the atmosphere? In other words, the argument of strong partitioning to the snow should also hold for gas-phase species and not only those (originally) at the surface.

Generally, I'm not fully convinced by the argumentation about sticking to the surface. Would you not agree that a) there is some uncertainty due to pH and/or acid-base equilibria at the air-ice interface on the partitioning? And b) I miss a more discussion on equilibrium processes. To start general, what is not clear to me is if you refer to the vapor pressure and to KH as parameters that actually describe the partitioning to ground surface and to aerosol, or just as indicators as the true parameters are not known. For snow surfaces, for example, I would have expected the Langmuir partitioning constant. As for the KH which describes the partitioning between the gas and solution phase. Which concentration would you need in the liquid phase to maintain the measured gas-phase concentrations? Taken that humidity is low and temperature is low, aerosol and aerosol deposits in snow are often highly concentrated. Further, the absence of high concentrations in winter and in summer could also indicate that snow and sun is needed and not necessarily that surface snow plays no role. In summary, this argumentation would benefit from a few clarifying numbers – that I might have missed elsewhere in the text. If so, my apologies. This includes the sampling height, a statement about transport limitations to the surface from that height/to that height, bulk concentrations needed to maintain the measured gas-phase concentrations. This comment refers to lines 137-155 and to some extent to 232-238

Overall, I recommend publication after tackling these minor suggestion of mine.

Detailed comments

LI 79-81: This sounds a little awkward. The oxyacids might have always been there, without detection. How then, can detecting them, perturb the chemistry? It can "perturb" our understanding of it.

LI 125: "in the arctic troposphere". This is -so far- limited to your measurement height, is it not? This is not the whole troposphere.

LI153: For which data sets was that pattern investigated? Spring only, or also summer and winter? I think this needs to be limited to spring. Please specify in the text.

Reviewer #3 (Remarks to the Author):

Review of paper "Widespread detection of chlorine oxyacids in the Arctic Atmosphere" by Tham et al.

This is an interesting paper that documents the detection of HClO₃ and HClO₄ in the Arctic boundary layer. I was kind of excited to see this finding. The authors have attempted to quantify their detected amounts (which are pretty small) and then tried connecting them to the boundary layer episodic ozone depletion events. At first blush, it all looks OK.

However, I have some significant reservations. It is possible that the authors can answer these concerns. Then, the question becomes, is the detection of these chemicals more a testament to the modern analytical capabilities, or does it have any significance. I will opine on the latter point at the end.

Major concerns:

- 1. Perchlorates and chlorates are ubiquitous in the aqueous systems. There is vast literature on it. These are often due to the widespread use of perchlorates for various purposes such as rocketry, flares, matches, etc. The aqueous concentrations of perchlorates have been documented. So, my first concern is: How do the authors know that the detected HClO₃ or HClO₄ are due to chlorine chemistry in the atmosphere and not simply reintroducing these chemicals from the aqueous system? It is important to note that the very processes that lead to the release of halogens in the Arctic could also lead to the gas phase release of perchloric and chloric acids. This process is equal to having up arrows from the ground/surface in Figure 3. (Please note that acidification of the droplets could conceivably release these acids to the gas phase....)**
- 2. The authors could not calibrate their system for detecting the two acids. So, they have gone through an indirect method to convert measured mass spec signals into concentrations. First, I am not sure they could not have devised a way to calibrate their system. Second, I do not see any discussion of the uncertainties in their quoted concentrations. Third, the methodology section did not convince me that converting the mass spec signal to concentration is devoid of problems.**
- 3. Isotope geochemists have been looking at perchlorates for signals of stratospheric influence and other sources. It is not clear to me that the authors have considered this issue. I do not see any ³⁶Cl signals (Did they not see it?) suggesting a lack of cosmogenic chlorine. I acknowledge that the signals may be too small.... But, the authors have not said anything about it. So I wonder if they have considered this issue.**
- 4. Can the isotope signals say anything about the origin of chlorine? It is possible that their detection limits (without preconcentration) are not sufficient to see it, but nothing was mentioned. Furthermore, they see ³⁵Cl/³⁷Cl ratios that are not really 3.1. (Incidentally, there are no error bars on any of these- I will comment on this below). It is also known that marine biology does a certain amount of fractionation. Are there any signals here? Lastly, did the authors detect any Cl₂ and ClBr? If yes, what were the isotope ratios?**
- 5. I am concerned about a lack of uncertainty treatment here. I have no idea how to interpret the data when there are no estimates of the uncertainties and no error bars on the data.**
- 6. Maybe I am missing it, but I did not see any data on the aerosol levels. Were they made? If yes, what do they look like? What was their composition? (I am sure that the authors are aware the entire finding about the halogen-induced episodic ozone depletion in the Arctic boundary layer was evoked by measurements of aerosol composition....)**
- 7. There is a lot of sloppy writing. Here are a few examples: (a) this is a detection of HClO₃ and HClO₄ in the Arctic boundary layer, not all across the Arctic atmosphere. It definitely is not in the free troposphere or the stratosphere; (b) the role of chlorine in determining CH₄ lifetime is minimal. Whatever it does in the Arctic boundary layer does not change CH₄ lifetime to any appreciable extent; (c) hydroxyl radical is not an oxidant for everything in the atmosphere. (I know this has become the standard phraseology,**

but it is simply not true!); (d) Are they really concerned about the atmospheric toxicity of HClO₃ and HClO₄ from this source compared to the anthropogenic source?; and (e) rates and rate coefficients are mixed up.

Minor comments:

- 1. Why are they wasting space and time showing the first-order rate coefficients (I assume these are J values in Figures 6b and 6c)? If there is no absorption in the near UV, nobody would expect a photolytic loss. Again, the values shown in Table S2 are J values and not rates. (rates have units of molecules per cm³ per sec!).**
- 2. Figure 8 shows the signal to noise for the detection. Are the authors using peak heights or peak areas to quantify the concentrations?**
- 3. Use of colors in figures: Yellow, pale green, pale blue, pale pink, etc., may look "pretty" for some, but they are hard to see (especially think of people who are color blind to various extents!).**

I have not commented on the modeling and atmospheric analyses since I am not convinced where these chemicals are coming from. If they convince themselves (and us!) that, indeed, it is coming from the chemistry involved in

Bottom line: I am convinced they have detected HClO₃ and HClO₄ in the gas phase. However, is that enough to publish it in this journal? Currently, I do not think that they are. However, the authors can dissuade me by addressing the points noted above. Then the question become what is the significance of these acids since HCl would still be a major path for the removal of chlorine from these atmospheres.

We thank the three reviewers for the detailed comments and constructive suggestions. Our point-by-point responses are listed in dark blue wording and the corresponding changes made to the revised manuscript are in light blue. The changes in the revised manuscript (track-change version) have been highlighted in light blue.

Reviewer #1 (Remarks to the Author):

(1.1) Tham et al. present unique chemical ionization mass spectrometry measurements that support the observation of HClO₃ and HClO₄ in the Arctic troposphere during springtime. Results are presented from northern Greenland, Svalbard, and during the MOSAiC expedition. These measurements are intriguing and the first for the atmospheric trace gases HClO₃ and HClO₄. As described below, my main comments focus on adding references for currently uncited background knowledge, addressing measurement uncertainties and issues with detection limit reporting, adding the bromide measurement and Svalbard measurements to the methods section, and addressing the role of NO_x in chlorine chemistry.

Reply: We thank the reviewer for her/his detailed and insightful comments which have improved the manuscript and have now been fully incorporated into the revised manuscript.

(1.2) ClO reaction with BrO, ClO, and HO₂ are considered, with ClO notably reacting with HO₂ to produce HOCl. However, there is no mention of NO_x, or inclusion of NO_x in the calculations or reaction mechanism. Wang and Pratt (2017, JGR-Atmos) showed that the production of ClONO₂ (from the reaction of ClO with NO₂) is estimated to be greater than HOCl production at even background NO₂ levels (even <10 ppt). This should be considered in the reaction mechanism presented on page 2, in the calculations of ClO loss (S1 and Fig S1), and in Fig 3. For section S1 and the associated discussion in the main text, both ClO + NO and ClO + NO₂ should be considered. Custard et al. (2016, Environ. Sci. Technol.) showed that [ClO] is highly sensitive to [NO_x]. Presumably, during MOSAiC, there were also times of significant NO_x influence from ship pollution that impacted the local environment, making this an important discussion point for even that study.

Reply: Thanks for this suggestion. We agree with the reviewer that NO_x may influence the reactions of ClO and OClO, particularly for the MOSAiC campaign. As noted in Beck *et al.* (2022) study, the measurements of the MOSAiC campaign can be influenced by the local ship pollution (i.e., aerosol pollution) from time to time, and they provided an example case of the influence of ship exhaust plume with measured NO level up to 15 ppb.

Unfortunately, due to the quality control challenges, the NO_x data set of the MOSAiC campaign are not available for use, especially in the springtime. The lacking of data prevents us from evaluating the actual influence of NO_x on the HClO₃ and HClO₄ formation. If we assume that the NO concentration in plumes with influence of ship exhaust is 15 ppb (Beck *et al.*, 2022), we can expect that the ClO (i.e., 40 ppt) and OClO (i.e., 24 ppt) will be depleted completely within 1 second or within 10 seconds when the NO is 1 ppb (Fig. R1.1), which means, at these NO levels the subsequent oxidation processes leading to gas-phase production of HClO₃ and HClO₄ will not take place (due to the lack of precursors, i.e., ClO and OClO). However, during the period with influence of aerosol pollution from the ship (shaded background in Fig. R1.2), where the NO_x level is presumed to be elevated too, the observed significant HClO₃ and HClO₄ concentrations indicate that the NO_x emission during MOSAiC probably will not completely shut down the gas-phase production of HClO₃ and HClO₄ in the region.

To include the potential influence of NO_x, we used the range of NO_x concentrations measured by chemiluminescence instrument (see Bourgeois *et al.*, 2022 for a description of the instrument) during the Atmospheric Tomography (ATom) aircraft campaign, collected in the Arctic (70–90°N) marine boundary layer (altitude <1 km) (see Ryerson *et al.*, 2019 for the details of the campaign). The concentrations of NO and NO₂ were observed to be up to 15 ppt and up to 40 ppt, respectively (Fig. R1.3), and these ranges of NO_x concentrations are used as the reference for estimating the loss rates for ClO and OClO.

A calculation test with 0.015 ppb of NO shows that the ClO and OCIO concentrations can remain significant after 30 s of reaction (Fig. 1.1), supporting our hypothesis that the gas-phase production of HClO₃ and HClO₄ can still occur in the presence of background levels of NO_x.

Following the referee's suggestion, we have added the NO_x to the calculations and reaction mechanism in the revised manuscript. The details of the calculations, and revisions on the reaction mechanism and discussions are discussed in the replies to comments (1.3) and (1.4) below.

Fig. R1.1 | Loss of ClO (a) and OCIO (b) by reaction with NO (15 ppb, 1 ppb and 0.015 ppb). The initial concentrations (C_0) for ClO and OCIO are 40 ppt and 24 ppt (e.g., Custard et al., 2016; Pöhler et al., 2010), respectively; the temperature of the reaction is 253 K; and the time step used in the calculation is 0.01 s. The calculated NO concentration remains significant after 30 s of reaction.

Fig.R1.2 | Expanded view of HClO₃ and HClO₄ levels during the measurement period with potential influence of ship pollution in the MOSAiC campaign. The grey shaded area (with a time resolution of

1 min) denotes the period where the particle number concentration measurements were impacted by the ship pollution, as defined in Beck *et al.* (2022). This figure is indicative only and does not necessarily reflect the percentage of clean data points collected by other instruments during the expedition. There is, however, a relatively high probability that NO_x levels were elevated during these periods.

Fig. R1.3 | The NO_x levels in the Arctic. The data collected in the Arctic (70–90°N) in the marine boundary layer (altitude <1 km) during the ATom aircraft campaign in February 2017, May 2018, August 2016, and October 2018 (Ryerson *et al.*, 2019).

References:

- Beck, I. *et al.* Automated identification of local contamination in remote atmospheric composition time series. *Atmos. Meas. Tech.* **15**, 4195-4224 (2022).
- Bourgeois, I. *et al.* Comparison of airborne measurements of NO, NO₂, HONO, NO_y, and CO during FIREX-AQ. *Atmos. Meas. Tech.* **15**, 4901-4930 (2022).
- Ryerson, T. B., Thompson, C. R., Peischl, J. & Bourgeois, I. ATom: L2 In Situ Measurements from NOAA Nitrogen Oxides and Ozone (NO_yO₃) Instrument. *ORNL Distributed Active Archive Center*, (2019).
- Custard, K. D., Pratt, K. A., Wang, S. & Shepson, P. B. Constraints on Arctic atmospheric chlorine production through measurements and simulations of Cl₂ and ClO. *Environ. Sci. Technol.* **50**, 12394-12400 (2016).
- Pöhler, D., Vogel, L., Frieß, U. & Platt, U. Observation of halogen species in the Amundsen Gulf, Arctic, by active long-path differential optical absorption spectroscopy. *Proc. Natl. Acad. Sci. U.S.A* **107**, 6582 (2010).

(1.3) Likewise, for section S2 and the loss rate of OCIO (including Table S1), the reaction of OCIO with NO should be considered.

Reply: We have calculated the reaction of OCIO with NO. The NO concentration assumed in the calculation is according to the measurements of the ATom campaign (Ryerson *et al.*, 2019), where the mean concentration in May 2018 was 1.5×10^8 molecules cm⁻³ (5 ppt) and the maximum concentration was 4.4×10^8 molecules cm⁻³ (15 ppt) (see Fig R1.3 in reply to comment (2) above). As shown in the revised Table S1, when [NO] is 4.4×10^8 molecules cm⁻³, the first order loss rate of OCIO ($k_{\text{OCIO}+\text{NO}}$) is 6.6×10^{-5} s⁻¹, while the $k_{\text{OCIO}+\text{NO}}$ is 2.2×10^{-5} s⁻¹ when the [NO] is 1.5×10^8 molecules cm⁻³. The ratios of $k_{\text{OCIO}+\text{OH}}/k_{\text{OCIO}+\text{NO}}$ are in the range of 6×10^{-1} – 3×10^1 . Note that the reaction of OCIO with NO will recycle ClO, which can form OCIO again for further reactions. Therefore, among the reactions of OCIO with OH, Cl, O₃, and NO, the reaction of OCIO + OH represents a significant fraction of OCIO loss (up to several orders of magnitude higher than other pathways), ultimately promoting the HClO₃ formation. We have revised the main text and added relevant information to the revised main text:

(Revised main manuscript, Line 199-205): OCIO can undergo further reactions, including (1) reacting with Cl to yield two ClO molecules; (2) oxidation by OH to form HClO₃; (3) reacting with NO to recycle ClO; and (4) oxidation by O₃ to produce ClO₃ (Fig. 3). Among these reactions, OCIO + OH exhibits the fastest rate, with $k_{\text{OCIO+OH}}[\text{OH}]/k_{\text{OCIO+Cl}}[\text{Cl}]$ and $k_{\text{OCIO+OH}}[\text{OH}]/k_{\text{OCIO+O}_3}[\text{O}_3]$ ratios calculated to be in the range of 2×10^{-1} – 1×10^4 and 1×10^3 – 1×10^6 , respectively, while the $k_{\text{OCIO+OH}}[\text{OH}]/k_{\text{OCIO+NO}}[\text{NO}]$ ratios fall in the range of 6×10^{-1} – 3×10^1 (section S3 in the SI). These results suggest that a significant fraction of OCIO can be directly oxidized by OH to convert into HClO₃, and produce ClO to recycle OCIO.

As in the revised supplementary information (SI), we have added the calculation of OCIO + NO to section S2 and revised Table S1 accordingly.

(Revised SI, Line 127-142):

Table S1 shows the estimated consumption rate of OCIO by OH, Cl, O₃ and NO. When OH concentration is about 0.1 – 1.0×10^6 molecules cm^{-3} , a typical range of OH during O₃ depletion as reported in previous Arctic measurements^{11,18,19}, the $k_{\text{OCIO+OH}}[\text{OH}]$ is $\approx 10^{-4}$ – 10^{-5} s^{-1} . Cl atom was reported to be in the range of $\approx 10^3$ – 10^6 molecules cm^{-3} in Arctic spring^{8,12,18,20}. The $k_{\text{OCIO+Cl}}[\text{Cl}]$ is calculated to be $\approx 6 \times 10^{-5}$ – 10^{-8} s^{-1} . From Fig.1, the typical O₃ concentration during an O₃ depletion event falls within the range of 3×10^8 – 6×10^{11} molecule cm^{-3} (1–20 parts per billion, ppb), the $k_{\text{OCIO+O}_3}[\text{O}_3]$ are estimated to be in between $\approx 1 \times 10^{-8}$ – $5 \times 10^{-8} \text{ s}^{-1}$. From the ATom aircraft measurements over the Arctic in May 2018¹⁶, the mean concentration was 1.5×10^8 molecules cm^{-3} (5 ppt) and the maximum concentration was 4.4×10^8 molecules cm^{-3} (15 ppt). When NO concentration is 4.4×10^8 molecules cm^{-3} , $k_{\text{OCIO+NO}}$ is $6.6 \times 10^{-5} \text{ s}^{-1}$, while the $k_{\text{OCIO+NO}}$ is $2.2 \times 10^{-5} \text{ s}^{-1}$ when the NO concentration is 1.5×10^8 molecules cm^{-3} . Note that the reaction of OCIO + NO will produce ClO, which can recycle OCIO via reaction RS1. Therefore, among these reactions, OCIO + OH represents a significant fraction of OCIO loss (up to several orders of magnitude higher than other pathways), ultimately promoting the HClO₃ formation.

Table S1 | Loss of OCIO. The calculated consumption rate of OCIO by OH, Cl, O₃ and NO. The reaction temperature is 253 K. Concentration is in molecules cm^{-3} .

Reaction	OCIO + OH		OCIO + Cl		OCIO + O ₃		OCIO + NO	
	[OH]	[OH]	[Cl]	[Cl]	[O ₃]	[O ₃]	[NO]	[NO]
	= 5×10^6	= 1×10^5	= 1×10^6	= 1×10^3	= 6×10^{11}	= 3×10^{10}	= 4.4×10^8	= 1.5×10^8
$k_{\text{OCIO+X}}[\text{X}]$ (s^{-1})	6.6×10^{-4}	1.3×10^{-5}	6.3×10^{-5}	6.3×10^{-8}	1.0×10^{-8}	5.2×10^{-10}	6.6×10^{-5}	2.2×10^{-5}

note X = OH; Cl; O₃ or NO

(1.4) In considering ClO loss, the reactions ClO + CH₃OO, ClO + CH₃COOO, and ClO + OH, previously considered by Thompson *et al.* (2015, Atmos. Chem. Phys.), are not included. In comparison to BrO, ClO, HO₂, NO, and NO₂, how are important are these reactions to ClO loss (Fig S1)?

Reply: The losses of ClO with OH, CH₃OO and CH₃COOO are very likely not important compared to the losses of ClO with HO₂ and BrO for two reasons. First, according to the reaction rate constants used in Thompson *et al.* (2015), the ClO + CH₃OO ($2.08 \times 10^{-12} \text{ cm}^3 \text{ molecule}^{-1} \text{ s}^{-1}$) and ClO + CH₃COOO ($2.03 \times 10^{-12} \text{ cm}^3 \text{ molecule}^{-1} \text{ s}^{-1}$) are about 4 times slower than the ClO+HO₂ ($8.67 \times 10^{-12} \text{ cm}^3 \text{ molecule}^{-1} \text{ s}^{-1}$), while the reaction constant for ClO + OH ($2.37 \times 10^{-13} \text{ cm}^3 \text{ molecule}^{-1} \text{ s}^{-1}$) is an order of magnitude slower than the reaction constant for ClO + HO₂. Second, the concentrations of CH₃OO and CH₃COOO in the Arctic are likely smaller than the concentration of HO₂ because their lifetime may be short as these RO₂ radicals typically react more rapidly with NO or HO₂ (Thompson *et al.*, 2015). Even if we assume

that their concentration equals the HO₂ concentration (2.9×10⁸ molecules cm⁻³), the loss rates of ClO (at a concentration of 5.8×10⁸ molecules cm⁻³) by reacting with CH₃OO and CH₃COOO are 3.5×10⁵ and 3.4×10⁵ molecules cm⁻³ s⁻¹, respectively, which are much lower than the ClO + HO₂ pathway (refer to Fig. S1). The loss rate of ClO (at a concentration of 5.8×10⁸ molecules cm⁻³) by reacting with 5×10⁶ molecules cm⁻³ of OH is negligible (rate is 6.9×10² molecules cm⁻³ s⁻¹). We have acknowledged this information in the revised main text and in the supplementary information (see below).

As for the reaction of ClO with NO_x, we have added the calculation of ClO + NO and ClO + NO₂ using the NO_x concentration range obtained from the ATom campaign (see reply to comment 1.2). The calculation results show that the loss rates of ClO with NO and NO₂ are comparable to the reaction rate of ClO + BrO (Fig. S1b), meaning the NO_x will compete with BrO in the reaction of ClO to form OClO. To include the potential influence of NO and NO₂ on the loss of ClO, we have made several changes in the revised manuscript.

As suggested by the reviewer, we have included the NO_x into the chlorine reaction mechanism in the Introduction:

(Revised main manuscript, Line 71-88): Upon photolysis, these chlorine species release reactive Cl atoms (R1 and R2), which rapidly react with O₃ to form chlorine monoxide, ClO (R3). ClO is subsequently oxidized by bromine monoxide (BrO) or ClO, producing chlorine dioxide (OClO; R4); reacting with HO₂ to form hypochlorous acid (HOCl; R5); or reacting with nitric oxide (NO) and nitrogen dioxide (NO₂) to produce Cl atoms and chlorine nitrate (ClONO₂) (R6–R7). Cl atoms can also degrade other hydrocarbons (RH) to generate hydrochloric acid (HCl) (R8). HCl can be converted into chloride (Cl⁻) via hydrolysis on aerosol surfaces (R9). Chloride can further undergo heterogeneous reaction with HOCl to produce Cl₂ (R10), which can in turn be photolyzed to recycle Cl atoms (R1).

The influence of NO_x on ClO reactions has now been added to the discussion in the section of potential formation mechanism of atmospheric chlorine oxyacids:

(Revised main manuscript, Line 182-183): In addition to photolysis, the produced ClO can then react with BrO/ClO to produce OClO, or undergo loss through reactions with OH, HO₂, NO and NO₂, CH₃OO, and CH₃COOO².

(Revised main manuscript, Line 190-192): The reaction of ClO + OH, ClO + CH₃OO and ClO + CH₃COOO are insignificant, however, the presence of typical levels of NO_x (NO and NO₂) in the Arctic (i.e., 1–40 ppt) can compete with BrO for ClO (section S2 in SI).

Detail information for the calculation of ClO + NO and ClO + NO₂ have been added to section S2 and Fig. S1 in the revised supplementary information.

(Revised SI, Line 74-117):

In this calculation, we used the BrO, ClO and HO₂ concentrations that have been previously observed in the Arctic during spring. Typically, the BrO concentrations were reported to peak in the range of about 1.5–10.2 × 10⁸ molecules cm⁻³ (5–35 parts per trillion, ppt) during an Arctic ozone depletion event⁵⁻⁹ and MOSAiC campaign¹⁰. The ClO concentration was reported to be up to 1.1 × 10⁹ molecules cm⁻³ (up to 40 ppt) in previous field observations in Arctic springtime^{8,11,12}. The HO₂ concentration was typically measured to peak at approximately 1.5–2.9 × 10⁸ molecules cm⁻³ (5–10 ppt) in the springtime O₃ depletion events^{13,14}. NO_x data were obtained from the Atmospheric Tomography (ATom) aircraft measurements over the Arctic (70–90°N) marine boundary layer (altitude <1 km), using the NOAA chemiluminescence instrument described in Bourgeois *et al.*¹⁵. The NO and NO₂ concentrations ranging from 0.3–4.4 × 10⁸ molecules cm⁻³ (1–15 ppt) and 0.6–11.6 × 10⁸ molecules cm⁻³ (2–40 ppt), respectively¹⁶.

Fig. S1 shows the range of ClO loss rates calculated with the above concentration ranges of BrO, ClO, HO₂, NO and NO₂. The ClO + BrO reaction channel is the dominant (rates up to 1.0 × 10⁷ molecules cm⁻³ s⁻¹), followed by the ClO + HO₂ (rates up to 2.9 × 10⁶ molecules cm⁻³ s⁻¹). The reaction of ClO + ClO is negligible as the rate (up to 2.1 × 10³ molecules cm⁻³ s⁻¹) is several orders of magnitude lower than the former two reaction channels. Although the *k*_{RS1} and *k*_{RS3} are quite comparable, the BrO concentration during an Arctic ozone depletion event is usually several times higher than HO₂ concentrations^{11,14}, suggesting that the ClO + BrO channel can lead to the enhancement of OCIO, and contribute to the formation of HClO₃ and HClO₄. Furthermore, the fraction of ClO reacting with HO₂ to form HOCl can recycle to produce Cl atoms via direct photolysis or heterogeneous uptake on chloride aerosol, which can eventually form ClO. The calculated reaction rates of ClO + NO (up to 1.0 × 10⁷ molecules cm⁻³ s⁻¹) and ClO + NO₂ (up to 9.6 × 10⁶ molecules cm⁻³ s⁻¹) are comparable to the reaction rate of ClO + BrO (Fig. S1b), meaning that the typical levels of NO_x will compete with BrO in the reaction of ClO to form OCIO.

According to the Thompson *et al.*¹¹, the ClO can also react with radicals such as OH, CH₃OO, and CH₃COOO. However, the reactions of ClO with OH, CH₃OO and CH₃COOO are very likely not important compared to the losses of ClO through HO₂ and BrO for two reasons. First, according to the reaction rate constants used in Thompson *et al.*¹¹, the ClO + CH₃OO (2.08 × 10⁻¹² cm³ molecule⁻¹ s⁻¹) and ClO + CH₃COOO (2.03 × 10⁻¹² cm³ molecule⁻¹ s⁻¹) are about 4 times slower than the ClO + HO₂ (8.67 × 10⁻¹² cm³ molecule⁻¹ s⁻¹), while the reaction constant for ClO + OH (2.37 × 10⁻¹³ cm³ molecule⁻¹ s⁻¹) is an order of magnitude slower than the reaction constant for ClO + HO₂. Second, the concentrations of CH₃OO and CH₃COOO in the Arctic are likely smaller than concentration of HO₂ because their lifetime may be short as these RO₂ radicals typically react more rapidly with NO or HO₂¹¹. Even if we assume that their concentration equals to the HO₂ concentration (i.e. 2.9 × 10⁸ molecules cm⁻³), the loss rates of ClO (at a concentration of 5.8 × 10⁸ molecules cm⁻³) by reacting with CH₃OO and CH₃COOO are 3.5 × 10⁵ and 3.4 × 10⁵ molecules cm⁻³ s⁻¹, respectively, which are much lower than the ClO + HO₂ pathway (refer to Fig. S1). The loss rate of ClO (at a concentration of 5.8 × 10⁸ molecules cm⁻³) by reacting with 5 × 10⁶ molecules cm⁻³ of OH is negligible (rate is 6.9 × 10² molecules cm⁻³ s⁻¹).

Fig.S1 | Loss rates of ClO. The calculated rates of (a) ClO consumed by BrO (blue dash line), HO₂ (pink) and ClO (green). (b) Comparison of the rates of ClO + BrO with rates of ClO + NO and ClO+NO₂. Note that the shaded area represents the range of loss rate calculated from different levels of BrO (5–35 ppt), HO₂ (5–10 ppt), and ClO (5–40 ppt) reported in the spring ozone depletion events in the Arctic^{5-9,11-14}. The concentrations of NO (1–15 ppt) and NO₂ (2–40 ppt) were adopted from the ATom flight measurements in the Arctic¹⁶.

References:

- Thompson, C. R. *et al.* Interactions of bromine, chlorine, and iodine photochemistry during ozone depletions in Barrow, Alaska. *Atmos. Chem. Phys.* **15**, 9651-9679 (2015).

(1.5) Figure 2 shows “bromide” Cl-API-TOF signals, also mentioned on Line 175. This is not included in the Methods section and must be added. Currently, the reader has no knowledge of what this corresponds to, what the uncertainties and LODs are, etc.

Reply: Thank you for the suggestion. The Br⁻ signal in the nitrate-based Cl-API-TOF is most likely attributed to the deprotonation of HBr by dominant reagent ions NO₃⁻, NO₂⁻ and O₂⁻ (Fan *et al.*, 2021).

However, without proper information on the calibration factor, zeroing measurement, etc., we are not able to determine its uncertainty, LODs, etc. Our main purpose of providing the normalized Br⁻ signal in this study is to support the increase of 'bromine chemistry' during the Arctic springtime ozone depletion, which is already a well-known phenomenon, widely reported in the literature (e.g., Barrie and Platt, 1997; Platt & Wagner, 1998; Wang *et al.*, 2019). Relevant information on the Br⁻ measurement and data treatment have been added to the revised Methods section.

(Revised main manuscript, Line 329-330): We also detected the Br⁻ signal from the peak at 78.919 m/z, which is most likely attributed to hydrobromic acid (HBr)⁶¹.

References:

- Fan, X. *et al.* Atmospheric gaseous hydrochloric and hydrobromic acid in urban Beijing, China: detection, source identification and potential atmospheric impacts. *Atmos. Chem. Phys.* **21**, 11437-11452, (2021).
- Barrie, L. & Platt, U. Arctic tropospheric chemistry: An overview. *J Tellus B: Chemical Physical Meteorology* **49**, 450-454 (1997).
- Platt, U. & Wagner, T. Satellite mapping of enhanced BrO concentrations in the troposphere. *Nature* **395**, 486-490 (1998).
- Wang, S. *et al.* Direct detection of atmospheric atomic bromine leading to mercury and ozone depletion. *Proc. Natl. Acad. Sci. U.S.A* **116**, 14479 (2019).

(1.6) The methods discuss the estimated HClO₃ and HClO₄ concentrations based on H₂SO₄ calibration. However, the authors oddly provide detection limits corresponding to 1 standard deviation (presumably during zeros? Please clarify.), as opposed to the well-accepted definition of the limit of detection as 3*standard deviation of the blank, which should be considered. The detection limit is particularly important for the HClO₄ signal, which appears to be below this level for most of the Greenland study in Figure 1, and this is not discussed/addressed in the manuscript. The authors also present these detection limits (on Line 330) with two significant figures, which is not appropriate, especially given that HClO₃ and HClO₄ themselves were not actually calibrated for, due to safety concerns. Also, the phrasing "determined" on Line 330 should be changed to "estimated", since the calibrations were not conducted on the molecules being presented. In addition, there is no discussion of estimated uncertainties, which should be considered, especially given the lack of calibration.

Reply: We apologize for causing the confusion here. Yes, the lower detection limits (LOD) were calculated from the background data (10 min integration) obtained during the zeroing measurements with a HEPA filter. We have recalculated the LODs from the background signals and found that the present values are 3 times of standard deviation (3σ) of the background. The typo has been fixed in the revised main manuscript (Line 368 and 370).

We acknowledge that without actual calibration, it is not appropriate to report two significant figures in the LOD values. We have therefore revised the LOD values to one significant number as suggested by the reviewer. The word on Line 330 (original manuscript) has been changed to "estimated" in the revised main manuscript (Line 368).

The major uncertainties of the measurement with a nitrate CI-API-TOF include the collision limit of the target compound with its charger ions (calibration) and potential inlet losses. Previous measurements of H₂SO₄ and HIO₃ with a nitrate CI-API-TOF estimated an uncertainty of +100/-50% or a factor of two (Sipilä *et al.*, 2016; Beck *et al.*, 2021). Since our quantum chemical calculation results suggest that the detection of HClO₃ and HClO₄ is very likely as efficient (at collision limit) as the detection of H₂SO₄. The values presented in this article are thus the lower limit estimations, since if HClO₃ and HClO₄ do not get charged at the collision limit, the calibration factor would be higher, leading to higher concentrations. In addition, considering the potential inlet losses is likely similar to H₂SO₄, it is reasonable to predict the

uncertainties for HClO_3 and HClO_4 measurement to be at least a factor of two. This information has now been added to the revised Methods section.

(Revised main manuscript, Line 375-378): The sum uncertainties of the HClO_3 and HClO_4 measurement from the collision limit of the target compound with its charger ions and potential inlet losses were predicted to be at least a factor of two. This maximum sensitivity resulted in lower limit estimates of the HClO_3 and HClO_4 concentrations.

References:

- Sipilä, M. et al. Molecular-scale evidence of aerosol particle formation via sequential addition of HIO_3 . *Nature* **537**, 532-534 (2016).
- Beck, L. J. et al. Differing Mechanisms of New Particle Formation at Two Arctic Sites. *Geophys. Res. Lett.* **48**, e2020GL091334 (2021).

(1.7) Lines 287-288: Provide more information about the “zero measurements”. The HEPA filter removes particles, but with what efficiency does it remove HClO_3 and HClO_4 ? At what frequency were zero measurements conducted during each of the field campaigns?

Reply: The HEPA filter is known for removing particles in the air, but our previous field measurements also used the HEPA filter for removing gaseous H_2SO_4 , as the HEPA filter contains a lot of surfaces which can promote the loss of acid or non-volatile gas. We do not use the activated carbon filter as it provides a higher background signal during the zeroing measurement. From the measured ambient signal of this study, we can clearly see that the background raw signals of ClO_3^- and ClO_4^- decay together with the HSO_4^- to near zero during the zeroing measurements (Fig. R1.4). We did not see the potential artefacts from the HEPA filter (e.g., off-gassing) during our measurement period. Therefore, we believe that the HEPA filter can also remove the HClO_3 and HClO_4 efficiently.

Occasional zero measurements were conducted with HEPA filter for at least 40 min in both Greenland and MOSAiC. Conducting zeroing measurement in MOSAiC is very challenging, as the inlet of CI-API-TOF is located on the edge of the platform of RV Polarstern but covers almost all seasons (more measurements during the winter season). We have revised the sentence to include more information on the zero measurements.

(Revised main manuscript, Line 318-320): The zero measurements were conducted occasionally with a high-efficiency particulate air (HEPA) filter for at least 40 min each measurement in both Greenland and MOSAiC, which cover different seasons during the measurement period.

Fig. R1.4 | Zero measurements in Greenland. Example of zero measurements in Greenland conducted on (a) 31 March 2015 and (b) 8 May 2015. The shaded area (light green) is to the zero measurement period. The ClO_3^- , ClO_4^- and HSO_4^- signals correspond to HClO_3 , HClO_4 and H_2SO_4 , respectively.

(1.8) Extended Data Fig. 5 presents HClO_3 and HClO_4 levels at <5 ppb, 5-30 ppb, and >30 ppb O_3 , but only means and ranges are presented, with no statistical analysis. Lines 194-196 states “higher mean HClO_4 concentrations [were] observed at higher O_3 levels”, but mean HClO_4 is actually highest at 5-30 ppb in Extended Data Fig. 5. Also, no statistical test is provided to support the author’s assertion. Further, measurement uncertainties, also not discussed, are not considered in this comparison. In addition, the authors elsewhere consider <20 ppb O_3 to be an ozone depletion event, so it would seem more appropriate to divide the data by <5 ppb, 5-20 ppb, 20-30 ppb, and >30 ppb.

Reply: Statistical analysis (ANOVA analysis) has been performed and the results show that there is significant difference ($p > 0.05$) between the concentrations of HClO_4 under different ranges of O_3 levels. We have divided the data to 4 different ranges as suggested, and added the uncertainty of the HClO_4 measurements to the revised Extended Data Fig. 5.

Extended Data Fig. 5 | HClO₄ under different O₃ levels. Distribution of HClO₄ under a different range of O₃ concentrations at (a) the Villum Research Station, Greenland; and (b) during the MOSAiC expedition from 22 February to 30 April 2020. Note that significant concentrations of HClO₃ and HClO₄ were frequently observed when the O₃ concentration was below 30 ppb (also refer to Fig.1 and Fig.2) and when O₃ is less than 5 ppb, we defined it as ‘complete’ depletion of O₃. In this plot, all the data points below the LODs were removed. The uncertainty of a factor of two for the HClO₄ measurements has been shown in the plot (grey shaded-area).

(1.9) The following sentences that provide background information (e.g. reaction mechanisms, vapour pressures, Henry’s Law constants, and other material) are missing references that need to be added: Lines 57-63, 79-81, 129, 145-147, 218, 233, and 237.

Reply: Thanks. Relevant references have been added to these sentences or the figure captions in the revised manuscript. For instance, the following references have been cited:

Jaeglé, L., Yung, Y. L., Toon, G. C., Sen, B. & Blavier, J.-F. Balloon observations of organic and inorganic chlorine in the stratosphere: The role of HClO₄ production on sulfate aerosols. *Geophys. Res. Lett.* **23**, 1749-1752 (1996).

Sander, R. Compilation of Henry’s law constants (version 4.0) for water as solvent. *Atmos. Chem. Phys.* **15**, 4399-4981 (2015).

Baran, K. P. & Gad, S. Perchloric Acid, in *Encyclopedia of Toxicology (Third Edition)* (ed Philip Wexler), 796-798 (Academic Press, 2014).

Custard, K. D., Pratt, K. A., Wang, S. & Shepson, P. B. Constraints on Arctic atmospheric chlorine production through measurements and simulations of Cl₂ and ClO. *Environ. Sci. Technol.* **50**, 12394-12400 (2016).

Stone, D., Whalley, L. K. & Heard, D. E. Tropospheric OH and HO₂ radicals: field measurements and model comparisons. *Chem. Soc. Rev.* **41**, 6348-6404 (2012).

Thompson, C. R. *et al.* Interactions of bromine, chlorine, and iodine photochemistry during ozone depletions in Barrow, Alaska. *Atmos. Chem. Phys.* **15**, 9651-9679 (2015).

Zhu, R. & Lin, M.-C. Ab initio study of ammonium perchlorate combustion initiation processes: unimolecular decomposition of perchloric acid and the related OH+ ClO₃ reaction. *Phys. Chem. Comm.* **4**, 127-132 (2001).

Atkinson, R. *et al.* Evaluated kinetic and photochemical data for atmospheric chemistry: Volume III gas phase reactions of inorganic halogens. *Atmos. Chem. Phys.* **7**, 981-1191 (2007).

(1.10) The [Cl] stated on Line 218 (without a reference) and SI Line 76 are low compared to that reported by Custard et al (2016, Environ. Sci. Technol.), who derived concentrations from simultaneous measurements of Cl₂ and ClO.

Reply: The concentration of Cl has been updated to 1×10⁶ molecules cm⁻³ according to the value derived by Custard et al. (2016), including for the calculation of OCIO+Cl (refer to reply 1.3). The citations have been added to the revised text.

(Revised main manuscript, Line 240-242): Assuming typical Cl (1×10⁶ molecules cm⁻³)³ and OH (5 × 10⁵ molecules cm⁻³)⁴ concentrations in the Arctic, the loss rates of $k_{Cl+HClO_4}[Cl]$ and $k_{OH+HClO_4}[OH]$ are estimated to be 1.0×10⁻²⁵ and 2.9×10⁻⁷ s⁻¹, respectively.

References:

- Custard, K. D., Pratt, K. A., Wang, S. & Shepson, P. B. Constraints on Arctic atmospheric chlorine production through measurements and simulations of Cl₂ and ClO. *Environ. Sci. Technol.* **50**, 12394-12400 (2016)

(1.11) Figure 3: I suggest adding references for the mechanism presented, as it is based primarily on previous work. As discussed above, NO and NO₂ should also be included. Also, I suggest replacing the generic “Ground-Surface” with “Snowpack”, as this is specific to the Arctic springtime.

Reply: The references for the mechanisms have been added to the revised caption of Fig. 3. We have also added the potential influence of NO and NO₂. The ‘ground-surface’ has been changed to snowpack in the figure and the revised main text. The revised Fig.3 is as below.

Fig. 3 | Atmospheric formation and fate of HClO₃ and HClO₄. Simplified diagram of the proposed potential formation mechanism of gas-phase HClO₃ (blue) and HClO₄ (pink) in the Arctic boundary layer during springtime after polar sunrise. The produced HClO₃ and HClO₄ can be taken up by the surface of

aerosols and converted into ClO_3^- and ClO_4^- , respectively. The deposition of aerosols and/or the direct deposition of gas-phase HClO_3 and HClO_4 onto the ground surface, such as snowpacks, can function as a sink for reactive chlorine in the Arctic troposphere. The reactions are based on the literatures^{2,63,77}.

(1.12) What is the correlation between $^{35}\text{ClO}_3^-$ and $\text{H}^{35}\text{ClO}_3\text{NO}_3^-$? Adding a scatter plot and correlation coefficient would further support the measurement of HClO_3 as ClO_3^- .

Reply: The scatter plot between $^{35}\text{ClO}_3^-$ and $\text{NO}_3(\text{H}^{35}\text{ClO}_3)^-$, as shown in Fig. R1.5, and additional information has been added to the revised Extended Fig. 8 to show the signal slope of 4.11 with a correlation coefficient (R^2) of 0.97.

Fig. R1.5 | Scatter plot of $^{35}\text{ClO}_3^-$ versus $\text{NO}_3(\text{H}^{35}\text{ClO}_3)^-$. The normalized data of $^{35}\text{ClO}_3^-$ versus $\text{NO}_3(\text{H}^{35}\text{ClO}_3)^-$ were obtained during the MOSAiC campaign from 22 February to 30 April 2020.

(1.13) Abstract, Paragraph on Lines 85-92, and Methods: Add mention of measurement of HClO_3 at Ny-Ålesund, as this is an important supporting measurement presented in Extended Data Fig. 1.

Reply: The measurement of HClO_3 at Ny-Ålesund has been added to:

(Revised main manuscript, Line 48-52, Abstract): Significant levels of HClO_3 were observed during springtime at Greenland (Villum Research Station), Ny-Ålesund research station and over the central Arctic Ocean, on-board research vessel Polarstern during the Multidisciplinary drifting Observatory for the Study of the Arctic Climate (MOSAIC) campaign, with estimated concentrations up to 7×10^6 molecule cm^{-3} .

(Revised main manuscript, Line 103-106): Measurements were made via mass spectrometry in the Arctic at the Villum Research Station, Greenland, Ny-Ålesund, Svalbard, and over the central Arctic Ocean onboard research vessel (RV) Polarstern during the Multidisciplinary drifting Observatory for the Study of the Arctic Climate (MOSAIC) expedition.

The information on the Ny-Ålesund site has been added to the sampling location section (*Revised main manuscript, Line 300-303*):

We also conducted measurements at the atmospheric observatory, Gruebadet, located 2 km southeast of Ny-Ålesund, Svalbard (78° 55' N, 11° 56' E), from 28 March to 30 May 2017 (spring). Detailed information on the Greenland and Ny-Ålesund sampling site can be found in Beck *et al.*⁴²

In addition, relevant information on the setup and measurements detail at Ny-Ålesund have been included in the text of the revised Methods, as following (*Revised main manuscript, Line 312-314*).

At Ny-Ålesund, the inlet tube length was 2 m (outer diameter of 3/4 inch) and the sample was taken through the roof (height = 2 m a.g.l.), with a flow rate of 10 lpm.

(1.14) The paragraph on Lines 95-114 is critical to the manuscript. The flow of this paragraph could be improved, as I had to read it several times to understand it switching between measurements in Greenland, Svalbard, and during MOSAiC. This was also particularly confusing since Svalbard wasn't measured in the preceding paragraph on Lines 85-92. I suggest revising the paragraph to focus on clarity.

Reply: Thank you for the suggestion. We have revised the flow of the paragraph accordingly.

(Revised main manuscript, Line 112-131):

Fig. 1 shows the time series of HClO₃ and HClO₄ measured at the Villum Research Station, Greenland and during the MOSAiC campaign. Our observations in Greenland indicated a significant increase in the HClO₃ signal measured with a nitrate-chemical ionization atmospheric pressure interface time-of-flight mass spectrometry (CI-API-TOF; Methods), with an estimated concentration up to 1×10⁶ molecules cm⁻³ in the spring of 2015. The HClO₃ concentration began to increase when sunlight increased towards the end of February. HClO₃ exhibited no diurnal pattern, but a unique feature is that a significant increase in HClO₃ concentration was observed during ozone depletion events, as shown in Fig. 1a. Typically, HClO₃ peaked under relatively low O₃ levels (< 30 ppb). We also measured HClO₃ with a nitrate-CI-API-TOF instrument during the MOSAiC expedition in 2019/2020 (Methods section). Similar to the observation in Greenland, the measurements onboard RV Polarstern in different seasons revealed a clear increment in HClO₃ starting at the end of February, when solar radiation started to increase after the polar night. The estimated springtime concentration of HClO₃ during the MOSAiC campaign ranged from approximately 1×10⁵ to 7×10⁶ molecules cm⁻³ (Fig. 1b). An increase in HClO₃ was also correlated with the depletion of O₃ over the Arctic Ocean during the MOSAiC campaign. The HClO₃ levels are relatively low in the other seasons, with concentrations less than 10⁵ molecule cm⁻³ (Extended Data Fig. 1). Further measurements at Ny-Ålesund, Svalbard also indicated the presence of HClO₃ in springtime, with concentrations up to 8×10⁵ molecules cm⁻³ (Extended Data Fig. 2). However, without direct measurement of O₃ during the campaign at Svalbard, we are not able to evaluate the relationship between HClO₃ and O₃.

(1.15) Line 108: The statement of “yearlong measurements” is misleading, as Extended Data Figure 2 only presents measurements from Jan.-Jun. Please rephrase the text.

Reply: Thanks for pointing this out. The sentence has been rephrased to the following:

(Revised main manuscript, Line 121-125): Similar to the observation in Greenland, the measurements onboard RV Polarstern in different seasons revealed a clear increment in HClO₃ starting at the end of February, when solar radiation started to increase after the polar night. The estimated springtime concentration of HClO₃ during the MOSAiC campaign ranged from approximately 1×10⁵ to 7×10⁶ molecules cm⁻³ (Fig. 1b).

(1.16) Line 162: The statement “snowpacks/sea-ice emissions of Cl₂ and BrCl” is misleading, as the references cited do not suggest emission of Cl₂ or BrCl from sea ice because the sea ice surface is

buffered and R8 requires an acidic surface. To my knowledge, Cl₂ and BrCl have only been measured to be emitted from snow. Please correct.

Reply: We have corrected the statement.

(Revised main manuscript, Line 121-125): The snowpack emissions of Cl₂ and BrCl⁹⁻¹¹ undergo fast photolysis, leading to the production of Cl atoms, which subsequently react with O₃ to form ClO (R1-R3)³

(1.17) Lines 163-164: Reference #3 (Custard et al. 2016, Environ. Sci. Technol.) would be appropriate to add here for the reaction of Cl with O₃ to form ClO.

Reply: Added.

(1.18) Line 167: Thompson et al. (2015, Atmos. Chem. Phys.) also measured ClO.

Reply: Thanks for pointing this out. This has been cited in the revised main text.

(1.19) Line 335: Fix the typo in the equation.

Reply: Corrected.

$$[\text{HClO}_x] = C \cdot \frac{\text{ClO}_x^- + (\text{HClO}_x) \cdot \text{NO}_3^-}{\text{NO}_3^- + (\text{HNO}_3) \cdot \text{NO}_3^-} \quad (\text{Eq.1})$$

(1.20) Extended Data Figure 4a & Extended Data Figure 7a: Fix y axis typos.

Reply: The typos in the figures have been fixed. The revised figures are as below:

Extended Data Fig. 4 | HClO_3 , HClO_4 , H_2SO_4 and aerosol surface area. Expanded view between HClO_3 , HClO_4 and the measured H_2SO_4 and aerosol surface area at (a) the Villum Research Station, Greenland (18 March to 17 April 2015) and (b) during the MOSAiC campaign (1 March to 28 April 2020). The gap in the aerosol surface area in the data set is either due to the removal of invalid data (i.e., particle counts too low or due to local emissions), instrument maintenance or instrumentation offline events.

Extended Data Fig. 7 | Aerosol surface area and humidity during the MOSAiC campaign. (a) Aerosol surface area estimated according to the SMPS measurements and (b) H₂O measured throughout the MOSAiC campaign from January to July 2020. The data of the aerosol surface area and H₂O exhibit a 30-min average of time resolution.

(1.21) Extended Data Fig 6: Add to the caption that these photolysis rates (presumably) correspond to May 1 at Villum (according to the Methods), as these rates would be significantly lower for early spring during MOSAiC, for example. It is important to be clear in the caption as to what is being presented.

Reply: Agree. We have added this information to the revised caption of Extended Data Fig. 6.

Extended Data Fig. 6 | Photolysis of HClO₃ and HClO₄. (a) Modelled absorption cross-section (σ) of HClO₃ and HClO₄. Photolysis rate calculated based on the absorption cross-section for (b) HClO₃ and (c) HClO₄ in the Arctic environment with a latitude of 81° 21' N (similar to the location of Villum Research Station, Greenland) on 1 May 2020.

(1.22) SI Line 49 – Ref #7 didn't measure ClO. However, Custard et al (2016, Environ. Sci. Technol.) and Thompson et al. (2015, Atmos. Chem. Phys.) both did measure ClO, and these publications are not cited here.

Reply: The reference has been corrected. Both publications have been cited in the revised SI text (*Line 81*).

Reviewer #2 (Remarks to the Author):

(2.1) Dear Yee Jun Tham et al.

It is with great pleasure that I read your manuscript on providing measurement evidence for novel chlorine species that terminate atmospheric cycles. Clearly, the identification of this termination step is a result noteworthy to the atmospheric science, arctic science, and beyond. Taken the negative impact on ecosystems, that the authors note, I'm confident that this result will inspire other scientist in the wider environmental community.

The manuscript is well written, the methods of data acquisition appear sound to me, conclusions are fully supported by data and/or well-argued based on relevant literature. I'm particularly impressed by discussion data from several regions/field campaigns to draw your conclusions which nicely emphasizes the locally wide spread importance of the findings.

Reply: We thank the reviewer for the positive and constructive suggestions.

(2.2) My only comment deals with the impact of surface processes as source of the oxyacids and later as sink process. If I understood correctly, measurements are reported from one specific height above the ground for each data set? Profiles or flux measurements were not done. What is unclear to me is what "air mass originated from the near ground surface» exactly means. I assume that those air masses are in contact with the ground? Would then not also any oxyacid formed in the air – as opposed to in the surface snow- partitioning to the snow and be lost from the atmosphere? In other words, the argument of strong partitioning to the snow should also hold for gas-phase species and not only those (originally) at the surface.

Generally, I'm not fully convinced by the argumentation about sticking to the surface. Would you not agree that a) there is some uncertainty due to pH and/or acid-base equilibria at the air-ice interface on the partitioning? And b) I miss a more discussion on equilibrium processes. To start general, what is not clear to me is if you refer to the vapor pressure and to KH as parameters that actually describe the partitioning to ground surface and to aerosol, or just as indicators as the true parameters are not known. For snow surfaces, for example, I would have expected the Langmuir partitioning constant. As for the KH which describes the partitioning between the gas and solution phase. Which concentration would you need in the liquid phase to maintain the measured gas-phase concentrations? Taken that humidity is low and temperature is low, aerosol and aerosol deposits in snow are often highly concentrated. Further, the absence of high concentrations in winter and in summer could also indicate that snow and sun is needed and not necessarily that surface snow plays no role. In summary, this argumentation would benefit from a few clarifying numbers – that I might have missed elsewhere in the text. If so, my apologies. This includes the sampling height, a statement about transport limitations to the surface from that height/to that height, bulk concentrations needed to maintain the measured gas-phase concentrations. This comment refers to lines 137-155 and to some extend to 232-238. Overall, I recommend publication after tackling these minor suggestion of mine.

Reply: Thanks for your insightful thoughts. Yes, the reported measurements for each campaign are from one specific height above the 'ground' level and profile measurements are not available in this study. We have noted the height of our measurements in the revised Methods (see *Revised main manuscript, Line 311, 313 and 317*). It is true that the air mass is always in contact with the 'ground' as our measurements are conducted at the ground level. Although the air mass is confined to near surface (a well-known phenomenon for O₃ depletion events in the Arctic), we do not think that the partitioning from snow and aerosol surface will be the major source of HClO₃ and HClO₄, but the snow surface may be an additional loss pathway for these chlorine oxyacids. Our explanations are as below.

First, our rationalization for HClO₃ and HClO₄ sticking to the surface originates from a study of Zhong *et al.* (2020) which shows that HOCl sticks preferentially to the ice surface via the –OH, interacting strongly via hydrogen bonds with water. We agree with the reviewer that there is some uncertainty of the partitioning due to pH and/or acid-base equilibria at the air-ice interface as has been seen in HCl and Cl-

system. However, so far, there is no evidence showing that ClO_4^- will be released as gas-phase HClO_4 to the atmosphere under ambient conditions, other than the possible degradation of ClO_4^- to Cl^- and O_2 by specific biological processes (Sijimol *et al.*, 2015).

Second, the vapour pressure and Henry's law constant (K_H) of HClO_4 values together with the information on their solubility in the manuscript are to indicate that the chlorine oxyacids are non-volatile and prefer to accommodate on the liquid surface, as the actual parameters are not known. Our simple calculation of the air-water equilibrium, based on the K_H of HClO_4 (in L atm mol^{-1}) with the assumption of an enclosed environment (partial pressure = $K_H \times$ concentration in liquid), requires about 35 nM in the liquid phase to maintain the observation of gas-phase 1×10^6 molecules cm^{-3} (maximum value of the observation). This value is much higher than those ClO_4^- concentrations reported in the ice-core and aerosol samples from the Arctic and Antarctica (e.g., Furdui and Tomassini, 2010; Jiang *et al.*, 2021). In comparison, we also calculated the equilibrium of a more volatile acid, HCl ($K_H = 0.2 \text{ mol m}^{-3} \text{ Pa}^{-1}$). In order to maintain 5×10^9 molecules cm^{-3} of HCl concentration in the atmosphere (de Lange *et al.*, 2012), it requires about 127 nM in the liquid phase, which is within the range of Cl^- observed in ice-core (e.g., Furdui and Tomassini, 2010). In other words, it is very unlikely that the partitioning from the surface (snow or aerosol) to the atmosphere will be the dominant source of gas-phase HClO_4 (and HClO_3). This information has been added to the section S1 of revised supplementary information (see below).

Third, we do not think that the surface snow will play a direct role in the emission of HClO_3 and HClO_4 as we have discussed in the original main text that the snow surface will emit chlorine species (i.e., Cl_2 , BrCl) and undergo photochemical processing leading to the formation of HClO_3 and HClO_4 (refer to Fig.3). Hence, it is true to some extent that the snow and sun are needed for the formation of gas-phase HClO_3 and HClO_4 . However, the absence of high concentrations in winter and in summer is most likely due to the lacking of photochemical processes and/or increase of the sink (e.g., increase of humidity during summertime).

Fourth, without the actual parameters and relevant information (e.g., the vertical wind pumping speed and diffusion, Langmuir constant or uptake coefficient on snow for HClO_3 and HClO_4 , chemistry in snow, snow grain size, etc.), we are not able to estimate the loss rate of chlorine oxyacids on the snow surface. In the meantime, we cannot exclude the transport limitations to the surface and the possibility of direct loss of chlorine oxyacids to the snow surface; hence, we have added this information to the revised discussion in the main text and Fig.3.

(Revised SI, Line 48-60):

S1. Estimation of air-water equilibrium for HClO_4 . The low vapour pressure, high Henry's law constant (K_H) values and high solubility of HClO_4 in water indicate that the chlorine oxyacids are non-volatile and prefer to accommodate on the liquid surface. We performed a simple calculation of the air-water equilibrium based on the K_H of HClO_4 (in L atm mol^{-1}) with the assumption of an enclosed environment (partial pressure = $K_H \times$ concentration in liquid), it requires about 35 nM in liquid phase to maintain the observation of gas-phase HClO_4 concentration of 1×10^6 molecules cm^{-3} (maximum value of our observation). This value is much higher than those ClO_4^- concentrations reported in the snow/ice-core and aerosol samples from the Arctic and Antarctica^{1,2}. For comparison, we also calculated the equilibrium of a more volatile acid, HCl ($K_H = 0.2 \text{ mol m}^{-3} \text{ Pa}^{-1}$). In order to maintain 5×10^9 molecules cm^{-3} of HCl concentration in the atmosphere³, it requires about 127 nM in liquid phase, which is within the range of Cl^- observed in snow/ice-core¹. In other words, it is very unlikely that the emission from aerosol or snow will be the dominant source of atmospheric HClO_4 (and HClO_3).

(Revised main manuscript, Line 264-270):

Therefore, the most relevant fate of HClO_3 and HClO_4 is their heterogeneous uptake by the surface of aerosol particles and subsequent deposition on the ground surface or undergo wet deposition. However, we cannot exclude the possibility of direct loss of these chlorine oxyacids to the snow surface (Fig. 3). In fact, our hypothesis is supported by previous studies in polar regions that have measured a considerable

amount of ClO_3^- and ClO_4^- in ice cores^{17,18} and snow⁴⁹, where atmospheric sources are strongly implicated.

References:

- Zhong, J., Zhang, W., Wu, S., An, T. & Francisco, J. S. Molecular interaction and orientation of HOCl on aqueous and ice surfaces. *J. Am. Chem. Soc.* **142**, 17329-17333 (2020).
- Sijimol, M. R. *et al.* Review on fate, toxicity, and remediation of perchlorate. *Environ. Forensics* **16**, 125-134 (2015).
- de Lange, A. *et al.* HCl and ClO in activated Arctic air; first retrieved vertical profiles from TELIS submillimetre limb spectra. *Atmos. Meas. Tech.* **5**, 487-500 (2012).
- Furdui, V. I. & Tomassini, F. Trends and sources of perchlorate in Arctic snow. *Environ. Sci. Technol.* **44**, 588-592 (2010).
- Jiang, S., Shi, G., Cole-Dai, J., An, C. & Sun, B. Occurrence, latitudinal gradient and potential sources of perchlorate in the atmosphere across the hemispheres (31°N to 80°S). *Environ. Int.* **156**, 106611 (2021).

Detailed comments:

(2.3) LI 79-81: This sounds a little awaked. The oxyacids might have always been there, without detection. How then, can detecting them, perturb the chemistry? It can “perturb” our understanding of it.

Reply: Thanks for pointing this out. We have revised the statement.

(Revised main manuscript, Line 96-98): Therefore, the atmospheric occurrence of chlorine oxyacids could enhance the chlorine sink, thereby affecting the oxidation capacity of the atmosphere and posing environmental threats once deposited to the Earth’s surface.

(2.4) LI 125: “in the arctic troposphere”. This is -so far- limited to your measurement height, is it not? This is not the whole troposphere.

Reply: Thanks. We have corrected it to ‘in the Arctic boundary layer during the springtime’ (*Revised main manuscript, Line 142*).

(2.5) LI153: For which data sets was that pattern investigated? Spring only, or also summer and winter? I think this needs to be limited to spring. Please specify in the text.

Reply: Yes, the data set used for the pattern comparison is during the springtime as noted in the caption of Extended Data Fig. 4. We have now specified it in the revised main text.

(Revised main manuscript, Line 171-173): Furthermore, the observed lack of a clear pattern between HClO_3 and HClO_4 and the aerosol surface area during the springtime (Extended Data Fig. 4) may point to their limited partitioning from the aerosol phase.

Reviewer #3 (Remarks to the Author):

(3.1) Review of paper “Widespread detection of chlorine oxyacids in the Arctic Atmosphere” by Tham et al.

This is an interesting paper that documents the detection of HClO_3 and HClO_4 in the Arctic boundary layer. I was kind of excited to see this finding. The authors have attempted to quantify their detected amounts (which are pretty small) and then tried connecting them to the boundary layer episodic ozone depletion events. At first blush, it all looks OK.

However, I have some significant reservations. It is possible that the authors can answer these concerns. Then, the question becomes, is the detection of these chemicals more a testament to the modern analytical capabilities, or does it have any significance. I will opine on the latter point at the end.

Reply: We thank the referee for the detailed comments and constructive suggestions.

Major concerns:

(3.2) Perchlorates and chlorates are ubiquitous in the aqueous systems. There is vast literature on it. These are often due to the widespread use of perchlorates for various purposes such as rocketry, flares, matches, etc. The aqueous concentrations of perchlorates have been documented. So, my first concern is: How do the authors know that the detected HClO_3 or HClO_4 are due to chlorine chemistry in the atmosphere and not simply reintroducing these chemicals from the aqueous system? It is important to note that the very processes that lead to the release of halogens in the Arctic could also lead to the gas phase release of perchloric and chloric acids. This process is equal to having up arrows from the ground/surface in Figure 3. (Please note that acidification of the droplets could conceivably release these acids to the gas phase....)

Reply: We acknowledge, as cited in the original manuscript, that there is a vast literature on the aqueous concentrations of perchlorates and chlorates around the world, with natural and anthropogenic sources. As for our detection of gas-phase HClO_3 or HClO_4 , we show that the emission of aqueous phase chlorates and perchlorates is not the dominant source for atmospheric HClO_3 and HClO_4 , and that they are predominantly produced in the Arctic boundary layer.

First, as have been shown in the original manuscript, the vapour pressure and Henry's law constant (K_H) of HClO_4 (and HClO_3) values together with the information on their solubility point to the fact that these chlorine oxyacids are non-volatile and prefer to accommodate on the liquid surface. Our simple calculation of the air-water equilibrium based on the K_H of HClO_4 (in L atm mol^{-1}) with the assumption of an enclosed environment (partial pressure = $K_H \times$ concentration in liquid), requires about 35 nM in the liquid phase to maintain the observation of gas-phase 1×10^6 molecules cm^{-3} (maximum value of the observation). This value is much higher than those ClO_4^- concentrations reported in the ice-core and aerosol samples from the Arctic and Antarctica (e.g., Furdui and Tomassini, 2010; Jiang *et al.*, 2021). In comparison, we also calculated the equilibrium of a more volatile acid, HCl ($K_H = 0.2 \text{ mol m}^{-3} \text{ Pa}^{-1}$). In order to maintain 5×10^9 molecules cm^{-3} of HCl in the atmosphere (de Lange *et al.*, 2012), it requires about 127 nM in the liquid phase, which is within the range of Cl^- observed in ice-core (e.g., Furdui and Tomassini, 2010). In other words, it is very unlikely that the partitioning from the aqueous phase (snow or aerosol) to the atmosphere will be the dominant source of gas-phase HClO_4 (and HClO_3).

Second, the observed HClO_3 and HClO_4 are unlikely from long-range transport (e.g., from the polluted continental region). The atmospheric lifetime of HClO_3 and HClO_4 due to the heterogeneous loss on aerosol is about 1 h under typical aerosol surface area of $20 \mu\text{m cm}^{-3}$ in the Arctic (please refer to our calculation of the heterogeneous loss of HClO_3 and HClO_4 on aerosol and Table S2 in supplementary information). Although a previous study has suggested the possibility of stratospheric injection (Liang *et al.*, 2009), we do not think that this is the major source since this previous study showed that vertical transport is slow in Arctic springtime, definitely slower than the chlorine oxyacids lifetime against

heterogeneous loss on aerosol or snow surfaces. This is consistent with our measurements that show the absence of a ^{36}Cl peak in the spectrum, which suggest a lack of cosmogenic chlorine (e.g., Cao *et al.*, 2019; also see reply to comment 3.4). These pieces of evidence mean that the atmospheric HClO_3 and HClO_4 cannot travel too long distance and mostly are produced through the atmospheric chlorine chemistry within the region.

Third, we agree with the reviewer that there is some uncertainty of the partitioning due to pH and/or acid-base equilibria in the aerosol/snow surface as has been seen in HCl and Cl^- system. However, so far, there is no evidence showing that ClO_4^- will be released as gas-phase HClO_4 to the atmosphere under ambient conditions. The possible loss of ClO_4^- is via degradation of ClO_4^- to Cl^- and O_2 by specific biological processes (e.g., Sijimol *et al.*, 2015).

Our justifications are highly consistent with previous studies of aqueous phase perchlorates. As shown in the review paper by Cao *et al.* (2019), previous research in the polar region (Antarctica) and Canada (near the Arctic) has indicated the atmospheric source of perchlorate.

We have added the calculation of the air-water equilibrium for HClO_4 to the revised supplementary information.

(Revised SI, Line 48-60):

S1. Estimation of air-water equilibrium for HClO_4 . The low vapour pressure, high Henry's law constant (K_H) values and high solubility of HClO_4 in water indicate that the chlorine oxyacids are non-volatile and prefer to accommodate on the liquid surface. We performed a simple calculation of the air-water equilibrium based on the K_H of HClO_4 (in L atm mol^{-1}) with the assumption of an enclosed environment (partial pressure = $K_H \times$ concentration in liquid), it requires about 35 nM in liquid phase to maintain the observation of gas-phase HClO_4 concentration of 1×10^6 molecules cm^{-3} (maximum value of our observation). This value is much higher than those ClO_4^- concentrations reported in the snow/ice-core and aerosol samples from the Arctic and Antarctica^{1,2}. For comparison, we also calculated the equilibrium of a more volatile acid, HCl ($K_H = 0.2 \text{ mol m}^{-3} \text{ Pa}^{-1}$). In order to maintain 5×10^9 molecules cm^{-3} of HCl concentration in the atmosphere³, it requires about 127 nM in liquid phase, which is within the range of Cl^- observed in snow/ice-core¹. In other words, it is very unlikely that the emission from aerosol or snow will be the dominant source of atmospheric HClO_4 (and HClO_3).

References:

- Furdui, V. I. & Tomassini, F. Trends and sources of perchlorate in Arctic snow. *Environ. Sci. Technol.* **44**, 588-592 (2010).
- Jiang, S., Shi, G., Cole-Dai, J., An, C. & Sun, B. Occurrence, latitudinal gradient and potential sources of perchlorate in the atmosphere across the hemispheres (31°N to 80°S). *Environ. Int.* **156**, 106611 (2021).
- de Lange, A. et al. HCl and ClO in activated Arctic air; first retrieved vertical profiles from TELIS submillimetre limb spectra. *Atmos. Meas. Tech.* **5**, 487-500 (2012).
- Liang, Q., Douglass, A. R., Duncan, B. N., Stolarski, R. S. & Witte, J. C. The governing processes and timescales of stratosphere-to-troposphere transport and its contribution to ozone in the Arctic troposphere. *Atmos. Chem. Phys.* **9**, 3011-3025 (2009).
- Sijimol, M. R. et al. Review on fate, toxicity, and remediation of perchlorate. *Environ. Forensics* **16**, 125-134 (2015).
- Cao, F. et al. Worldwide occurrence and origin of perchlorate ion in waters: A review. *Sci. Total Environ.* **661**, 737-749 (2019).

(3.3) The authors could not calibrate their system for detecting the two acids. So, they have gone through an indirect method to convert measured mass spec signals into concentrations. First, I am not sure they could not have devised a way to calibrate their system. Second, I do not see any discussion of the

uncertainties in their quoted concentrations. Third, the methodology section did not convince me that converting the mass spec signal to concentration is devoid of problems.

Reply: Thank you for the comment. We would like to clarify that the nitrate CI-API-TOF is designed for direct online measurement of trace chemical compounds in the ambient gas samples (down to parts per quadrillion (ppq) levels). High concentrations (e.g., tens of ppb) of chemical compounds can saturate the system and deplete the reagent ion (NO_3^-) leading to inaccurate measurement. In order to design a suitable calibration method, we need to produce a quantifiable low concentration of gas-phase standard. Unfortunately, there is currently no reliable way to produce low levels of HClO_3 and HClO_4 (i.e., pptv levels) as the calibration standard. If we evaporate a high concentration of HClO_4 , there is a safety issue concern in the laboratory, which we acknowledged in the original Methods section.

Instead, we have opted for an indirect calibration method by combining the quantum chemical calculation and calibration of H_2SO_4 , an alternative approach in the field of CI-API-TOF for the detection of species that cannot be directly calibrated due to a lack of authentic standards or generation methods (Sipilä *et al.*, 2016; Iyer *et al.*, 2016; Lopez-Hilfiker *et al.*, 2016; Wang *et al.*, 2021). In the instrument, ion clusters, formed from reactions between analytes and reagent ions, are guided and focused by ion optics during transmission to the detector. The electric forces applied to the clusters enhance their collision energies with carrier gas molecules. Analytes that bind to the reagent ions with enthalpies close to a critical level are likely detected at maximum sensitivity (kinetic-limited charging) by the instrument (Lopez-Hilfiker *et al.*, 2016; Wang *et al.*, 2021). Our quantum chemical calculation results clearly show that binding free energies of NO_3^- with HClO_3 and HClO_4 are close to or lower than that of the critical value of NO_3^- with H_2SO_4 . Although the cluster of HClO_3 with NO_3^- has a slightly lower binding free energy than the value of the cluster of NO_3^- with H_2SO_4 , the dissociation pathway is proton transfer ($\text{HClO}_3 \cdot \text{NO}_3^- \rightarrow \text{ClO}_3^- + \text{HNO}_3$); HClO_3 can thus be considered as a maximum sensitivity species detectable as ClO_3^- ions after proton transfer, consistent with the detection of H_2SO_4 (both $\text{H}_2\text{SO}_4 \cdot \text{NO}_3^-$ and HSO_4^-). Therefore, we can conclude that the detection of HClO_3 and HClO_4 by our instrument is very likely as efficient as the detection of H_2SO_4 by NO_3^- , whose reaction rate is expected to occur within the kinetic limit range (Viggiano *et al.*, 1997). We acknowledge that this approach is likely not valid for analytes that require much higher binding free energies as we are unable to experimentally establish a correlation between sensitivities and binding free energies due to limited quantifiable species. We also note that when transferring the calibration factor from one species to another, the diffusivity difference should be accounted for since it affects the inlet line loss.

As discussed above, the major uncertainties of the measurement with a nitrate CI-API-TOF include the collision limit of the target compound with its charger ions (calibration) and potential inlet losses. Previous measurements of H_2SO_4 and HIO_3 with a nitrate CI-API-TOF estimated an uncertainty of +100/-50% or a factor of two (Sipilä *et al.*, 2016; Beck *et al.*, 2021). Since our quantum chemical calculation results suggest that the detection of HClO_3 and HClO_4 is very likely as efficient (at collision limit) as the detection of H_2SO_4 . The values presented in this article are thus the lower limit estimations, since if HClO_3 and HClO_4 do not get charged at the collision limit, the calibration factor would be higher, leading to higher concentrations. In addition, considering the potential inlet losses are likely similar to H_2SO_4 , it is reasonable to predict the uncertainties for HClO_3 and HClO_4 measurement to be at least a factor of two. We, therefore, have reported our measured concentration as a lower limit. This information has been added to the revised Methods section.

(Revised main manuscript, Line 375-378): The sum uncertainties of the HClO_3 and HClO_4 measurement from the collision limit of the target compound with its charger ions and potential inlet losses were predicted to be at least a factor of two. This maximum sensitivity resulted in lower limit estimates of the HClO_3 and HClO_4 concentrations.

References:

- Sipilä, M. *et al.* Molecular-scale evidence of aerosol particle formation via sequential addition of HIO₃. *Nature* **537**, 532-534 (2016).
- Iyer, S., Lopez-Hilfiker, F., Lee, B. H., Thornton, J. A. & Kurtén, T. Modeling the detection of organic and inorganic compounds using iodide-based chemical ionization. *J. Phys. Chem. A* **120**, 576-587 (2016).
- Lopez-Hilfiker, F. D. *et al.* Constraining the sensitivity of iodide adduct chemical ionization mass spectrometry to multifunctional organic molecules using the collision limit and thermodynamic stability of iodide ion adducts. *Atmos. Meas. Tech.* **9**, 1505-1512 (2016).
- Wang, M. *et al.* Measurement of iodine species and sulfuric acid using bromide chemical ionization mass spectrometers. *Atmos. Meas. Tech.* **14**, 4187-4202 (2021).
- Viggiano, A. A., Seeley, J. V., Mundis, P. L., Williamson, J. S. & Morris, R. A. Rate constants for the reactions of XO₃⁻(H₂O)_n (X = C, HC, and N) and NO₃⁻(HNO₃)_n with H₂SO₄: Implications for atmospheric detection of H₂SO₄. *J. Phys. Chem. A* **101**, 8275-8278 (1997).
- Beck, L. J. *et al.* Differing Mechanisms of New Particle Formation at Two Arctic Sites. *Geophys. Res. Lett.* **48**, e2020GL091334 (2021).

(3.4) Isotope geochemists have been looking at perchlorates for signals of stratospheric influence and other sources. It is not clear to me that the authors have considered this issue. I do not see any ³⁶Cl signals (Did they not see it?) suggesting a lack of cosmogenic chlorine. I acknowledge that the signals may be too small.... But, the authors have not said anything about it. So I wonder if they have considered this issue.

Reply: Thanks for the suggestion. We agree that the absence peak of ³⁶Cl for HClO₃ and HClO₄ may suggest a lack of cosmogenic chlorine (e.g., Cao *et al.*, 2019). This could be due to our instrument limitation in detecting extremely small signals. We have added a sentence on the absence of ³⁶Cl in the spectrum to the revised Methods.

(Revised main manuscript, Line 323-324): The absence of a ³⁶Cl peak in the spectrum may suggest a lack of influence from the stratospheric air mass⁶⁰.

Reference:

- Cao, F. *et al.* Worldwide occurrence and origin of perchlorate ion in waters: A review. *Sci. Total Environ.* **661**, 737-749 (2019).

(3.5) Can the isotope signals say anything about the origin of chlorine? It is possible that their detection limits (without preconcentration) are not sufficient to see it, but nothing was mentioned. Furthermore, they see ³⁵Cl/³⁷Cl ratios that are not really 3.1. (Incidentally, there are no error bars on any of these- I will comment on this below). It is also known that marine biology does a certain amount of fractionation. Are there any signals here? Lastly, did the authors detect any Cl₂ and ClBr? If yes, what were the isotope ratios?

Reply: We think that the chlorine isotope signals cannot give any further indication for the origin of the chlorine. Other than the absence of ³⁶Cl peaks in the spectrum, the ratio of ³⁵Cl/³⁷Cl for the processed raw signal is very close to the theoretical value of 3:1 (Fig. R3.1). We have added the information on the absence of ³⁶Cl peaks to the revised Methods (refer to reply to comment 3.4 above). Unfortunately, we did not see other fractionation of chlorine ions signal in the spectrum. This could be due to the fact that our instrument (using NO₃⁻ as reagent ions) is a very selective tool and not all analytes can be detected by this method (depending on their electron affinity). The measurements of Cl₂ and BrCl are also not available in these campaigns.

Fig. R3.1 | Scatter plot of $^{35}\text{ClO}_3^-$ versus $^{37}\text{ClO}_3^-$ signal. The raw data of $^{35}\text{ClO}_3^-$ and its isotope ($^{37}\text{ClO}_3^-$), and the red line represents the 3:1 ratio, which is the theoretical isotopic ratio of ^{35}Cl and ^{37}Cl .

(3.6) I am concerned about a lack of uncertainty treatment here. I have no idea how to interpret the data when there are no estimates of the uncertainties and no error bars on the data.

Reply: The estimated uncertainty of the measurement for HClO_3 and HClO_4 has been added to the revised manuscript (see the reply to comment 3.3 above). As for the error bar, we do not think it is appropriate to include an error bar in the time series figures because it will make the graph hard to read. Instead, we have added the estimated uncertainty to the revised caption of Fig.1, Extended Data Fig. 1 and Extended Data Fig. 2.

(3.7) Maybe I am missing it, but I did not see any data on the aerosol levels. Were they made? If yes, what do they look like? What was their composition? (I am sure that the authors are aware the entire finding about the halogen-induced episodic ozone depletion in the Arctic boundary layer was evoked by measurements of aerosol composition....)

Reply: Yes, measurements of aerosol levels were made with a scanning mobility particle sizer (SMPS) in both Greenland and MOSAiC campaigns (see Ancillary measurements in Methods). We have presented the aerosol levels with the predicted aerosol surface area, as shown in Extended Data Fig. 4 and Extended Data Fig. 7. To our knowledge, there is aerosol composition measurement from an aerosol mass spectrometer (AMS) during the MOSAiC campaign, but unfortunately, the aerosol compositions data are not publicly available yet. Furthermore, we do not think that the aerosol compositions data from the AMS will be important for the current analysis and can affect the conclusion of our study.

(3.8) There is a lot of sloppy writing. Here are a few examples: (a) this is a detection of HClO_3 and HClO_4 in the Arctic boundary layer, not all across the Arctic atmosphere. It definitely is not in the free troposphere or the stratosphere; (b) the role of chlorine in determining CH_4 lifetime is minimal. Whatever it does in the Arctic boundary layer does not change CH_4 lifetime to any appreciable extent; (c) hydroxyl radical is not an oxidant for everything in the atmosphere. (I know this has become the standard phraseology, but it is simply not true!); (d) Are they really concerned about the atmospheric toxicity of HClO_3 and HClO_4 from this source compared to the anthropogenic source?; and (e) rates and rate coefficients are mixed up.

Reply: Thank you for pointing out. We have revised them accordingly. Below are our replies to the specific comments:

- (a) Agree. We have changed the word “atmosphere” or “troposphere” to “boundary layer” throughout the manuscript in the revised text.
- (b) We do not think that the statement “It is also well established that the direct reaction with Cl provides a chemical sink of methane (CH₄) in the atmosphere” has misled the meaning. Yes, we agree that the dominant chemical sink pathway determining the global CH₄ lifetime in the atmosphere is through the reaction of OH+CH₄. Nevertheless, our current statement in the original text did not mention that the Cl will change the CH₄ lifetime. It is well known that CH₄ is highly reactive to Cl, which has a much faster reaction coefficient compared to the OH reaction with CH₄, and provides an additional sink pathway for CH₄ in the atmosphere, as suggested by previous studies cited in the text (Hossaini *et al.*, 2016; Strode *et al.*, 2020). In addition, a recent study has shown that atmospheric chlorine chemistry can also affect the OH levels thus perturbing the global CH₄ lifetime (Li *et al.* 2022). We have added this citation to the revised main manuscript (Line 66-68).
- (c) The reviewer is correct that OH is not an oxidant for everything in the atmosphere. However, what we really mean in the text is that OH is an important radical in atmospheric photochemistry, and the reaction of OH radical can be a dominant chemical sink for many atmospheric pollutants (including CH₄, VOCs, etc). Therefore, we would prefer to maintain the phrase “the main atmospheric oxidant” in the revised main text (Line 66).
- (d) The reviewer has a point here. Indeed, the chlorate (ClO₃⁻) or perchlorate (ClO₄⁻) in the polar region are much lower than the levels of ClO₄⁻ in the environments with strong anthropogenic influence. The elevated levels of perchlorate (e.g., in mg L⁻¹) in the samples from polluted environments can cause toxicity (e.g., Sijimol *et al.*, 2015), but the actual effects of the perchlorate levels found in the polar region are still unknown. A study hypothesized that the presence of ClO₄⁻ in polar ice may suppress the freezing point of water (Kounaves *et al.*, 2010), while another study reported that the ClO₄⁻ levels measured in the marine sediment of Antarctica could present a toxicity risk to resident biota (Acevedo-Barrios *et al.*, 2022). These studies suggest that the deposition of atmospheric HClO₃ and HClO₄ on the surface can have environmental implications in polar regions. Thus, we have revised the statement to:
(Revised main manuscript, Line 96-98): Therefore, the atmospheric occurrence of chlorine oxyacids could enhance the chlorine sink, thereby affecting the oxidation capacity of the atmosphere and posing environmental threats once deposited to the Earth’s surface.

(Revised main manuscript, Line 281-284): Furthermore, once HClO₃ and HClO₄ deposit on the ground surface (i.e., snowpack and sea-ice), they could cause environmental problems as their ions, ClO₃⁻ and ClO₄⁻, are extremely mobile in water, and may suppress the freezing point of water, and can present a toxicity risk to both flora and fauna⁵⁴⁻⁵⁶.
- (e) We have fixed the mistake in the rates and rate coefficients in the revised text (also refer to the reply to comment 3.9 below).

References:

- Hossaini, R. *et al.* A global model of tropospheric chlorine chemistry: Organic versus inorganic sources and impact on methane oxidation. *J. Geophys. Res. Atmos.* **121**, 14,271-214,297 (2016).
- Strode, S. A. *et al.* Strong sensitivity of the isotopic composition of methane to the plausible range of tropospheric chlorine. *Atmos. Chem. Phys.* **20**, 8405-8419 (2020).
- Li, Q. *et al.* Reactive halogens increase the global methane lifetime and radiative forcing in the 21st century. *Nat. Commun.* **13**, 2768 (2022).
- Sijimol, M. R. *et al.* Review on fate, toxicity, and remediation of perchlorate. *Environ. Forensics* **16**, 125-134 (2015).
- Kounaves, S. P. *et al.* Discovery of natural perchlorate in the Antarctic Dry Valleys and its global implications. *Environ. Sci. Technol.* **44**, 2360-2364 (2010).

- Acevedo-Barrios, R., Rubiano-Labrador, C. & Miranda-Castro, W. Presence of perchlorate in marine sediments from Antarctica during 2017–2020. *Environ. Monit. Assess.* **194**, 102 (2022).

Minor comments:

(3.9) Why are they wasting space and time showing the first-order rate coefficients (I assume these are J values in Figures 6b and 6c)? If there is no absorption in the near UV, nobody would expect a photolytic loss. Again, the values shown in Table S2 are J values and not rates. (rates have units of molecules per cm³ per sec!).

Reply: Sorry that we have caused the confusion here. The purpose of showing the Extended Data Fig. 6 (b and c) is to indicate the slow photolysis (s⁻¹) of HClO₃ and HClO₄ compared to the other loss pathways of HClO₃ and HClO₄, including the loss on the surface and reaction with radicals (OH and Cl). We have changed the text in revised Table S2 to **heterogeneous loss rate coefficient** in the revised supplementary information.

(3.10) Figure 8 shows the signal to noise for the detection. Are the authors using peak heights or peak areas to quantify the concentrations?

Reply: We used the fitted peak areas of the identified peak to quantify the concentration, and the preprocessing method has been described in detail in Jokinen *et al.* (2012). Before processing, the raw data were pre-averaged over 10 min. Briefly, the data were processed with the MATLAB tofTools package (Junninen *et al.*, 2010). Mass calibration was performed using known masses that were persistently present in the spectrum, for instance, nitrate ion and its dimer and trimer (NO₃⁻, HNO₃·NO₃⁻, (HNO₃)₂·NO₃⁻). The peak assignment of halogen-containing species including all their isotopes in the mass spectra was performed within a mass tolerance of about 0.1 ppm, and the fitted area ranged from 99 to 101%. This information has been added to the revised Methods.

(Revised main manuscript, Line 331-332): The raw data was pre-averaged over 10 min and processed with the MATLAB tofTools package according to the procedures described in Jokinen *et al.*⁵⁸.

References:

- Jokinen, T. *et al.* Atmospheric sulphuric acid and neutral cluster measurements using CI-API-TOF. *Atmos. Chem. Phys.* **12**, 4117-4125 (2012).
- Junninen, H. *et al.* A high-resolution mass spectrometer to measure atmospheric ion composition. *Atmos. Meas. Tech.* **3**, 1039-1053 (2010).

(3.11) Use of colors in figures: Yellow, pale green, pale blue, pale pink, etc., may look “pretty” for some, but they are hard to see (especially think of people who are color blind to various extents!).

Reply: Thanks for the suggestion. We have revised the graphs (i.e., *Extended Data Fig. 4, 5, 7, and 8*) to use more colorblind-friendly colours.

(3.12) I have not commented on the modeling and atmospheric analyses since I am not convinced where these chemicals are coming from. If they convince themselves (and us!) that, indeed, it is coming from the chemistry involved in

Bottom line: I am convinced they have detected HClO₃ and HClO₄ in the gas phase. However, is that enough to publish it in this journal? Currently, I do not think that they are. However, the authors can

dissuade me by addressing the points noted above. Then the question become what is the significance of these acids since HCl would still be a major path for the removal of chlorine from these atmospheres.

Reply: Regarding the origin of the atmospheric HClO_3 and HClO_4 , we have discussed the chemistry in the reply to comment (3.2) above.

We believe that the first ambient detection of any species in the real environment (in our case, gas-phase HClO_3 and HClO_4 in the real atmosphere) is an important breakthrough in geosciences. Without the detection, an in-depth understanding and the associated environmental implications of these chemicals will not be possible, even though the formation of these chlorine oxyacids has long been speculated to be present in the atmosphere. Our findings provide new insights into the chlorine chemistry with a newly-found sink pathway for reactive chlorine in troposphere, which certainly has implications for atmospheric chemistry in the Arctic. For instance, HClO_3 and HClO_4 are termination step or irreversible sink of chlorine, at least more irreversible than HCl, which will certainly reduce the levels of reactive chlorine in the Arctic atmosphere, thereby implying a reduction in the chlorine-driven oxidation capacity of the polar boundary layer. One important implication includes that since chlorine is known to play an important role in springtime surface ozone depletion, the discovery of these two new species will affect the role of chlorine radicals in ozone loss and the ozone budget in the Arctic troposphere. Therefore, atmospheric models need now to include the sources and chemistry of these two species for a correct representation of the chlorine cycle, and its associated environmental implications, in the Arctic troposphere.

We agree with the reviewer that the conversion of HCl to Cl^- (aq) has been regarded as a major pathway for the removal of chlorine in the atmosphere. However, recent studies have shown that in the presence of NO_x and other reactive halogens (i.e., HOI, HOBr), Cl^- can be efficiently recycled back into reactive gas-phase chlorine, e.g., ClNO_2 , ICl, etc. (e.g., Finlayson-Pitts *et al.*, 1989; Osthoff *et al.*, 2008; Tham *et al.*, 2021). On the other hand, HClO_3 and HClO_4 are not susceptible to photolysis and radical attack, and their conversion into ClO_3^- and ClO_4^- on aerosols is efficient in the Arctic boundary layer. To our best knowledge, the ClO_3^- and ClO_4^- do not recycle back to the gas-phase directly, thus, the homogeneous formation of HClO_3 and HClO_4 could terminate chlorine recycling in the atmosphere. The inclusion of these chlorine oxyacids in atmospheric models will reduce the chlorine-driven oxidation capacity in the Arctic atmosphere, although more research is needed to better understand the HClO_3 and HClO_4 in future. Overall, we strongly believe that the first detection of two 'new' molecules in the real atmosphere and their associated environmental impacts, as listed above and in the manuscript, is of significance to the wider atmospheric science, climate science, and geoscience communities, and therefore we think this journal is a great venue to communicate these findings.

References:

- Finlayson-Pitts, B. J., Ezell, M. J. & Pitts, J. N. Formation of chemically active chlorine compounds by reactions of atmospheric NaCl particles with gaseous N_2O_5 and ClONO_2 . *Nature* **337**, 241-244 (1989).
- Osthoff, H. D. *et al.* High levels of nitryl chloride in the polluted subtropical marine boundary layer. *Nature Geoscience* **1**, 324-328 (2008).
- Tham, Y. J. *et al.* Direct field evidence of autocatalytic iodine release from atmospheric aerosol. *Proc. Natl. Acad. Sci. U.S.A* **118**, e2009951118 (2021).

Reviewer #1 (Remarks to the Author):

The authors put effort into revising this manuscript describing the detection of HClO₃ and HClO₄ at three Arctic locations, and it has improved as a result. In particular, the manuscript is improved with the addition of the consideration of the role of NO_x, including from ship pollution. However, there still many major concerns about both the observational data and calculations that I describe below.

Regarding Comment 1.6, the authors did not address the following: "The detection limit is particularly important for the HClO₄ signal, which appears to be below this level for most of the Greenland study in Figure 1, and this is not discussed/addressed in the manuscript." When comparing the reported LODs to Figure 1, roughly 1/4 of the HClO₃ data at Greenland, most of the HClO₄ data at Greenland, and roughly 1/3 of the HClO₄ data during MOSAiC appear to be below LOD. Further, measurement uncertainties become even more important near LOD, and given that the "uncertainty of HClO₃ and HClO₄ measurements was estimated to be at least a factor of two", this lack of consideration of LOD is very concerning. It is absolutely critical that the LODs are clearly communicated and discussed in the manuscript. It is misleading to show data as concrete when they are actually below LOD. The estimated LODs need to be shown in each of the Figures showing HClO₃ and HClO₄ to illustrate which data are actually likely above LOD.

Related to the above discussion about the critical discussion of LODs in this study and the fact that this study reports the first atmospheric observations of gas-phase HClO₃ and HClO₄, Fig.R1.4 needs to be included to the manuscript supplemental information.

In response to Comment 1.5, the authors added one "method" sentence about the Br-CI-API-TOF signal shown in Figure 2. This sentence states: "We also detected the Br- signal from the peak at 78.919 m/z, which is most likely attributed to hydrobromic acid (HBr).61" This sentence is useful as it references another study that confirmed, through a laboratory study, that HBr produces Br- (as well as Br-.HNO₃) when reacted with NO₃. However, significant concerns still remain for this data shown in a main text figure: (1) Figure 2 still does not refer to the Br- signal as potentially HBr(g), making it misleading to a reader not familiar with the CI-API-TOF (i.e. a casual reader who might think that this is particulate bromide, since HClO₃ and HClO₄ are not shown as ions in this figure). (2) The authors did not address whether the CI-API-TOF also measured Br-.HNO₃, which would further justify the measurement of HBr. (3) The authors state that "without proper information on the calibration factor, zeroing measurement, etc., we are not able to determine its uncertainty, LODs, etc.". Without any knowledge of the LOD or error, it is inappropriate to present these data, as much of the data shown could be below LOD without our knowledge. Further, no acknowledgement of this major uncertainty is provided in Figure 2 or the methods, making showing these data misleading. (4) In their response, the authors justify showing the raw, uncalibrated Br- signal on the basis of supporting active bromine chemistry, but it is important to note that HBr can be produced simply from the reaction of HNO₃ with NaBr, for example. For MOSAiC, some of the coauthors of this paper already published BrO data, which would be more useful and conclusively show active reactive bromine chemistry.

The discussion about bromine chemistry is critical because a primary result of the study is stated in the abstract as: "The increase in HClO₃, concomitantly with that in HClO₄, was linked to the increase in bromine levels during ozone depletion events. These observations indicated that bromine chemistry enhances the formation of OClO, which is subsequently oxidized into HClO₃ and HClO₄ by hydroxyl radicals." Yet, in Figure 2, there is no clear relationship between the Br- signal and O₃. Lines 118-119 state "a significant increase in HClO₃ concentration was observed during ozone depletion events, as shown in Fig 1a", and this can be discerned. Extended Data Figure 5 (and response to Comment 1.8) stated that there are significant differences between the concentrations of HClO₄ under different ranges of O₃, and it does appear that higher mean concentrations (but still perhaps within a factor of 2?) were observed at 5-20 ppb O₃

compared to the other concentrations. However, stating that higher levels of HClO₃ and HClO₄ were observed at 5-20 ppb O₃ is a different statement than that in the abstract. Further, Extended Data Fig 5 oddly no longer includes HClO₃, which is critical to quantitatively evaluate this result.

In response to Comment 1.10, the authors updated the [Cl] used in their calculations/modeling to reflect that reported by Custard et al., which is excellent. However, they refer in the text to this value as: "typical Cl (1 x 10⁶ molecules cm⁻³)". However, the daytime maxima Cl concentrations reported by Custard et al. ranged from ~0.2 to 1.4 x 10⁶ molecules cm⁻³, with a daytime average that is much lower than 1 x 10⁶ molecules cm⁻³. Further, when O₃ was depleted (a focus of this paper), Custard et al. found the lowest [Cl], making a value of 0.4 x 10⁶ molecules cm⁻³ (60% lower than their current value) more appropriate to use for modeling typical ozone depleted conditions. Cl is key to the calculations in the manuscript, so it is critical that an appropriate value is chosen.

Related to the above comment the authors state on Lines 183-185: "It is unambiguous that abundant BrO and ClO (tens of pptv) must have been present during the encountered ozone depletion events..." Since neither BrO or ClO data are reported in this manuscript, it is not appropriate and misleading to use the phrasing "unambiguous" and refer to "tens of pptv". Further, Custard et al. (cited in the authors' sentence) reported that "when O₃ was partially depleted (<10 ppb)..., the lowest daytime ClO values were measured at less than 5 ppt." Therefore, the authors' statement in the main text is also inaccurate.

Related to the above comment, this misinterpretation of prior ClO observations appears to propagate into the calculations/modeling as well, as the authors state on Lines 187-188: "By using the previously reported typical ranges of BrO, ClO, and HO₂ levels during Arctic ozone-depletion events, we estimate...". Then Lines 80-81 of the Supplemental Information refer to a ClO level of 40 ppt (a factor of 8 higher than appropriate) being used in the modeling. Similar to the above comment, ClO is key to the calculations in the manuscript, so it is critical that an appropriate value is chosen.

In responses to Comments 2.2 and 3.2, the authors discuss heterogeneous uptake of HClO₃ and HClO₄ to snow as important, and this is included in the caption of Figure 3. Yet, Line 56 of the abstract still only refers to uptake on aerosols and needs to be revised to also include snow.

Additional Comments:

- Fig.R1.2 is useful, and I recommend adding it to the manuscript supplemental information.

- In response to Comment 1.6, the authors state "We have therefore revised the LOD values to one significant number", but lines 368 and 369 still show two significant figures for each of the LODs.

- The authors chose not to include error bars in their HClO₃ & HClO₄ figures, but they now at least address the uncertainty of "at least a factor of two" in most of the figure captions. Please add this to the Fig 2 caption as well for clarity.

- I appreciate that the authors did not include data below LOD in the revised Extended Data Fig 5, that statistical analysis was performed, and that the O₃ ranges were revised to allow assessment of ODE conditions.

- In response to Comment 1.9, the authors stated that references were added to the sentences noted. However, Lines 71-78 still do not include references. I did not check all of the previously referred to sentences because the authors did not provide line numbers.

- An LOD is not reported for the HClO₃ observations at Ny-Alesund (Extended Data Fig. 2).

Reviewer #2 (Remarks to the Author):

The detection of chlorine oxyacids and the discussion of their impact on Arctic oxidation chemistry is a key issue of broad interest to the environmental chemistry community. The study convincingly presents very low concentrations – a significant achievement. The detection of the oxyacids appears sound to me, and I am impressed that data from 3 different sites have been included. Further, to assess the environmental importance, UV absorption and photolysis are estimated and discussed for the first time. This makes the findings attractive to a broader audience in the environmental science community.

Although there have been significant additions to the manuscript, some major doubts remain concerning the arguments made surrounding the conclusions.

- 1). Flux estimates and some indication of mixing and boundary layer height need to be discussed when dealing with potential surface sources of trace gases.
- 2). With all due respect, the quantitative argument for the surface release of HClO₃ and HClO₄, based on the Henry constant, is wrong. The SI states that one needs 35 nM HClO₄ in the liquid phase to maintain Henry's equilibrium. This cannot be compared to the levels found in ice cores, as these are in reference to the total ice volume, and the Henry coefficient does not give the ice-air equilibrium. Secondly, the amount of perchlorate in snow/ice might be present at high concentrations in aerosol deposits or liquid frequently found in Arctic and Antarctic snow and ice, which could establish the Henry equilibrium.
- 3). I can't follow the argument that snow is irrelevant because perchlorates are also found in summer (line 170). Firstly, it is unclear where the precursors come from; the authors give snowpack emission as a source (line 180). Secondly, I find it surprising that the authors assume one source and production route dominating the entire data set. There could be different sources in winter and summer – again, looking at the boundary layer height and estimating emissions might be helpful.

Minor

Line 283, Can the authors please give more details on the freezing point depression – how many degrees would you expect based on the nM concentrations? Is that significant relative to the larger concentration of sea salt?

The same holds for toxicity. Why exactly are those low concentrations toxic?

The authors may comment on why no atmospheric chemistry model was used to discuss the importance of the many sources and sink processes. I'm kind of surprised that this has not been done.

Reviewer #3 (Remarks to the Author):

Overall, the authors have addressed my major technical questions. I have a few minor comments below. I am still on the fence about importance of this finding to the environment. I would have been happier if the authors note emphasize their detection of these highly oxidized chlorine species, and the likelihood that they are produced in the gas phase through a set of complex reactions. That is a sufficient advance to publish this paper. But, the authors have opted to emphasize the atmospheric significance.

I am OK with the way it is now.

A couple minor comments:

1. It is not true that HClO₃ and HClO₄ cannot travel long distances in the atmosphere. I would buy it for near surface transport, but they can be transported on/in particles and also via strat-trop exchange especially in the polar region where there is a massive

descent from the stratosphere. I suggest that they be clear about their assertion that it is not transported in the boundary layer....

2. I would have toned down the statement that their measurements are a lower limit for the HClO₃ and HClO₄ concentration. This is not logical. They have not calibrated directly, they are assuming a Langevin (association) reaction rate coefficient (which begs the question of the capture cross section), and there are mass effects in detection. I think that they can claim that their numbers are within a factor of two or three of the real value.

3. I don't understand their assertion in the rebuttal that the aerosol composition would not affect their analysis. What if the aerosols are very acidic? They did not justify why they "believe" this assertion to be true.

4. I don't understand why they insist that OH is the main atmospheric oxidant for hydrocarbons etc. Even for olefins, their reaction with Ozone can be a major pathway. Lastly, OH is a daytime species (for the most part) and so it is not the main oxidant under darkness and other conditions.

5. I don't understand what lines 281-284 mean. What does the phrase "ions, ClO₃⁻ and ClO₄⁻, are extremely mobile in water? Their diffusion coefficients are large? What does "mobile" mean here.

6. Extended Data Fig. 9: Don't they mean $\Delta G = \Delta G(\text{zero}) + RT \ln Q$, where Q is the reaction coefficient.

We thank the three reviewers for their additional constructive suggestions in this round of review. Our point-by-point responses are listed in dark blue wording and the corresponding changes made to the revised manuscript are in light blue. The changes in the revised manuscript (track-change version) have been highlighted in light blue.

Reviewer #1 (Remarks to the Author):

1.1. The authors put effort into revising this manuscript describing the detection of HClO_3 and HClO_4 at three Arctic locations, and it has improved as a result. In particular, the manuscript is improved with the addition of the consideration of the role of NO_x , including from ship pollution. However, there still many major concerns about both the observational data and calculations that I describe below.

Reply: We thank the reviewer for the detailed and insightful comments, which have helped us to improve the manuscript.

1.2. Regarding Comment 1.6, the authors did not address the following: “The detection limit is particularly important for the HClO_4 signal, which appears to be below this level for most of the Greenland study in Figure 1, and this is not discussed/addressed in the manuscript.” When comparing the reported LODs to Figure 1, roughly 1/4 of the HClO_3 data at Greenland, most of the HClO_4 data at Greenland, and roughly 1/3 of the HClO_4 data during MOSAiC appear to be below LOD. Further, measurement uncertainties become even more important near LOD, and given that the “uncertainty of HClO_3 and HClO_4 measurements was estimated to be at least a factor of two”, this lack of consideration of LOD is very concerning. It is absolutely critical that the LODs are clearly communicated and discussed in the manuscript. It is misleading to show data as concrete when they are actually below LOD. The estimated LODs need to be shown in each of the Figures showing HClO_3 and HClO_4 to illustrate which data are actually likely above LOD.

Reply: From our mass spectrum (Fig. R1.1 below or refer to Extended Data Fig. 8 of the main text), it is clear that there are absences of peaks for HClO_3 and HClO_4 during the zero measurement (also see the newly added Fig. S3 in the revised supplementary information), suggesting that the detection of the peaks for HClO_3 and HClO_4 are real signals. With these zero measurements, we have established the limit of detection (LOD) for HClO_3 and HClO_4 . We agree with the reviewer that the discussion of LODs are very important, especially for the data points near the LOD, given that our measurement uncertainty was estimated to be at least a factor of two. Therefore, based on the reviewer’s suggestions, we have marked the LODs of HClO_3 and HClO_4 with a dashed-line in each figure (Fig. 1, Fig. 2, Extended Data Fig. 1, Extended Data Fig. 2, Extended Data Fig. 3, and Extended Data Fig. 4) in the revised manuscript to clearly distinguish the measurement data points above the detection limits.

As the HClO_4 concentrations are relatively low, communicating the information of LOD is critical. We, therefore, have revised the statement in the main text to include this information.

(Revised main manuscript, Line 137-140): The HClO_4 concentrations in Greenland and MOSAiC were estimated to be in the range of near detection limits (7×10^3) to 8×10^4 molecules cm^{-3} and near detection limits (3×10^4) to 1×10^6 molecules cm^{-3} , respectively, during springtime, which were typically lower than the HClO_3 concentration.

Fig. R1.1 | Peak identification for HClO₃ and HClO₄. The selected mass spectra reveal the corresponding peaks (black line) and mass spectrum of the zero measurements (red line) by the nitrate CI-API-TOF for (a) deprotonated ions of HClO₃ and ClO₃⁻, (b) deprotonated ions of HClO₄ and ClO₄⁻.

Fig. 1 | HClO₃ and HClO₄ over the Arctic. Time series of HClO₃, HClO₄ and O₃, together with the temperature and incoming solar radiation measured at (a) the Villum Research Station from 1 March–15 May 2015 and (b) during the MOSAiC expedition from 22 February–30 April 2020. The data are displayed at 30-min average resolution, and any gaps in the time series are the results from instrumentation offline and maintenance periods. The dashed-line represents the detection limits for HClO₃ (blue) and HClO₄ (pink) measurements. The uncertainty of HClO₃ and HClO₄ measurements was estimated to be at least a factor of two (see Methods). The map shows the location of the Villum Research Station (Nord) in Greenland, Ny-Ålesund in Svalbard, and RV Polarstern passive drifting track across the Arctic Ocean during the springtime sampling period. Note that all the time reported here is in Coordinated Universal Time (UTC).

Fig. 2 | Relationships between HClO_3 , HClO_4 , O_3 and bromine chemistry. Expanded view of HClO_3 (blue solid line) tracking with the HClO_4 (pink solid line) and HBr (grey shaded-area; based on the Br^- normalized signal from nitrate CI-API-TOF measurements which is most likely HBr (refer to Methods) to represent bromine chemistry), at (a) the Villum Research Station, Greenland, from 19 to 29 March 2015; and (b) onboard RV Polarstern during the MOSAiC campaign, from 15 to 25 March 2020. The dashed-line represents the detection limits for HClO_3 (blue) and HClO_4 (pink) measurements. The uncertainty of HClO_3 and HClO_4 measurements was estimated to be at least a factor of two.

Extended Data Fig. 1 | HClO_3 and HClO_4 measurements during the MOSAiC campaign. The variation in the HClO_3 (blue) and HClO_4 (red) concentrations observed from January to June 2020 during the MOSAiC campaign. This period covers the winter, spring and summer in the Arctic. The dashed-line represents the detection limits for HClO_3 (blue) and HClO_4 (pink) measurements. The uncertainty of HClO_3 and HClO_4 measurements was estimated to be at least a factor of two (see Methods).

Extended Data Fig. 2 | Observation of HClO₃ at Ny-Ålesund. Example of the HClO₃ concentration data measured with a nitrate-Cl-API-TOF instrument at Ny-Ålesund, Svalbard (78° 55' N, 11° 56' E), from 28 March to 30 May 2017 (spring). The measurements occurred at the atmospheric observatory, Gruvebadet, located 2 km southeast of Ny-Ålesund (refer to the map in Fig. 1). The dashed-line represents the detection limit for HClO₃ measurement. The uncertainty of HClO₃ and HClO₄ measurements was estimated to be at least a factor of two (see Methods). Details on the site (description), instrumentation setup and calibration can be found in Beck *et al.*⁴⁴.

Extended Data Fig. 3 | Air mass trajectories during non-O₃ depletion and O₃ depletion events. Example cases (area shaded in light green) revealing the difference between the air mass origin and height during ozone depletion events and non-ozone depletion events at the (a) Villum Research Station, Greenland, and during the (b) MOSAiC campaign. The 48-hour backward air mass trajectory and altitude are calculated with the Hybrid Single-Particle Lagrangian Integrated Trajectory (HYSPPLIT) model, developed by the NOAA Air Resources Laboratory⁸⁰. The starting height of the trajectory was set at 50 m above ground level.

Extended Data Fig. 4 | HClO₃, HClO₄, H₂SO₄ and aerosol surface area. Expanded view between HClO₃, HClO₄ and the measured H₂SO₄ and aerosol surface area at (a) the Villum Research Station, Greenland (18 March to 17 April 2015) and (b) during the MOSAiC campaign (1 March to 28 April 2020). The gap in the aerosol surface area in the data set is either due to the removal of invalid data (i.e., particle counts too low or due to local emissions), instrument maintenance or instrumentation offline events. The dashed-line represents the detection limits for HClO₃ (blue) and HClO₄ (pink) measurements. The uncertainty of HClO₃ and HClO₄ measurements was estimated to be at least a factor of two (see Methods).

1.3. Related to the above discussion about the critical discussion of LODs in this study and the fact that this study reports the first atmospheric observations of gas-phase HClO₃ and HClO₄, Fig.R1.4 needs to be included to the manuscript supplemental information.

Reply. We thank the reviewer for this suggestion. The referred figure (below) has now been added to the revised supplementary information.

Fig. S2 | Zero measurements in Greenland. Example of zero measurements in Greenland conducted on (a) 31 March 2015 and (b) 8 May 2015. The shaded area (light green) is to the zero measurement period. The ClO_3^- , ClO_4^- and HSO_4^- signals correspond to HClO_3 , HClO_4 and H_2SO_4 , respectively.

1.4. In response to Comment 1.5, the authors added one “method” sentence about the Br^- CI-APi-TOF signal shown in Figure 2. This sentence states: “We also detected the Br^- signal from the peak at 78.919 m/z , which is most likely attributed to hydrobromic acid (HBr).⁶¹” This sentence is useful as it references another study that confirmed, through a laboratory study, that HBr produces Br^- (as well as $\text{Br}\cdot\text{HNO}_3$) when reacted with NO_3^- . However, significant concerns still remain for this data shown in a main text figure: (1) Figure 2 still does not refer to the Br^- signal as potentially HBr (g), making it misleading to a reader not familiar with the CI-APi-TOF (i.e. a casual reader who might think that this is particulate bromide, since HClO_3 and HClO_4 are not shown as ions in this figure). (2) The authors did not address whether the CI-APi-TOF also measured $\text{Br}\cdot\text{HNO}_3$, which would further justify the measurement of HBr . (3) The authors state that “without proper information on the calibration factor, zeroing measurement, etc., we are not able to determine its uncertainty, LODs, etc”. Without any knowledge of the LOD or error, it is inappropriate to present these data, as much of the data shown could be below LOD without our knowledge. Further, no acknowledgement of this major uncertainty is provided in Figure 2 or the methods, making showing these data misleading. (4) In their response, the authors justify showing the raw, uncalibrated Br^- signal on the basis of supporting active bromine chemistry, but it is important to note that HBr can be produced simply from the reaction of HNO_3 with NaBr , for example. For MOSAiC, some of the coauthors of this paper already published BrO data, which would be more useful and conclusively show active reactive bromine chemistry.

Reply: We thank the reviewer for these constructive suggestions. We have made the following changes:

- (1) According to our previous study, the Br^- detected in our instrument is most likely corresponding to the deprotonation of HBr (Fan *et al.*, 2021); therefore, we agree to change the Br^- in the Fig. 2 to HBr to avoid misleading. We have also made a note in the figure captions that the HBr is correspond to the Br^- signal from our instrument (see the caption of revised Fig. 2 in reply to comment 1.2 above).
- (2) A very good suggestion. We have checked the nitrate CI-APi-TOF raw data and found that we can see a clear $\text{NO}_3(\text{H}^{79}\text{Br})^-$ peak together with its isotope peak ($\text{NO}_3(\text{H}^{81}\text{Br})^-$), as shown in Fig. R1.2a. The very good correlation ($R^2 = 0.97$) between $\text{NO}_3(\text{HBr})^-$ and Br^- signal (Fig. R1.2b) further justify our observation that the Br^- is more likely from the HBr . We have add the following statement to the revised methods. (Revised main manuscript, Line 332-334): We also detected the Br^- signal from the peak at 78.919 m/z (together with $\text{NO}_3(\text{HBr})^-$ peak at 141.915 m/z), which is most likely attributed to hydrobromic acid (HBr)⁶³.
- (3) We want to emphasize that the purpose of using the Br^- signal (in arbitrary units) was to show that active bromine chemistry exists (regardless of the sources of the bromine) during our observation, and qualitatively supporting our hypothesis of the involvement of bromine chemistry in the formation of the

measured chlorine oxyacids. Of course, we agree that it would be better if knowledge on the LOD is known, but the actual quantification of Br^- signal is not the major message of this work. Furthermore, the HBr (Br^- signal) is the only high-resolution full data set that are available in our studies that can represent the bromine chemistry, although the MOSAiC campaign also measured the BrO (see reply number (4) below). It is quite a 'common' practice for mass spectrometer users to report data in arbitrary unit for qualitative analysis (e.g., Bianchi *et al.*, 2016; Wang *et al.*, 2020; Benavent *et al.*, 2022; Wan *et al.*, 2022).

To show that the variation of HBr (Br^- signal) is actually above the detection, we have compared the observed HBr signal to the zero measurements. From Fig. S1, the HBr (Br^- signal) is higher than the background (zero) signal during the spring, and they are close to the background (zero) signal when the bromine chemistry is less active (i.e., June). This information and Fig. S1 have been added to the revised supplementary information.

- (4) We acknowledge the concern raised by the reviewer that HBr can be produced from the reaction of HNO_3 with NaBr. Yet, this reaction produce HBr that is a key species of the atmospheric bromine chemistry. The HBr (Br^- signal) data during the MOSAiC campaign has been published and compared with the BrO measurement (Benavent *et al.*, 2022). From Fig. S1, the ground-based BrO observations agree with our HBr (Br^- signal) with higher values observed during the spring, but no signal detected during the summer period. Since the ground-based BrO data is only available in MOSAiC starting from late March, we have decided to add the MOSAiC BrO dataset and to retain the HBr (Br^- signal) as the basis for supporting the active bromine chemistry. The Fig. S1 has been added to the revised supplementary information to further support our results. These two bromine observations strongly support that bromine chemistry was active in coincidence with the formation of HClO_3 and HClO_4 .

Fig. R1.2 | Observation of $\text{NO}_3(\text{HBr})^-$ signal. (a) The peak of $\text{NO}_3(\text{H}^{79}\text{Br})^-$ together with its isotope peak ($\text{NO}_3(\text{H}^{81}\text{Br})^-$), compared to the zero measurement (red line), detected by nitrate CI-API-TOF. (b) An example of the $\text{NO}_3(\text{HBr})^-$ versus Br^- data obtained during the MOSAiC from 18 to 30 April 2020.

Fig. S1 | HClO₃, HClO₄ and bromine observations during MOSAiC. The time series for ground-based BrO, HBr (Br⁻ signal), corresponding to the HClO₃ and HClO₄ observations during the MOSAiC campaign. Details of the ground-based BrO measurement can be found in Benavent *et al.* (2022). The dashed-line represents the detection limits (LOD) for HClO₃ (blue) and HClO₄ (pink). The grey dashed-line is the background signal obtained during the zero measurements of HBr (Br⁻ signal). Note that the uncertainty of HClO₃ and HClO₄ measurements was estimated to be at least a factor of two.

References:

- Fan, X. *et al.* Atmospheric gaseous hydrochloric and hydrobromic acid in urban Beijing, China: detection, source identification and potential atmospheric impacts. *Atmos. Chem. Phys.* **21**, 11437-11452, (2021).
- Bianchi, F. *et al.* New particle formation in the free troposphere: A question of chemistry and timing. **352**, 1109-1112 (2016).
- Wang, Y. *et al.* Formation of highly oxygenated organic molecules from chlorine-atom-initiated oxidation of alpha-pinene. *Atmos. Chem. Phys.* **20**, 5145-5155 (2020).
- Benavent, N. *et al.* Substantial contribution of iodine to Arctic ozone destruction. *Nat. Geosci.* **15**, 770-773 (2022).
- Wan, Y. *et al.* Chemical characterization of organic compounds involved in iodine-initiated new particle formation from coastal macroalgal emission. *Atmos. Chem. Phys.* **22**, 15413-15423 (2022).

1.5. The discussion about bromine chemistry is critical because a primary result of the study is stated in the abstract as: “The increase in HClO₃, concomitantly with that in HClO₄, was linked to the increase in bromine levels during ozone depletion events. These observations indicated that bromine chemistry enhances the formation of OClO, which is subsequently oxidized into HClO₃ and HClO₄ by hydroxyl radicals.” Yet, in Figure 2, there is no clear relationship between the Br⁻ signal and O₃. Lines 118-119 state “a significant increase in HClO₃ concentration was observed during ozone depletion events, as shown in Fig 1a”, and this can be discerned. Extended Data Figure 5 (and response to Comment 1.8) stated that there are significant differences between the concentrations of HClO₄ under different ranges of O₃, and it does appear that higher mean concentrations (but still perhaps within a factor of 2?) were observed at 5-20 ppb O₃ compared to the other concentrations. However, stating that higher levels of HClO₃ and HClO₄ were observed at 5-20 ppb O₃ is a different statement than that in the abstract. Further, Extended Data Fig 5 oddly no longer includes HClO₃, which is critical to quantitatively evaluate this result.

Reply: We thank the reviewer for this comment. We have discussed the justification of using HBr (Br⁻ signal) to indicate active bromine chemistry in the reply to comment 1.4 above. Here, we would like to clarify that the

observation of HClO_3 and HClO_4 peaks coincide with the lower levels of O_3 or when the O_3 is depleted (as shown in Fig. 1). In general, Fig. 2 and Fig S1 show that the general trend of HBr (Br^- signal) follows the HClO_3 and HClO_4 , and the decrease of O_3 levels. The lack of good relationship between the bromine and O_3 in certain observation period can be expected as similar phenomenon, which maybe attribute to the freshly emission of halogen or rapid entrainment of O_3 from aloft, has been reported in previous polar studies (Saiz-Lopez *et al.*, 2007; Simpson *et al.*, 2007). To avoid confusion, we have decided to revise the statement in the abstract and main text to the followings:

(Revised abstract, Line 55-56): The increase in HClO_3 , concomitantly with that in HClO_4 , was linked to the increase in bromine levels.

(Revised main manuscript, Line 120-122): HClO_3 exhibited no diurnal pattern, but a unique feature is that a significant increase in HClO_3 concentration was observed in coincidence with the depletion of O_3 , as shown in Fig. 1a.

(Revised main manuscript, Line 128-129): An increase in HClO_3 was also observed in coincidence with the depletion of O_3 over the Arctic Ocean during the MOSAiC campaign.

As suggested by the reviewer, we have included the plot of HClO_3 versus O_3 to the revised Extended Data Fig. 5. Similar to the discussion in the text, higher HClO_3 concentration was observed when the O_3 level was lower than 30 ppb. We have also revised the figure caption to note that relatively higher levels of HClO_3 and HClO_4 were observed at O_3 level below 30 ppb.

Extended Data Fig. 5 | HClO_4 and HClO_3 under different O_3 levels. Distribution of HClO_4 and HClO_3 under a different range of O_3 concentrations at (a, b) the Villum Research Station, Greenland; and (c, d) during the MOSAiC expedition from 22 February to 30 April 2020. Significant concentrations of HClO_3 and HClO_4 were frequently observed when the O_3 concentration was below 30 ppb (also refer to Fig.1 and Fig.2). Note that when O_3 is less than 5 ppb, we defined it as 'complete' depletion of O_3 . In this plot, all the data points below the detection limits were removed. The uncertainty of a factor of two for the HClO_4 measurements has been shown in the plot (grey shaded-area).

References:

- Saiz-Lopez, A. *et al.* Boundary layer halogens in coastal Antarctica. *Science* **317**, 348-351 (2007).
- Simpson, W. R. *et al.* Halogens and their role in polar boundary-layer ozone depletion. *Atmos. Chem. Phys.* **7**, 4375-4418 (2007).

1.6. In response to Comment 1.10, the authors updated the [Cl] used in their calculations/modeling to reflect that reported by Custard *et al.*, which is excellent. However, they refer in the text to this value as: “typical Cl (1×10^6 molecules cm^{-3})”. However, the daytime maxima Cl concentrations reported by Custard *et al.* ranged from ~ 0.2 to 1.4×10^6 molecules cm^{-3} , with an daytime average that is much lower than 1×10^6 molecules cm^{-3} . Further, when O_3 was depleted (a focus of this paper), Custard *et al.* found the lowest [Cl], making a value of 0.4×10^6 molecules cm^{-3} (60% lower than their current value) more appropriate to use for modeling typical ozone depleted conditions. Cl is key to the calculations in the manuscript, so it is critical that an appropriate value is chosen.

Reply: We follow the suggestion made by the reviewer to change the Cl concentration to 4×10^5 molecules cm^{-3} , according to those reported by Custard *et al.* (2016), to calculate the reactivity of Cl to HClO_4 in the revised manuscript.

(Revised main manuscript, Line 244-246): Assuming typical Cl (4×10^5 molecules cm^{-3})³ and OH (5×10^5 molecules cm^{-3})⁴ concentrations in the Arctic, the loss rates of $k_{\text{Cl}+\text{HClO}_4}[\text{Cl}]$ and $k_{\text{OH}+\text{HClO}_4}[\text{OH}]$ are estimated to be 4.0×10^{-26} and $2.9 \times 10^{-7} \text{ s}^{-1}$, respectively.

As for the Cl concentrations used for calculating the consumption rate of OClO by Cl in the revised supplementary information (see section S2), we have used the concentration range of $\approx 10^3$ – 10^6 molecules cm^{-3} , which were reported by previous studies during the spring in Arctic (including those values reported by Custard *et al.*). We think this is an appropriate range, therefore, we decide to retain this range for the calculation.

References:

- Custard, K. D., Pratt, K. A., Wang, S. & Shepson, P. B. Constraints on Arctic atmospheric chlorine production through measurements and simulations of Cl_2 and ClO . *Environ. Sci. Technol.* **50**, 12394-12400 (2016)

1.7. Related to the above comment the authors state on Lines 183-185: “It is unambiguous that abundant BrO and ClO (tens of pptv) must have been present during the encountered ozone depletion events...” Since neither BrO or ClO data are reported in this manuscript, it is not appropriate and misleading to use the phrasing “unambiguous” and refer to “tens of pptv”. Further, Custard *et al.* (cited in the authors’ sentence) reported that “when O_3 was partially depleted (<10 ppb)..., the lowest daytime ClO values were measured at less than 5 ppt.” Therefore, the authors’ statement in the main text is also inaccurate.

Reply: We thank the reviewer for pointing this out. We have removed the ‘unambiguous’ and ‘tens of pptv’ from the sentence.

(Revised main manuscript, Line 187-189): Abundant BrO and ClO must have been present during the encountered ozone depletion events, as have been previously demonstrated by many studies^{2,3,31-37}, and significant levels of BrO have been observed in spring during the MOSAiC campaign³⁸.

1.8. Related to the above comment, this misinterpretation of prior ClO observations appears to propagate into the calculations/modeling as well, as the authors state on Lines 187-188: “By using the previously reported typical ranges of BrO , ClO , and HO_2 levels during Arctic ozone-depletion events, we estimate...”. Then Lines 80-81 of the Supplemental Information refer to a ClO level of 40 ppt (a factor of 8 higher than appropriate) being used in the modeling. Similar to the above comment, ClO is key to the calculations in the manuscript, so it is critical that an appropriate value is chosen.

Reply: We want to clarify that the range of BrO, ClO, and HO₂ concentrations used for calculation were chosen based on the concentrations that have been reported in several previous studies over the Arctic instead of using a single reported concentration value (see Fig S3 in the revised supplementary information). We believe that the range of concentrations will be closer to the real atmospheric condition since we do not have concurrent measurements for these compounds (except for BrO; see Fig S1 in the revised supplementary information). Regarding the ClO value, we have used a range of 5–40 ppt for the calculation. From Custard *et al.* (2016) study, the daytime maxima ClO reached approximately 5 ppt when the O₃ is below 10 ppb, and can reach approximately 25 ppt when the O₃ is below 20 ppb. Another study in Ny-Ålesund has reported ClO concentration up to 40 ppt during the springtime O₃ depletion (Tuckermann *et al.*, 1997). Therefore, we think this range is appropriate for the calculation. To avoid further misinterpretation, we have revised the statement on the ClO concentration in Section S1 of the revised supplementary information as below.

(Revised supplementary information, Line 68-70): The daytime maximum ClO concentration was reported to be in the range of $1.5\text{--}11.6 \times 10^8$ molecules cm⁻³ (5–40 ppt) in previous field observations in the Arctic springtime^{5,8,9}.

References:

- Custard, K. D., Pratt, K. A., Wang, S. & Shepson, P. B. Constraints on Arctic atmospheric chlorine production through measurements and simulations of Cl₂ and ClO. *Environ. Sci. Technol.* **50**, 12394–12400 (2016).
- Tuckermann, M. *et al.* DOAS-observation of halogen radical-catalysed arctic boundary layer ozone destruction during the ARCTOC-campaigns 1995 and 1996 in Ny-Ålesund, Spitsbergen. *Tellus B: Chem. Phys. Meteorol.* **49**, 533–555 (1997).

1.9. In responses to Comments 2.2 and 3.2, the authors discuss heterogeneous uptake of HClO₃ and HClO₄ to snow as important, and this is included in the caption of Figure 3. Yet, Line 56 of the abstract still only refers to uptake on aerosols and needs to be revised to also include snow.

Reply: Thanks. This information has now been included in the abstract.

(Revised abstract, Line 58-60): HClO₃ and HClO₄ are not photoactive and therefore their loss through heterogeneous uptake on aerosol and snow surfaces can function as a previously missing atmospheric sink for reactive chlorine, thereby reducing the chlorine-driven oxidation capacity in the Arctic boundary layer.

Additional Comments:

1.10. Fig.R1.2 is useful, and I recommend adding it to the manuscript supplemental information.

Reply: Agree. The Fig R1.2 (in previous reply) has been added to the revised supplementary information as Fig. S4.

Fig. S4 | Expanded view of HClO₃ and HClO₄ levels during the measurement period with potential influence of ship pollution in the MOSAiC campaign. The grey shaded area (with a time resolution of 1 min) denotes the period where the particle number concentration measurements were impacted by the ship pollution, as defined in Beck *et al.*³⁵. This figure is indicative only and does not necessarily reflect the percentage of clean data points collected by other instruments during the expedition. There is, however, a relatively high probability that NO_x levels were elevated during these periods. The dashed-line represents the detection limits for HClO₃ (blue) and HClO₄ (pink) measurements. Note that the uncertainty of HClO₃ and HClO₄ measurements was estimated to be at least a factor of two.

1.11. In response to Comment 1.6, the authors state “We have therefore revised the LOD values to one significant number”, but lines 368 and 369 still show two significant figures for each of the LODs.

Reply: Revised.

(Revised main manuscript, Line 371-374): Based on these calibration factors, the detection limits of HClO₃ and HClO₄ were estimated as 3×10^3 and 7×10^3 molecules cm⁻³ (10 min-average, 3σ), respectively, during the Greenland measurement campaign and 2×10^4 and 3×10^4 molecules cm⁻³ (10 min-average, 3σ), respectively, during the MOSAiC measurement campaign.

1.12. The authors chose not to include error bars in their HClO₃ & HClO₄ figures, but they now at least address the uncertainty of “at least a factor of two” in most of the figure captions. Please add this to the Fig 2 caption as well for clarity.

Reply: This has been added following the suggestion of the reviewer.

(Revised caption of Fig. 2, Line 664-671): **Fig. 2 | Relationships between HClO₃, HClO₄, O₃ and bromine chemistry.** Expanded view of HClO₃ (blue solid line) tracking with the HClO₄ (pink solid line) and HBr (grey shaded-area; based on the Br⁻ normalized signal from nitrate CI-API-TOF measurements which is most likely HBr (refer to Methods) to represent bromine chemistry), at (a) the Villum Research Station, Greenland, from 19 to 29 March 2015; and (b) onboard RV Polarstern during the MOSAiC campaign, from 15 to 25 March 2020. The dashed-line represents the detection limits for HClO₃ (blue) and HClO₄ (pink) measurements. The uncertainty of HClO₃ and HClO₄ measurements was estimated to be at least a factor of two.

1.13. I appreciate that the authors did not include data below LOD in the revised Extended Data Fig 5, that statistical analysis was performed, and that the O₃ ranges were revised to allow assessment of ODE conditions.

Reply: Thank you for the comment.

1.14. In response to Comment 1.9, the authors stated that references were added to the sentences noted. However, Lines 71-78 still do not include references. I did not check all of the previously referred to sentences because the authors did not provide line numbers.

Reply: The references have been cited.

(Revised main manuscript, Line 74-81): Upon photolysis, these chlorine species release reactive Cl atoms (R1 and R2), which rapidly react with O₃ to form chlorine monoxide, ClO (R3)³. ClO is subsequently oxidized by bromine monoxide (BrO) or ClO, producing chlorine dioxide (ClO₂); reacting with HO₂ to form hypochlorous acid (HOCl); or reacting with nitric oxide (NO) and nitrogen dioxide (NO₂) to produce Cl atoms and chlorine nitrate (ClONO₂) (as shown in reactions R4–R7)¹². Cl atoms can also degrade other hydrocarbons (RH) to generate hydrochloric acid (HCl; R8)¹³. HCl can be converted into chloride (Cl⁻) via hydrolysis on aerosol surfaces (R9)¹⁴. Chloride can further undergo heterogeneous reaction with HOCl to produce Cl₂ (R10), which can in turn be photolyzed to recycle Cl atoms (R1)¹⁵.

Examples of other referenced sentences that have been included previously are:

(Revised main manuscript, Line 151-152): Although the initial steps of atmospheric chlorine oxidation are well understood (R1–R10)^{13,14}, the final oxidation steps leading to chlorine oxyacid formation are not well characterized.

(Revised main manuscript, Line 167-169): This assumption is justified by the remarkably low vapour pressure (6.8 mm Hg under a 70% concentration, at 298 K)²⁸ and high Henry's law constant (K_H) of HClO₄ ($9.9 \times 10^3 \text{ mol m}^{-3} \text{ Pa}^{-1}$)¹⁸.

(Revised main manuscript, Line 244-245): Assuming typical Cl ($4 \times 10^5 \text{ molecules cm}^{-3}$)³ and OH ($5 \times 10^5 \text{ molecules cm}^{-3}$)⁴ concentrations in the Arctic,...

(Revised main manuscript, Line 263-264): By assuming that the heterogeneous uptake is accommodation limited and the γ values of HClO₃ and HClO₄ are similar to that of HCl ($\gamma = 0.2$)^{49,50},...

1.15. An LOD is not reported for the HClO₃ observations at Ny-Alesund (Extended Data Fig. 2).

Reply: The LOD for HClO₃ observations at Ny-Alesund is determined to be $1 \times 10^3 \text{ molecules cm}^{-3}$ (10 min-average, 3σ). This information has now been added to the revised Methods and Extended Data Fig. 2.

(Revised main manuscript, Line 374-376): The detection limit for HClO₃ measurements in Ny-Ålesund was calculated to be $1 \times 10^3 \text{ molecules cm}^{-3}$ (10 min-average, 3σ).

Extended Data Fig. 2 | Observation of HClO₃ at Ny-Ålesund. Example of the HClO₃ concentration data measured with a nitrate-Cl-API-TOF instrument at Ny-Ålesund, Svalbard (78° 55' N, 11° 56' E), from 28 March to 30 May 2017 (spring). The measurements occurred at the atmospheric observatory, Gruebadet, located 2 km southeast of Ny-Ålesund (refer to the map in Fig. 1). The dashed-line represents the detection limit for HClO₃. The uncertainty of HClO₃ and HClO₄ measurements was estimated to be at least a factor of two (see Methods). Details on the site (description), instrumentation setup and calibration can be found in Beck *et al.*⁴⁴.

Reviewer #2 (Remarks to the Author):

2.1. The detection of chlorine oxyacids and the discussion of their impact on Arctic oxidation chemistry is a key issue of broad interest to the environmental chemistry community. The study convincingly presents very low concentrations – a significant achievement. The detection of the oxyacids appears sound to me, and I am impressed that data from 3 different sites have been included. Further, to assess the environmental importance, UV absorption and photolysis are estimated and discussed for the first time. This makes the findings attractive to a broader audience in the environmental science community.

Although there have been significant additions to the manuscript, some major doubts remain concerning the arguments made surrounding the conclusions.

Reply. We thank reviewer for the constructive suggestions, and below we address the specific recommended suggestions.

2.2. Flux estimates and some indication of mixing and boundary layer height need to be discussed when dealing with potential surface sources of trace gases.

Reply. Thank you for the comment. We have now plotted the boundary layer height observed during the MOSAiC campaign, which shows that the atmospheric boundary layer is relatively stable throughout the year (Fig. R2.1). As for the Greenland observations, the boundary layer height was most of the time at about 220 m, based on 9 years of data (Gryning *et al.*, under review).

We have added this information to the caption of Fig. 3 to indicate the boundary layer height observed during the MOSAiC campaign.

(Revised main manuscript, caption of Fig. 3): **Fig. 3 | Atmospheric formation and fate of HClO_3 and HClO_4 .** Simplified diagram of the proposed potential formation mechanism of gas-phase HClO_3 (blue) and HClO_4 (pink) in the Arctic boundary layer during springtime after polar sunrise. The produced HClO_3 and HClO_4 can be taken up by the surface of aerosols and converted into ClO_3^- and ClO_4^- , respectively. The deposition of aerosols and/or the direct deposition of gas-phase HClO_3 and HClO_4 onto the ground surface, such as snowpacks, can function as a sink for reactive chlorine in the Arctic troposphere. The reactions are based on the literatures^{2,12-14,65}. The mean boundary layer height was reported to vary between 100 and 200 m during the MOSAiC campaign^{75,79}.

Fig. R2.1 | Atmospheric boundary layer heights measured by the radiosondes during the MOSAiC campaign (Jozef *et al.*, 2022; Maturilli *et al.*, 2021). The circle is the monthly mean value, with standard deviation represents by the shaded area.

References:

- Jozef, G., Cassano, J., Dahlke, S. & de Boer, G. Testing the efficacy of atmospheric boundary layer height detection algorithms using uncrewed aircraft system data from MOSAiC. *Atmos. Meas. Tech.* **15**, 4001-4022 (2022).
- Maturilli, M. *et al.* Initial radiosonde data from 2019-10 to 2020-09 during project MOSAiC. Alfred Wegener Institute, Helmholtz Centre for Polar and Marine Research, Bremerhaven. *PANGAEA*, doi:10.1594/PANGAEA.928660 (2021)
- Gryning, S.-E. Batchvarova, E. Floors, R. Munkel, C. Sørensen, L.L. and Skov, H. Observed aerosol-layer depth at Station Nord in the high Arctic. *Int. J. Climatol.* **Manuscript under review.**

2.3. With all due respect, the quantitative argument for the surface release of HClO_3 and HClO_4 , based on the Henry constant, is wrong. The SI states that one needs 35 nM HClO_4 in the liquid phase to maintain Henry's equilibrium. This cannot be compared to the levels found in ice cores, as these are in reference to the total ice volume, and the Henry coefficient does not give the ice-air equilibrium. Secondly, the amount of perchlorate in snow/ice might be present at high concentrations in aerosol deposits or liquid frequently found in Arctic and Antarctic snow and ice, which could establish the Henry equilibrium.

Reply: The reviewer makes a good point. Due to lack of relevant information, i.e. Henry's coefficient for the ice-air equilibrium, we are not able to establish the quantitative Henry's law equilibrium for detail analysis. We have decided to remove the discussion of section S1 in the previous supplementary information to avoid confusion in revised version of supplementary information.

Besides the large Henry's law constants and low vapour pressure reported in the main text that may point to their preference to remain in liquid phase, our rationalization for HClO_3 and HClO_4 sticking to the surface originates from a study of Zhong *et al.* (2020) which shows that HOCl sticks preferentially to the ice/liquid surface via the $-\text{OH}$, interacting strongly via hydrogen bonds with water (H_2O). As noted in the text, the detected isomers structure are likely in the form of HOClO_2 and HOClO_3 (Francisco and Sander, 1996; Zhu and Lin, 2001), may behave similarly to the HOCl which has strong interaction with H_2O on the ice/liquid interface. Indeed, previous studies of perchlorates in the atmospheric aerosol and snow of Antarctica have implied the atmospheric production origins, and suggested the possibility of post-depositional loss of snowpack perchlorate, which is unlikely from the loss as gas phase HClO_4 and photolysis process (Jiang *et al.* 2016; 2021). Therefore, even with the possibility of significant perchlorate concentrations in the snow and aerosol, it is unlikely that the snow or aerosol will be the dominant source of gas-phase HClO_4 (and HClO_3) during the springtime. We would like to emphasize that the quantification of sources and sink pathway of these compounds is not the main objective of this study, instead we report the occurrence in the atmosphere of the newly found HClO_3 and HClO_4 and their potential chemical processes in the Arctic boundary layer, and certainly further work is needed to unravel the sources and fate of HClO_3 and HClO_4 in the Arctic environment. We have revised the wording in the main text to tone down statements and add in relevant information.

(Revised main manuscript, Line 170-172): These low vapour pressure and high K_H suggest that the formed ClO_3^- and ClO_4^- on the aerosols or snow surface are unlikely being emitted directly as gas-phase HClO_3 or HClO_4 into the atmosphere.

(Revised main manuscript, Line 271-273): In fact, our hypothesis is supported by previous studies in polar regions that have measured a considerable amount of ClO_3^- and ClO_4^- in ice cores^{21,22}, snow⁵¹ and aerosols⁵², where atmospheric sources are strongly implicated.

References:

- Zhong, J., Zhang, W., Wu, S., An, T. & Francisco, J. S. Molecular interaction and orientation of HOCl on aqueous and ice surfaces. *J. Am. Chem. Soc.* **142**, 17329-17333 (2020).
- Francisco, J. S. & Sander, S. P. Structures, relative stabilities, and vibrational spectra of isomers of HClO_3 . *J. Phys. Chem.* **100**, 573-579 (1996).
- Zhu, R. & Lin, M.-C. Ab initio study of ammonium perchlorate combustion initiation processes: unimolecular decomposition of perchloric acid and the related $\text{OH} + \text{ClO}_3$ reaction. *Phys. Chem. Comm.* **4**, 127-132 (2001).

- Jiang, S., Cox, T. S., Cole-Dai, J., Peterson, K. M. & Shi, G. Trends of perchlorate in Antarctic snow: Implications for atmospheric production and preservation in snow. *Geophys. Res. Lett.* **43**, 9913-9919 (2016).
- Jiang, S., Shi, G., Cole-Dai, J., An, C. & Sun, B. Occurrence, latitudinal gradient and potential sources of perchlorate in the atmosphere across the hemispheres (31°N to 80°S). *Environ. Int.* **156**, 106611 (2021).

2.4. I can't follow the argument that snow is irrelevant because perchlorates are also found in summer (line 170). Firstly, it is unclear where the precursors come from; the authors give snowpack emission as a source (line 180). Secondly, I find it surprising that the authors assume one source and production route dominating the entire data set. There could be different sources in winter and summer – again, looking at the boundary layer height and estimating emissions might be helpful.

Reply: Sorry that we do not understand what do the reviewer meant by perchlorates are also found in summer. From the Extended Data Fig. 2, we can see that the HClO_3 and HClO_4 concentrations are relatively low (near the detection limits of our instrument) in winter (i.e., February) and summer (i.e., June). This shows that the HClO_3 and HClO_4 were not directly emitted to the atmosphere when the Arctic is covered by snow during winter. To avoid further confusion, we have revised the sentence to the following:

(Revised main manuscript, Line 172-174): This is further supported by the detection of low HClO_3 and HClO_4 atmospheric concentrations in winter when the Arctic is covered by snow (see Extended Data Fig. 1).

We agree with the reviewer that there could be different sources of precursors/formation mechanism of HClO_3 and HClO_4 in winter and summer. We would like to clarify that the proposed mechanism in our paper is mainly for the Arctic springtime, when the bromine chemistry is active, and we do not exclude the possibility of different sources or mechanism in other season. However, detail analysis of the sources or mechanism (if exists) of different seasons is out of the scope for the current study. Therefore, we have revised the following sentence in the main text to make it clear that we are proposing a more likely mechanism in Arctic springtime.

(Revised main manuscript, Line 182-183): Here, we propose a more likely formation mechanism of HClO_3 and HClO_4 over the Arctic environment during springtime, as illustrated in Fig. 3.

Minor

2.5. Line 283, Can the authors please give more details on the freezing point depression – how many degrees would you expect based on the nM concentrations? Is that significant relative to the larger concentration of sea salt?

The same holds for toxicity. Why exactly are those low concentrations toxic?

Reply: We thank the reviewer for raising these questions. It is true that other than their potential impacts on the atmospheric chlorine oxidation chemistry, the short-term environmental threats of our observed HClO_3 and HClO_4 remains uncertain. Nevertheless, previous studies of chlorate (ClO_3^-) or perchlorate (ClO_4^-) in the polar regions may provide some insights to their potential environmental impacts over the long run.

As mentioned in the main text, recent studies have hypothesized that the potential formation of HClO_3 and HClO_4 in the lower atmosphere through observations of significant ClO_3^- and ClO_4^- levels in snow and Arctic ice core samples (Dasgupta *et al.*, 2005; Furdui and Tomassini, 2010). This suggests that the atmospheric formed HClO_3 and HClO_4 could contribute to the accumulation of the ClO_3^- and ClO_4^- in polar ice. According to Kounaves *et al.* (2010), a possible implication for the presence of ClO_4^- in various ice caps is the ability of perchlorate salts to significantly suppress the freezing point of water. The alkaline earth salts of ClO_4^- have eutectic points that can form brines and depress the freezing point of water to as low as -70°C (Pestova *et al.*, 2005). They hypothesized that continuous deposition of ClO_4^- on polar ice sheets over tens of millions of years (e.g., Antarctica and Greenland) could plausibly result in the concentration of its salts in the basal ice layers.

Another study by Acevedo-Barrios *et al.* (2022) reported that the significant levels of ClO_4^- found in Antarctic marine sediment samples could possibly be related to their natural origin in the atmosphere (such as atmospheric formation or from volcano activities). Based on the potential effects on amphibian and fishes, they proposed that the ClO_4^- levels measured in the Antarctic marine sediments could present a risk to resident biota. This study suggests that the accumulation of ClO_4^- in the marine sediment could be a risk to the resident biota.

In other words, despite the low concentration, we can imply that our observed HClO_3 and HClO_4 , once deposited to the surface, may contribute to the accumulation of the ClO_3^- and ClO_4^- in polar ice over the time. The long-period accumulations of the ClO_3^- and ClO_4^- may potentially pose threats to the environment or risk to biota. Of course, more future work is needed to evaluate the actual impacts of HClO_3 and HClO_4 .

To reduce the uncertainty, we have decided to tone down the statement on their potential environmental impacts in the main text as follows:

(Revised main manuscript, Line 99-101): Therefore, the atmospheric occurrence of chlorine oxyacids could enhance the chlorine sink, thereby affecting the oxidation capacity of the atmosphere and potentially posing environmental threats once deposited to the Earth's surface.

(Revised main manuscript, Line 284-287): Furthermore, once HClO_3 and HClO_4 deposit on the ground surface (i.e., snowpack and sea-ice), they may have environmental implications as their ions, ClO_3^- and ClO_4^- , can accumulate in the polar ice and marine sediment, potentially suppressing the freezing point of water, and may present a toxicity risk to resident biota^{57,58}

References:

- Dasgupta, P. K. *et al.* The origin of naturally occurring perchlorate: The role of atmospheric processes. *Environ. Sci. Technol.* **39**, 1569-1575 (2005).
- Furdui, V. I. & Tomassini, F. Trends and sources of perchlorate in Arctic snow. *Environ. Sci. Technol.* **44**, 588-592 (2010).
- Kounaves, S. P. *et al.* Discovery of natural perchlorate in the Antarctic Dry Valleys and its global implications. *Environ. Sci. Technol.* **44**, 2360-2364 (2010).
- Pestova, O. N., Myund, L. A., Khripun, M. K. & Prigaro, A. V. Polythermal Study of the systems $\text{M}(\text{ClO}_4)_2 \cdot \text{H}_2\text{O}$ ($\text{M}^{2+} = \text{Mg}^{2+}, \text{Ca}^{2+}, \text{Sr}^{2+}, \text{Ba}^{2+}$). *Russ. J. Appl. Chem.* **78**, 409-413 (2005).
- Acevedo-Barrios, R., Rubiano-Labrador, C. & Miranda-Castro, W. Presence of perchlorate in marine sediments from Antarctica during 2017–2020. *Environ. Monit. Assess.* **194**, 102 (2022).

2.6. The authors may comment on why no atmospheric chemistry model was used to discuss the importance of the many sources and sink processes. I'm kind of surprised that this has not been done.

Reply: Thank you for the suggestion. However, we do not think using atmospheric chemistry model is appropriate at this stage of study for two major reasons. First, we do not know on the actual mechanism (including the reaction rates) that involve in the formation and loss of these oxyacids. We are proposing the potential mechanism based on findings from previous studies, and by using the 'incomplete' mechanism to evaluate the source and sink processes will provide bias results that can lead to wrong conclusions. Second, we do not have the concurrent measurements of the chlorine and bromine species that can be the precursors of HClO_3 and HClO_4 . If we opt to use the chlorine and bromine emission inventories from the current models, this could largely increase the uncertainty of the results that again may lead to making inappropriate conclusions. We believe that even without the quantitative analysis on the sources and sink processes; it will not change the novelty of our paper. Perhaps, our observations of HClO_3 and HClO_4 are the first step to proof the existence of these chemical compounds in the Arctic boundary layer during springtime, and urging for more future studies to better understand their fates and roles in the atmosphere, before we can establish a more reliable modeling work.

Reviewer #3 (Remarks to the Author):

3.1. Overall, the authors have addressed my major technical questions. I have a few minor comments below. I am still on the fence about importance of this finding to the environment. I would have been happier if the authors note emphasize their detection of these highly oxidized chlorine species, and the likelihood that they are produced in the gas phase through a set of complex reactions. That is a sufficient advance to publish this paper. But, the authors have opted to emphasize the atmospheric significance.

I am OK with the way it is now.

Reply: Thank you very much for the positive comments.

A couple minor comments:

3.2. It is not true that HClO_3 and HClO_4 cannot travel long distances in the atmosphere. I would buy it for near surface transport, but they can be transported on/in particles and also via strat-trop exchange especially in the polar region where there is a massive descent from the stratosphere. I suggest that they be clear about their assertion that it is not transported in the boundary layer....

Reply: Thank you for the comment. We agree with the reviewer that HClO_3 and HClO_4 can be transported long-distance on the aerosol or via stratosphere-troposphere exchange in the polar region. For the long-distance transportation of perchlorate aerosols, we did not exclude this process, but we have discussed in the text that aerosols are unlikely the predominant source of gas-phase HClO_3 and HClO_4 . The possibility of HClO_3 and HClO_4 -laden air mass descent from the stratosphere may exist; however, we do not think that this process is the dominant source for the observed gas-phase HClO_3 and HClO_4 in Arctic boundary layer in springtime. As has been reported in another study during the MOSAiC campaign (Benavent *et al.*, 2022), the air masses observed in spring (March and April 2020) were in close contact with the surface (below 100 m), including the sea ice (see results in Fig. R3.1). Indeed, the absence isotopic peak of ^{36}Cl for HClO_3 and HClO_4 may also suggest a possible lack of cosmogenic chlorine. Therefore, we have toned down the sentence on the possibility of the stratospheric influence.

(Revised main manuscript, Line 326-328): The absence of a ^{36}Cl peak in the spectrum may suggest a possible lack of influence from the stratospheric air mass⁶².

Fig. R3.1 | Time series of the surface influence for the whole domain, as well as for areas covered with all sea ice and, separately, first-year sea-ice (Benavent *et al.*, 2022).

References:

- Benavent, N. *et al.* Substantial contribution of iodine to Arctic ozone destruction. *Nat. Geosci.* **15**, 770-773 (2022).

3.3. I would have toned down the statement that their measurements are a lower limit for the HClO₃ and HClO₄ concentration. This is not logical. They have not calibrated directly, they are assuming a Langevin (association) reaction rate coefficient (which begs the question of the capture cross section), and there are mass effects in detection. I think that they can claim that their numbers are within a factor of two or three of the real value.

Reply: We agree with the reviewer to tone down the statement to be less certain on the lower limit assumption. It is true that we have no direct calibrations, but we have shown in the text that the detection of HClO₃ and HClO₄ by our instrument is very likely as efficient as the detection of H₂SO₄ by NO₃⁻, whose reaction rate is expected to occur within the kinetic limit range (Viggiano *et al.*, 1997). The kinetic-limited detection means that the H₂SO₄ is detected at maximum sensitivity by the instrument. If the detected HClO₃ and HClO₄ clusters are not charged as efficiently as H₂SO₄ by NO₃⁻, it could lead to a lower sensitivity, thereby underestimating the concentration of HClO₃ and HClO₄. We have revised the statement accordingly.

(Revised main manuscript, Line 381-384): If the detected HClO₃ and HClO₄ clusters are not charged as efficiently as H₂SO₄, it could lead to underestimating the concentration of HClO₃ and HClO₄. The sum uncertainties of the HClO₃ and HClO₄ measurement from the collision limit of the target compound with its charger ions and potential inlet losses were predicted to be at least a factor of two.

References:

- Viggiano, A. A., Seeley, J. V., Mundis, P. L., Williamson, J. S. & Morris, R. A. Rate constants for the reactions of XO₃⁻(H₂O)_n (X = C, HC, and N) and NO₃⁻(HNO₃)_n with H₂SO₄: Implications for atmospheric detection of H₂SO₄. *J. Phys. Chem. A* **101**, 8275-8278 (1997).

3.4. I don't understand their assertion in the rebuttal that the aerosol composition would not affect their analysis. What if the aerosols are very acidic? They did not justify why they "believe" this assertion to be true.

Reply: Our previous reply stated that there is aerosol composition measurement from an aerosol mass spectrometer (AMS) during the MOSAiC campaign, but unfortunately, the aerosol compositions data are not publicly available yet. We would like to emphasize that the novelty of our study is the detection of HClO₃ and HClO₄ in the Arctic boundary layer for the first time, and our discussion mainly focus on the gas-phase and heterogeneous uptake reactions. Without the aerosol compositions data at this stage, it will not affect our detection of HClO₃ and HClO₄ and report on their presence in the atmosphere. Even if we have the aerosol chemical compositions of PM₁ (particulate matter with diameter below 1 μm) from the AMS measurement, they may be just an extra information of typical chemical substance in the ultrafine particle, which will not change the overall focus of our paper.

Yes, we agree with the reviewer that there is some uncertainty of the partitioning of HClO₃ and HClO₄ due to pH and/or acid-base equilibria in the aerosol/snow surface as has been seen in HCl and Cl⁻ system. From the literature, so far, there is no evidence showing that ClO₃⁻ and ClO₄⁻ will be released as gas-phase HClO₃ and HClO₄ to the atmosphere under ambient conditions. Our results in the main text has demonstrated no direct relationship between HClO₃ (or HClO₄) and our measured H₂SO₄ concentrations (coefficient of determination, R² ≤ 0.04) during both the Greenland and MOSAiC campaigns (see Extended Data Fig. 4 in the main text). This may indirectly imply that the pH variations of aerosol (assuming H₂SO₄ will be converted to SO₄²⁻ efficiently and changed the pH of the aerosol) may have no effect on the levels of HClO₃ and HClO₄ observed in our study.

Of course, it would be useful for future study to include as much concurrent measurements as possible (including gaseous precursors and aerosol physicochemical properties) to better understand the chemical behavior HClO₃ and HClO₄ in the Arctic boundary layer.

3.5. I don't understand why they insist that OH is the main atmospheric oxidant for hydrocarbons etc. Even for olefins, their reaction with Ozone can be a major pathway. Lastly, OH is a daytime species (for the most part) and so it is not the main oxidant under darkness and other conditions.

Reply: The reviewer makes a good point here. We have revised the wording of the statement.

(Revised main manuscript, Line 68-69): Chlorine atoms (Cl) are also a strong oxidant in the polar troposphere, where the levels of hydroxyl radicals, another major atmospheric oxidant, are relatively low⁴.

3.6. I don't understand what lines 281-284 mean. What does the phrase "ions, ClO₃⁻ and ClO₄⁻, are extremely mobile in water? Their diffusion coefficients are large? What does "mobile" mean here.

Reply: Sorry for causing the confusion. Cao *et al.* (2019) and references therein stated that the high water solubility and the poor adsorption of ClO₄⁻ to typical soil minerals and organic carbon make this oxyanion highly mobile in surface water and groundwater. In other words, the ClO₄⁻ can be easily transport in the water body. Since the phrase is irrelevant to our study, we have removed it from the revised statement.

(Revised main manuscript, Line 284-287): Furthermore, once HClO₃ and HClO₄ deposit on the ground surface (i.e., snowpack and sea-ice), they may have environmental implications as their ions, ClO₃⁻ and ClO₄⁻, can accumulate in the polar ice and marine sediment, potentially suppressing the freezing point of water, and may present a toxicity risk to resident biota^{57,58}.

References:

- Cao, F. *et al.* Worldwide occurrence and origin of perchlorate ion in waters: A review. *Sci. Total Environ.* **661**, 737-749 (2019).

3.7. Extended Date Fig. 9: Don't they mean DeltaG(zero)? DeltaG= DeltaG(zero) + RT lnQ, where Q is the reaction coefficient.

Reply: We agree with the reviewer that ΔG in Extended Data Fig. 9 is misleading. The y-axis denotes the free energy of the cluster and products relative to the reactants. Thus, the y-axis label of the figure has been changed to 'Relative free energy (kcal mol⁻¹)'.

Extended Data Fig. 9 | Cluster formation free energy. Potential energy surface (PES) indicating the cluster formation free energies for (a) NO₃⁻ clustering of HClO₃, (b) NO₃⁻ clustering of HClO₄, and (c) NO₃⁻ clustering of H₂SO₄. The cluster geometries are included: green = chlorine, red = oxygen, white = hydrogen, blue = nitrogen, and yellow = sulfur atoms.

Reviewer #1 (Remarks to the Author):

I thank the authors for their careful attention to the reviewers' suggestions in this round of their revisions. I found the revisions overall to be very good, addressing the reviewers' concerns and improving the manuscript in the process. Most notably, the authors' revisions to the technical aspects of the work (LODs, uncertainty, further investigation of the Br- signal, etc) have significantly improved the manuscript by more clearly communicating these data for improved interpretation by the reader. Critically, the authors now clearly indicate the method limits of detection for HClO₃ and HClO₄ and show which data are below LOD, as well as communicate the significant data uncertainties. I only have two minor comments for the revised manuscript:

- Please add Fig R1.2 to the supplemental information, as this figure critically illustrates the justification of the Br- signal as HBr, which is a new and important update to the manuscript.
- In response to Comment 3.2, the authors added a sentence about the absence of a ³⁶Cl peak suggesting a lack of stratospheric influence. However, given how close the signals were to the noise for even the ³⁵Cl and ³⁷Cl signals, do the authors predict that the ³⁶Cl would be discerned if there were influence? i.e. Is the method even sensitive enough to measure the ³⁶Cl peak? If the method is not sufficiently sensitive, then the added sentence cannot be justified with the data available.

Reviewer #3 (Remarks to the Author):

I am happy with the overall revisions and how they answered my queries.

In the process of revision, they did two things that are problematic.

1. In response to reviewer 1, they are now mixing up the detection and quantification. Their mass spectral detection has a S/N issue. But, quantification depends on both the detection S/N and the uncertainty in the quantification because the authors do not have a suitable calibration. So, detection limits (if posed as quantitative estimates) are not right! The authors should clarify this issue.

2. In response to one of my comments they made a change that does not make sense to me. In Line 284-287 they now say "Furthermore, once HClO₃ and HClO₄ deposit on the ground surface (i.e., snowpack and sea-ice), they may have environmental implications as their ions, ClO₃⁻ and ClO₄⁻, can accumulate in the polar ice and marine sediment, potentially suppressing the freezing point of water, and may present a toxicity risk to resident biota^{57,58}". I don't think that the concentrations of these ions would be significant enough to suppress the freezing point to any significant extent. Please do a Raoult's law calculation and you will see that you need a lot of stuff down there. Please take out the phrase "potentially suppressing the freezing point of water." That is it from me.

We thank the reviewers for their careful reading of the manuscript and for sending us their remaining suggestions. Our point-by-point responses are listed in dark blue wording and the corresponding changes made to the revised manuscript are in light blue. The changes in the revised manuscript (track-change version) have been highlighted in light blue.

Reviewer #1 (Remarks to the Author):

I thank the authors for their careful attention to the reviewers' suggestions in this round of their revisions. I found the revisions overall to be very good, addressing the reviewers' concerns and improving the manuscript in the process. Most notably, the authors' revisions to the technical aspects of the work (LODs, uncertainty, further investigation of the Br⁻ signal, etc) have significantly improved the manuscript by more clearly communicating these data for improved interpretation by the reader. Critically, the authors now clearly indicate the method limits of detection for HClO₃ and HClO₄ and show which data are below LOD, as well as communicate the significant data uncertainties. I only have two minor comments for the revised manuscript:

1.1. Please add Fig R1.2 to the supplemental information, as this figure critically illustrates the justification of the Br⁻ signal as HBr, which is a new and important update to the manuscript.

Reply. Thanks for the suggestion. We have added Fig R1.2 to the revised supplemental information.

Supplementary Fig. S10 | Observation of NO₃(HBr)⁻ signal. (a) The peak of NO₃(H⁷⁹Br)⁻ together with its isotope peak (NO₃(H⁸¹Br)⁻), compared to the zero measurement (red line), detected by nitrate CI-API-TOF. (b) An example of the NO₃(HBr)⁻ versus Br⁻ data obtained during the MOSAiC from 18 to 30 April 2020.

1.2. In response to Comment 3.2, the authors added a sentence about the absence of a ³⁶Cl peak suggesting a lack of stratospheric influence. However, given how close the signals were to the noise for even the ³⁵Cl and ³⁷Cl signals, do the authors predict that the ³⁶Cl would be discerned if there were influence? i.e. Is the method even sensitive enough to measure the ³⁶Cl peak? If the method is not sufficiently sensitive, then the added sentence cannot be justified with the data available.

Reply: Thank you for the suggestion. We agree with the reviewer that our method is maybe not sensitive enough (or capable) to detect the signal of extremely low ^{36}Cl in the environment (if exists). To be on the safer side, we have removed the sentence from the text to avoid further confusion.

Reviewer #3 (Remarks to the Author):

I am happy with the overall revisions and how they answered my queries. In the process of revision, they did two things that are problematic.

3.1. In response to reviewer 1, they are now mixing up the detection and quantification. Their mass spectral detection has a S/N issue. But, quantification depends on both the detection S/N and the uncertainty in the quantification because the authors do not have a suitable calibration. So, detection limits (if posed as quantitative estimates) are not right! The authors should clarify this issue.

Reply: We thank the reviewer for the comment. The reviewer is right that the detections are not truly quantitative, and are based on estimation since we do not have a suitable full calibration system. This has been acknowledged in the original text. We are not aware that the mass spectral detection has an S/N issue. From our mass spectrum, we can see a positive detection ($3\times\text{Sigma}$) of the isotopic compounds for these chlorine species compared to the zero measurements.

3.2. In response to one of my comments they made a change that does not make sense to me. In Line 284-287 they now say "Furthermore, once HClO_3 and HClO_4 deposit on the ground surface (i.e., snowpack and sea-ice), they may have environmental implications as their ions, ClO_3^- and ClO_4^- , can accumulate in the polar ice and marine sediment, potentially suppressing the freezing point of water, and may present a toxicity risk to resident biota^{57,58}." I dont think that the concentrations of these ions would be significant enough to suppress the freezing point to any significant extent. Please do a Raoult's law calculation and you will see that you need a lot of stuff down there. Please take out the phrase "potentially suppressing the freezing point of water."

That is it from me.

Reply: Agree. We have deleted the phrase and revised the sentence as the following:

Furthermore, once HClO_3 and HClO_4 deposit on the ground surface (i.e., snowpack and sea-ice), they may have environmental implications as their ions, ClO_3^- and ClO_4^- , can accumulate in the polar ice and marine sediment, and may present a toxicity risk to resident biota^{57,58}.